# ZERO-SHOT FORECASTING BY SIMULATION ALONE

**Boris N. Oreshkin**[1]**, Mayank Jauhari**[1]**, Ravi Kiran Selvam**[1]**, Malcolm Wolff**[1]**, Wenhao Pan**[1,2]
**Shankar Ramasubramanian**[1]**, Kin G. Olivares**[1]**, Tatiana Konstantinova**[1]**, Andres Potapczynski**[1,3]
**Mengfei Cao**[1]**, Dmitry Efimov**[1]**, Michael W. Mahoney**[1]**, Andrew G. Wilson**[1,3]
[1]Amazon Science [2]University of Washington [3]New York University
`oreshkin@amazon.com`

## ABSTRACT

Zero-shot time-series forecasting holds great promise, but is still in its infancy, hindered by limited and biased data corpora, leakage-prone evaluation, and privacy and licensing constraints. Motivated by these challenges, we propose the first practical univariate time series simulation pipeline which is simultaneously fast enough for on-the-fly data generation and enables notable zero-shot forecasting performance on M-Series and GiftEval benchmarks that capture trend/seasonality/intermittency patterns, typical of industrial forecasting applications. Our simulator, which we call `SarSim0` (**SAR**IMA **Sim**ulator for **Zero**-Shot Forecasting), is based off of a seasonal autoregressive integrated moving average (SARIMA) model as its core data source. Due to instability in the autoregressive component, naive SARIMA simulation often leads to unusable paths. Instead, we follow a three-step procedure: (1) we sample well-behaved trajectories from its characteristic polynomial stability region; (2) we introduce a superposition scheme that combines multiple paths into rich multi-seasonality traces; and (3) we add rate-based heavy-tailed noise models to capture burstiness and intermittency alongside seasonalities and trends. `SarSim0` is orders of magnitude faster than kernel-based generators, and it enables training on *circa* 1B unique *purely simulated* series, generated on the fly; after which well-established neural network backbones exhibit strong zero-shot generalization, surpassing strong statistical forecasters and recent foundation baselines, while operating under strict zero-shot protocol. Notably, on GiftEval we observe a "student-beats-teacher" effect: models trained on our simulations exceed the forecasting accuracy of the `AutoARIMA` generating processes.

## 1 INTRODUCTION

*Zero-shot* learning fixes a pretrained model and predicts on target data with no parameter updates. While this paradigm has reshaped natural language processing and computer vision (Yosinski et al., 2014; Wei et al., 2022), zero-shot time series forecasting is still in its infancy, but it is rapidly gaining interest. Indeed, zero-shot forecasting (Garza & Mergenthaler-Canseco, 2023; Gruver et al., 2023) offers compelling advantages for deployment scenarios, *e.g.*, because using a frozen pre-trained model eliminates target-side selection loops and tuning heuristics. By collapsing rollout to pure inference—no backpropagation, no per-dataset adaptation—zero-shot forecasting meets tight latency and cost envelopes, and it simplifies platform optimization (train once, reuse everywhere), while lowering compute and energy budgets per forecast. It also aligns with privacy and governance constraints: inference on sensitive targets remains strictly local, with no gradient flow or parameter updates that could leak information, easing compliance and audits. While recent work in time series forecasting focuses on architectural novelty (Liang et al., 2024), data is the lifeblood of zero-shot prediction. Curated compilations of real series can be helpful, but they inevitably run into problems, including licensing barriers, limited scale, and domain and cadence biases. Most problematic for zero-shot integrity is *leakage*: overlapping datasets in train/test, as well as target-side hyperparameter search confound evaluation. Using just synthetic data has shown promise for zero-shot forecasting, but it has not yet been able to fully substitute for real data. For example, Ansari et al. (2024) show that augmenting real data with a kernel-based synthetic generation procedure yields better results than using real data alone; however, they find training solely on synthetic data can significantly degrade performance; and Kuvshinova et al. (2024) find that zero-shot pretraining on synthetic data is not comparable relative to using even a small amount of real data.

In this paper, our goal is to bridge this gap and develop an effective synthetic-only training pipeline for zero-shot time series forecasting. Synthetic data offers unique levers: controllable coverage

of seasonalities and sampling rates, programmatic rare events, debiasing, and guaranteed leakage-free generation—things that real compilations cannot easily match. Building on this premise, we provide the following contributions in this work. (1) We propose `SarSim0`, a fast three-stage synthetic time series simulator (see Figure 1) that produces rich, realistic patterns and that is efficient enough to synthesize sequences on-the-fly, thereby enabling pretraining at the scale of billions of series. To start, we first sample stable poles of the characteristic polynomial of a seasonal auto-regressive integrated moving average (SARIMA) process (Section 4.1). SARIMA provides a natural choice as our simulator backbone both because it is deeply rooted in stochastic time series modeling, and because it is fast and expressive, subsuming many other time series models as special cases (Hyndman et al., 2025). We also show that it captures the fidelity of real time series data with rich structure (Figure 3). Then, we introduce a superposition scheme, combining sample paths for multi-seasonality via modulation (Section 4.2). Finally, we add rate-based heavy-tailed noise models to capture burstiness and intermittency (Section 4.3). (2) We provide a detailed empirical evaluation of `SarSim0` (Section 5) on the M-Series (Makridakis et al., 1982; Makridakis & Hibon, 2000; Makridakis et al., 2020; Athanasopoulos et al., 2011) and GiftEval (Aksu et al., 2024) benchmarks. Our scope is zero-shot forecasting in industrial settings, where series are largely driven by complex trend, seasonality, and intermittency / heavy-tail patterns. M-Series and GiftEval, which span multiple domains (Nature, Web/CloudOps, Sales, Energy, Transport, Healthcare, Demographic, Finance, Industry, Macro/Micro Economic) and frequencies (yearly, quarterly, monthly, weekly, daily, hourly, and some sub-hourly), provide a comprehensive testbed for this regime. We find that training exclusively on time series generated by our simulator yields compact, fast, and accurate models that generalize under a strict no-leakage guarantee. We show that, on these benchmarks, simulated data can be competitive with, or even outperform, real-data pretraining, substantially closing the gap between small efficient architectures and large foundation models; and on GiftEval, we see evidence for a "student beats teacher" generalization behavior. On the more regular, relatively short and low-noise business and macroeconomic M-Series, this effect is mixed, with `AutoARIMA` remaining very strong. This indicates that such emergent generalization is dataset- and domain-dependent, and is currently most pronounced on more heterogeneous and noisier benchmarks like GiftEval.

## 2 BACKGROUND

**Univariate quantile forecasting.** Let $y_{1:t} \in \mathbb{R}^t$ be a univariate time series observed up to time $t$, and let $H$ be the forecast horizon. We aim to predict, for each forecast horizon $h = 1, \ldots, H$, a set of $\mathcal{Q}$ conditional quantiles of $y_{t+h}$. We fix quantile levels $\boldsymbol{\tau} := (\tau_1, \ldots, \tau_{\mathcal{Q}})$, $0 < \tau_1 < \cdots < \tau_{\mathcal{Q}} < 1$, and we define a lookback (context) window of length $\ell < T$, $\mathbf{x}_t := y_{t-\ell+1:t} \in \mathbb{R}^\ell$, and forecasting model, $\mathcal{F}_\psi : \mathbb{R}^\ell \to \mathbb{R}^{H \times \mathcal{Q}}$, producing quantile matrix, $\mathbf{x}_t \mapsto \widehat{\mathbf{Y}}_{T+1:T+H}^{(\boldsymbol{\tau})}$, with elements $\hat{y}_{t+h}^{(\tau)}$.

**Learning via Empirical Risk Minimization (ERM) with multi-horizon, multi-quantile loss.** From a training corpus, we extract $N$ supervised examples by rolling windows across one or more univariate series, $\mathcal{D} = \left\{ (\mathbf{x}_{t_i}, y_{t_i+1:t_i+H}) \right\}_{i=1}^N$. We further define the quantile loss at level $\tau$ as:

$$\rho_\tau(y, \hat{y}) = \tau\,(y - \hat{y})_+ + (1 - \tau)(\hat{y} - y)_+ \quad \text{with} \quad u_+ := \max\{u, 0\}.$$

The multi-horizon, multi-quantile loss for a single example $i$ is:

$$\mathcal{L}_{H,\mathcal{Q}}\left(y_{T+1:T+H},\, \widehat{\mathbf{Y}}_{T+1:T+H}^{(\boldsymbol{\tau})}\right) := \frac{1}{H\mathcal{Q}} \sum_{h=1}^H \sum_{\tau \in \boldsymbol{\tau}} \rho_\tau\left(y_{T+h},\, \hat{y}_{T+h}^{(\tau)}\right). \tag{1}$$

The ERM objective optimizing the network parameters $\psi$ to approximate the conditional quantile function $\tau \mapsto F_{y_{t+h}|\mathbf{x}_t}^{-1}(\tau)$ on the grid $\boldsymbol{\tau}$ uniformly over horizons $h = 1, \ldots, H$ is then given by:

$$\psi^\star \in \arg\min_{\psi \in \Psi} \frac{1}{N} \sum_{(\mathbf{x},\mathbf{y}) \in \mathcal{D}} \mathcal{L}_{H,\mathcal{Q}}\left(\mathbf{y},\, \mathcal{F}_\psi(\mathbf{x})\right).$$

**Zero-shot forecasting.** By *zero-shot forecasting*, we mean that we freeze all model parameters after pre-training on a source corpus, and we then condition on a target series to make forecasts, while keeping the model parameters frozen. More precisely, following the definition of the zero-shot forecasting task from Oreshkin et al. (2021) and Olivares et al. (2025), let $\mathcal{D}^{(S)}$ be a *source* training corpus and let $\mathcal{D}^{(T)}$ be a *target* test corpus constructed from series disjoint from $\mathcal{D}^{(S)}$ (no entity or segment overlap). A forecasting model $\mathcal{F}_\psi$ is first trained on $\mathcal{D}^{(S)}$ (all model selection

is performed using splits of $D^{(S)}$ only), yielding parameters $\psi^\star$. *Zero-shot evaluation* on $\mathcal{D}^{(T)}$ proceeds by applying the *frozen* (weights, normalizers, embeddings) predictor $\mathcal{F}_{\psi^\star}$ to each target series independently, *one at a time*.

## 3 RELATED WORK

**Transfer Learning.** Early approaches to transfer learning in forecasting focused on frequency-specialized models (*i.e.*, one model for monthly frequency, one model for quarterly frequency, etc). Success was measured by the model's ability to outperform statistical baselines: `AutoARIMA` (Hyndman & Khandakar, 2008), `ETS` (Holt, 1957), and `Theta` (Fiorucci et al., 2016), when applied to large time series corpora. Progress in this area has been driven by adopting cross-learning via training of global models on large time series collections to extract shared patterns; and this approach underpins the success of top-performing models in competitions like `M4` and `M5` (Smyl, 2019; Anderer & Li, 2022), as well as industry models such as `DeepAR`, `MQCNN` and `TFT` (Salinas et al., 2020; Wen et al., 2017; Lim et al., 2021). Zero-shot forecasting in time series emerged from transfer-learning formulations such as (Oreshkin et al., 2021), which showed that a frequency-specific source-trained model can be deployed to unseen series without target-side updates.

**Foundation Models.** Subsequent work moves beyond frequency-specific forecasters toward foundation-style pretraining, leveraging large sequence models for zero-shot prediction. `TimeGPT` (Garza & Mergenthaler-Canseco, 2023) pioneered the use of pre-trained transformers for forecasting, and `LLMTime` (Gruver et al., 2023) extended this approach by adapting `Llama` (Touvron et al., 2023). `TimesFM` adopts patch tokens and a decoder-only backbone for fast multi-step generation (Das et al., 2024); `MOIRAI` (Woo et al., 2024) couples patching with any-variate projections and mixture-likelihood heads for cross-frequency robustness; and `LagLlama` retains a decoder-only design with lag features and parametric distribution heads (Rasul et al., 2024). `Chronos` adopts a different approach by quantizing series values and uses encoder-decoder LMs with next-token cross-entropy for probabilistic decoding (Ansari et al., 2024). Beyond attention, `TTM` targets accuracy-latency Pareto efficiency with all-MLP mixers (Ekambaram et al., 2024). Recent work has highlighted subtle tradeoffs as training knobs are varied for different classes of models (Yu et al., 2025b;a).

**Synthetic time series data.** `ForecastPFN` (Dooley et al., 2023) generates synthetic training corpora by composing simple priors over trend and seasonal components, combined with multiplicative Weibull noise. `KernelSynth` from `Chronos` (Ansari et al., 2024) augments real datasets with GP-kernel-based synthetic samples. `TimePFN` (Taga et al., 2025) extends these approaches to multivariate settings using GP-kernel compositions and channel mixing. Unlike other approaches, our simulator is grounded in stochastic time series dynamics (stable SARIMA) with broad expressive power (complemented with hierarchical seasonality and heavy-tailed noisers) rather than hand-crafted seasonal-trend templates or slow kernel synthesis. This yields high-fidelity temporal dependence, heteroscedasticity, and burstiness, while remaining orders of magnitude faster. Consequently, we pretrain foundation models *exclusively* on simulated data at scale; the resulting models exhibit strong zero-shot generalization across heterogeneous benchmarks, surpassing results of prior synthetic approaches (*e.g.*, `ForecastPFN`, `KernelSynth`).

## 4 SARSIM0: TIME SERIES SIMULATOR PIPELINE

A central hypothesis of this work is that high-quality synthetic data can unlock scalable pretraining for zero-shot forecasting. In domains such as computer vision and natural language processing, foundation models are fueled by massive, diverse corpora. Time series, by contrast, suffer from (relatively) limited public datasets, licensing constraints, and pervasive risks of statistical leakage. A realistic, efficient simulator offers an alternative path: it enables pretraining at scale while guaranteeing leakage-free evaluation. To this end, we propose a modular simulation pipeline, which we call `SarSim0` (**SAR**IMA **Sim**ulator for **Zero**-Shot Forecasting). `SarSim0` can be formalized as:

$$y_{1:T} = \mathbb{N} \circ \mathbb{I} \circ \mathbb{S}(\epsilon). \tag{2}$$

Here $\mathbb{S}$ is a structured base-signal generator, $\mathbb{I}$ is the interaction component that consumes tuples of structured signals and composes them into richer waveforms via parameterized interaction patterns (*e.g.*, additive mixing or multiplicative modulation); and $\mathbb{N}$ overlays the structured output with stochastic perturbations, including heavy-tailed noise to capture burstiness and intermittency. The overall `SarSim0` system diagram, along with generated examples is shown in Figure 1. The design

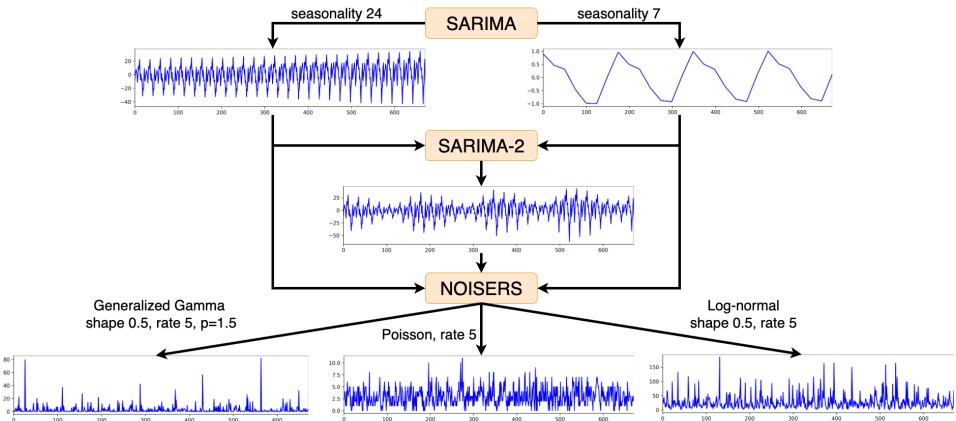

Figure 1: **SarSim0 simulator pipeline.** Top: two base components are generated by SARIMA with AR (and seasonal) roots sampled via the characteristic polynomial inside the stability region, yielding well-behaved paths at seasonalities $s = 24$ and $s = 7$. Middle: a SARIMA-2 superposition/modulation block combines the components to produce a double-seasonal series with rich cross-frequency structure. Bottom: a noise module injects heavy-tailed disturbances: *e.g.*, Poisson spikes, generalized-gamma bursts, and log-normal volatility capturing burstiness, intermittency, and realistic deviations from Gaussian noise.

is guided by three requirements for effective pretraining: (i) structural fidelity to recurring motifs in real series (seasonality, trends, intermittency); (ii) scalability to billions of samples without storage overhead; and (iii) diversity sufficient to support generalization across heterogeneous benchmarks.

In this paper, for the base generator of `SarSim0`, we adopt SARIMA, because it combines deep statistical time series grounding with broad expressive power and computational efficiency. First, SARIMA subsumes many statistical models as special cases: it recovers Simple Exponential Smoothing, linear trend and seasonal Holt's method, as well as damped-trend Holt-Winters variant (Hyndman et al., 2025). Additionally, random walk (Naïve), seasonal Naïve, Airline model (Box et al., 2015), and `Theta` method (Hyndman & Billah, 2003a) formulations are all nested within the SARIMA family, making it a unifying lens for widely adopted forecasting approaches (Hyndman et al., 2025). Second, SARIMA underlies the `AutoARIMA` procedure (Hyndman & Khandakar, 2008), which remains one of the strongest zero-shot baselines in forecasting competitions and applied practice. By grounding `SarSim0` in SARIMA, we inherit the statistical foundations of classical models and retain a direct link to `AutoARIMA` as a natural "teacher", while aligning with and complementing the inductive biases of neural "student" architectures. We will see that `SarSim0`-trained models outperform `AutoARIMA`; moreover, as Figure 3 shows, SARIMA reproduces the statistical structure of real-world time series, providing a credible training signal.

## 4.1 SARIMA SIMULATOR

Without loss of generality, we start the exposition with the *Auto-Regressive Moving Average* (ARMA) process of order $p', q$ that can be defined as:

$$y_t - \alpha_1 y_{t-1} - \cdots - \alpha_p y_{t-p'} = \varepsilon_t + \vartheta_1 \varepsilon_{t-1} + \cdots + \vartheta_q \varepsilon_{t-q}. \tag{3}$$

Here $\alpha_{1:p'}$ are auto-regressive coefficients, $\vartheta_{1:q}$ are moving average coefficients, and $\varepsilon_{t-q:t}$ are zero-mean i.i.d. innovations. With the lag operator $L$ defined as $Ly_t = y_{t-1}$, this process can be written compactly in polynomial form:

$$\left(1 - \sum_{i=1}^{p} \phi_i L^i\right)(1 - L)^d y_t = \left(1 + \sum_{i=1}^{q} \vartheta_i L^i\right)\varepsilon_t, \tag{4}$$

where $p = p' - d$ and coefficients $\phi$ are derived from coefficients $\alpha$ to factor out $d$ unit roots of the polynomial, corresponding to the finite difference time domain integrator, $y_t \leftarrow y_t + y_{t-1}$. Consequently, ARMA with integrator ($d > 0$) is called ARIMA (integrated ARMA).

In principle, ARIMA could be used directly as a time series simulator, by randomly generating initial state $y_0$, orders $p', q$; coefficients $\alpha, \vartheta$; excitation noise $\varepsilon$, and unrolling Equation (3) through time. However, as shown in Figure 2 (bottom), setting the coefficients arbitrarily destabilizes the system and results in divergent nonsensical time series that are not useful for training a forecasting

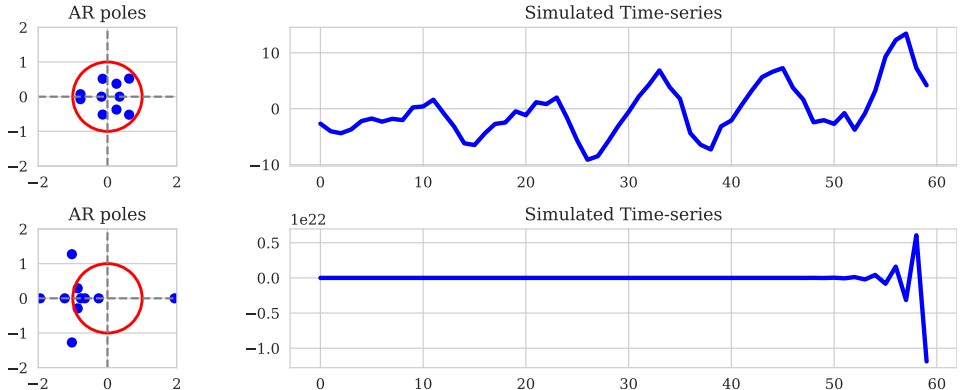

Figure 2: **Sampling of SARIMA poles by `SarSim0`.** The SARIMA order-10 AR process poles are shown along with the unit circle on the left. The resulting generated processes with these poles are shown on the right. The top pane shows poles sampled according to the proposed procedure, resulting in a realistic looking time series. The bottom pane shows the result of unconstrained random generation of AR coefficients, resulting in a divergent time series that is useless from the perspective of training time series foundation models.

model. To circumvent this stability issue, we propose instead to operate directly in the model's pole space, guaranteeing valid process dynamics by construction. In particular, there is a link between the summation-based polynomial form (4) and the product-based pole representation:

$$\phi(L) = 1 - \sum_{i=1}^{p} \phi_i L^i = \prod_{i=1}^{p}(1 - \varphi_i L). \tag{5}$$

Well-behavedness of the AR part is guaranteed if the poles, $\varphi_i$, lie within unit circle, $|\varphi_i| < 1$, (Hamilton, 1994, Chapter 3). Thus, we propose to sample the poles of the AR and MA transfer functions, $\{\varphi_i\}_{i=1}^{p}$ and $\{\vartheta_j\}_{j=1}^{q}$ respectively, and then derive the system coefficients $\alpha$ and $\vartheta$ from the pole representation using product expansion (*e.g.*, numpy's np.poly). A typical realization of the process sampled using this approach is shown in Figure 2 (top). In contrast to Figure 2 (bottom), all poles of the AR system are located within unit circle, resulting in a well-behaved simulation result.

To complete the polynomial specification of the base SARIMA, we include the seasonal part with parameters $s$ (period), $D$ (integration order), $P$ (seasonal AR order), $Q$ (seasonal MA order):

$$\left(1 - \sum_{i=1}^{p} \phi_i L^i - \sum_{j=1}^{P} \Phi_j L^{js}\right)(1-L)^d(1-L^s)^D y_t = \left(1 + \sum_{i=1}^{q} \vartheta_i L^i + \sum_{j=1}^{Q} \Theta_j L^{js}\right)\varepsilon_t.$$

We note that the AR side is the *joint* lacunary (sparse) polynomial $A(L) = 1 - \sum_{i=1}^{p} \phi_i L^i - \sum_{j=1}^{P} \Phi_j (L^s)^j$. In the additive form, joint stability is non-factorizable: $A$ does not split into independent nonseasonal and seasonal factors. This nonseparability, together with the gappy support $\{1, \ldots, p\} \cup \{s, 2s, \ldots, Ps\}$, makes the stability region nonconvex and numerically thin: small changes in $(\phi, \Phi)$ can move poles across the unit circle, especially near seasonal frequencies. Therefore, to obtain well-behaved simulations, we adopt a mixture approach: with probability 0.5 we draw an AR-only model ($\Phi_j = 0$ for all $j$) and with probability 0.5 a seasonal-AR-only model ($\phi_i = 0$ for all $i$); each subcase is trivial to stabilize.

These considerations lead to the following practical algorithm for sampling a general SARIMA process. First, sample model orders $p, q, P, Q$ uniformly at random (from 0 to the maximum integer value defined for each of them). For the AR side, flip a fair coin and set either $p = 0$ or $P = 0$ with probability 0.5. Then, once the model order is defined, sample poles uniformly within the unit circle by first sampling the radius uniformly between 0 and $r_{\max}$ and then the complex angle in $[0, 2\pi]$. Next, convert roots to coefficients by expanding the poles product into polynomial form (*e.g.*, see Equation 5)); we also set the integration order to $d = 1$ and $D = 1$, since higher integration orders tend to be numerically unstable. Finally, sample the warmup trajectory state $y_{1:w}$ and innovations process $\varepsilon_{1:w}$ from i.i.d. zero-mean normal distribution, setting $w = \max(p, q, P \cdot s, Q \cdot s, d + D \cdot s)$, and unroll the SARIMA process according to the following time-domain recursion starting at $t = w + 1$:

$$y_t = \sum_{i=1}^{p} \phi_i y_{t-i} + \sum_{j=1}^{P} \Phi_j y_{t-j \cdot s} + \sum_{i=1}^{q} \vartheta_i \varepsilon_{t-i} + \sum_{j=1}^{Q} \Theta_j \varepsilon_{t-j \cdot s} + \varepsilon_t . \tag{6}$$

We apply the non-seasonal and seasonal integrators as a separate pass ($y_t \leftarrow y_t + y_{t-1}$, $y_t \leftarrow y_t + y_{t-s}$). To produce more diverse trend patterns, we further enrich the integration framework by subjecting $y_t$ to a fractional differencing operator, which results in simulating fractional integration orders between 0 and 1 (by sampling fractional difference order $d'$ uniformly at random in $[0, 1]$). We approximate the fractional differencing operator by a finite impulse response filter with binomial coefficients, following the work of Hosking (1981): $\varrho_i = \Gamma(i - d')/(\Gamma(-d')\Gamma(i + 1))$. Simulating trajectories via the recursion (6) is reasonably fast, but the time dependence imposes a sequential loop that cannot be vectorized across *time*. To remove this bottleneck, we vectorize across trajectories: for each sampled parameter tuple $p, q, P, Q, s, d'$, we draw $B$ independent sets of initial conditions (and innovations), and we unroll $B$ paths in parallel. This delivers substantial speedups from vectorized computation and preserves diversity as distinct initial states yield diverse realizations under the same parameters.

## 4.2 SARIMA-2 SIMULATOR

Many real-world series exhibit *bi-seasonality*: a fast rhythm modulated by a slower envelope. Examples include road traffic, web activity, call-center volumes (intra-day peaks modulated by weekday effects) (Moreira-Matias et al., 2013; Strowes, 2016; Ibrahim & L'Ecuyer, 2016); electric load (intraday cycles shaped by weekday and annual patterns) (Hong & Fan, 2016). Single-season models capture the fast cadence but miss amplitude modulation and the induced heteroscedasticity. These observations motivate a compositional construction with two seasonal processes: one high-frequency "base," and one low-frequency "envelope," whose additive or multiplicative interaction reproduces the envelope-on-carrier structure observed in practice. To model this structure, we introduce SARIMA-2, which composes two independent, pole-stable SARIMA processes: a high-frequency *base* series $y^{(b)}$, and a low-frequency *envelope* series $y^{(e)}$. The envelope is upsampled to the base rate and combined with the base either additively or multiplicatively:

$$\text{Additive:} \quad y_t \leftarrow y_t^{(b)} + y_t^{(e)}, \tag{7}$$

$$\text{Multiplicative:} \quad y_t \leftarrow \left(1 + \omega\, \tilde{y}_t^{(e)}\right) y_t^{(b)}, \tag{8}$$

where, in the multiplicative case, $\tilde{y}_t^{(e)}$ is normalized to $[-1, 1]$ and modulation depth, $\omega$, is sampled from $\text{Uniform}(0, 1)$. This composition induces controlled amplitude modulation, capturing effects such as weekday intensity swings. We show an input-output pair for multiplicative SARIMA-2 in Figure 1 (middle).

## 4.3 NOISERS

Many domains exhibit statistical features that cannot be captured by seasonal dynamics (that have a given time scale) alone. Retail and spare-parts demand often shows intermittent spikes and long runs of zeros, motivating Poisson- or count-based models (Croston, 1972; Syntetos & Boylan, 2005). Internet traffic and web workloads are well known to follow heavy-tailed distributions and exhibit burstiness across multiple time scales (Willinger et al., 1995; Feldmann et al., 1998). Environmental processes such as daily precipitation amounts are classically modeled with Gamma-family distributions, highlighting the need for positive, heavy-tailed shocks (Wilks, 1999; Ghil et al., 2002). Together, these examples establish that real-world series frequently combine heteroscedasticity, intermittency, and heavy-tailed disturbances with their seasonal structure. To capture these effects, our simulator incorporates a dedicated *Noiser* module, $\mathbb{N}$, which implements the random rate noise process:

$$\eta_t \sim \mathbb{N}_\kappa(\lambda_t), \qquad \lambda_t = g(y_t) \geq 0,$$

where $y_t$ is the structured signal (output of SARIMA or SARIMA-2), $g(\cdot)$ maps level to a time-varying *rate* (*e.g.*, normalization plus a log-uniform scale), and $\kappa$ denotes the distribution's *shape* hyperparameter(s). The rate $\lambda_t$ ties variability to the local mean level (heteroscedasticity), while $\kappa$ governs the distributional form, typically controlling overdispersion and skew (*e.g.*, for the Gamma distribution the coefficient of variation is $1/\sqrt{\kappa}$ (Johnson et al., 1994)). Thus, $\lambda_t$ sets the scale of fluctuations conditional on $y_t$, and $\kappa$ calibrates how those fluctuations are distributed (spread, asymmetry, and tail behavior). We implement three complementary noiser families: Poisson, Generalized Gamma and Lognormal. All noisers condition their variance on the local series level, normalized by series min and max values and scaled by the rate intensity sampled from LogUniform:

$$\lambda_t = \lambda_0(y_t - \min_t y_t)/(\max_t y_t - \min_t y_t), \quad \lambda_0 \sim \text{LogUniform}[\lambda_{\min}, \lambda_{\max}]. \tag{9}$$

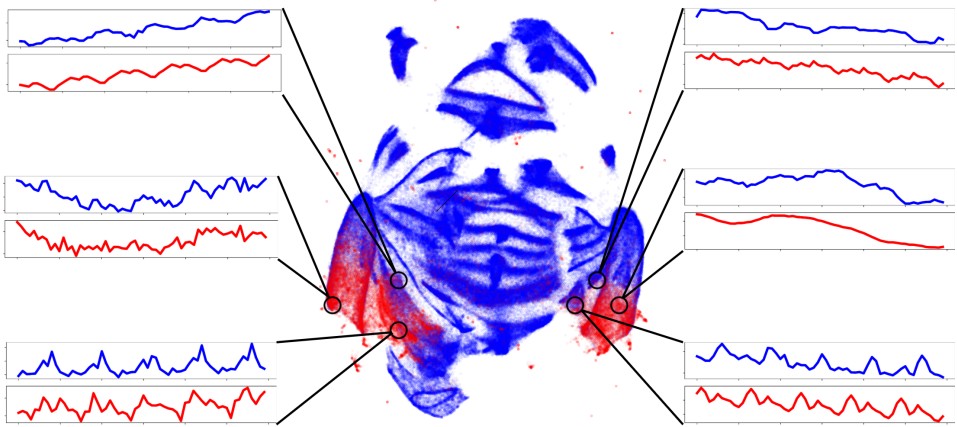

Figure 3: **`SarSim0` Fidelity with Real time series Data.** Center: UMAP embedding space of 60-step windows from real dataset M4-Monthly (red) overlaid on equal-length samples from SARIMA simulator with uniformly sampled seasonality $s \in \{0, \ldots, 24\}$ (blue). Left and right: real (red) and synthetic (blue) series drawn from the circled regions. Key take-aways include: (1) the simulator spans the seasonal regimes seen in real data (*e.g.*, M4-Monthly), including the overall dataset as a special case; and (2) within a seasonality, synthetic series closely match real exemplars co-located in the embedding space, yielding realistic, fine-grained patterns.

*Poisson noiser* generates level-dependent count spikes characteristic of intermittent demand and arrivals, $\eta_t \sim \text{Poisson}(\lambda_t)$. *Generalized Gamma noiser* introduces controllable burstiness by sampling from a gamma distribution, $\eta'_t \sim \text{Gamma}(\lambda_t, \kappa)$, $\kappa \sim \text{LogUniform}[\kappa_{\min}, \kappa_{\max}]$, and further applying a random power transform $\eta_t = [\eta'_t]^\zeta$, $\zeta \sim \text{Uniform}[\zeta_{\min}, \zeta_{\max}]$. *Lognormal noiser* produces multiplicative, heavy-tailed shocks that emulate volatility fluctuations by sampling $\eta'_t \sim \text{LogNormal}(\lambda_t, \kappa)$, $\kappa \sim \text{LogUniform}[\kappa_{\min}, \kappa_{\max}]$. This combination provides a compact, simulator-friendly toolkit to reproduce the main classes of non-Gaussian, level-dependent disturbances observed in practice, while remaining efficient for large-scale, on-the-fly generation. The examples of noise traces produced by the three Noisers are provided in Figure 1 (bottom).

## 5 EMPIRICAL RESULTS

In this section, we evaluate `SarSim0` along five axes. (i) **Fidelity:** Do synthetic series reproduce the statistical structure of real data, providing a credible training signal? (ii) **Generalization:** Does pretraining on simulated data yield models that transfer zero-shot to heterogeneous real datasets, and how does performance compare to models pretrained on real data (*e.g.*, `Chronos`, `Moirai`, `TimesFM`)? (iii) **Architecture robustness:** Can architectures with distinct inductive biases, trained on synthetic diversity at scale, reach performance competitive with recent neural forecasters; and can diversity+scale substitute for architectural complexity? (iv) **Student-beats-teacher effects:** Do models trained only on simulated data outperform their "teacher" (*i.e.*, `AutoARIMA` derived from SARIMA) on real benchmarks? (v) **Ablations:** Do each of the proposed components, SARIMA, SARIMA-2, and the Noisers, contribute measurably to zero-shot generalization? We start by detailing benchmarks, baselines, training and testing protocols, followed by quantitative results and ablation studies addressing these questions.

### 5.1 BENCHMARKS

**GiftEval** is a curated benchmark for evaluating time series foundation models: models are applied to standardized test splits from 23 datasets spanning seven domains and ten sampling frequencies. The benchmark reports results for an array of statistical, deep, and foundation baselines (Aksu et al., 2024). The detailed composition of the benchmark and constituent datasets is provided in Appendix A.

**M-Series** is composed of four forecasting tasks comprising over 100,000 time series, curated from established forecasting competitions: `M1` (Makridakis et al., 1982), `M3` (Makridakis & Hibon, 2000), `M4` (Makridakis et al., 2020), and `Tourism` (Athanasopoulos et al., 2011). These datasets, summarized in Table 4 of Appendix B, represent a broad range of domains and temporal frequencies.

| | GiftEval | | M-SERIES | | Inference | Zero-Shot |
| | sCRPS | MASE | sCRPS | MASE | Time (m) | |
|---|---|---|---|---|---|---|
| Chronos-Base | 0.647 | 0.870 | 0.103 | 0.878 | 2103 | ✓ |
| Chronos-Small | 0.662 | 0.892 | 0.103 | 0.882 | 702 | ✓ |
| MOIRAI-Large | 0.599 | 0.874 | 0.128 | 1.027 | 3976 | ✓ |
| MOIRAI-Small | 0.650 | 0.946 | 0.124 | 1.089 | 994 | ✓ |
| TTM-R2-Pretrained | 0.873 | 1.020 | 0.155 | 1.118 | 43 | ✓ |
| TimesFM | 0.680 | 1.077 | 0.098 | 0.930 | 155 | ✓ |
| NBEATS | 0.815 | 0.937 | 0.095 | 1.134 | 271* | ✗ |
| PatchTST | 0.587 | 0.848 | **0.090** | 1.613 | 165* | ✗ |
| AutoARIMA | 0.912 | 1.074 | 0.096 | **0.843** | 420 | ✗ |
| NBEATS-KernelSynth | 0.686 | 0.978 | 0.116 | 1.033 | - | ✓ |
| NBEATS-ForecastPFN | 1.070 | 1.354 | 0.113 | 0.979 | - | ✓ |
| PatchTST-KernelSynth | 0.702 | 0.977 | 0.119 | 1.051 | - | ✓ |
| PatchTST-ForecastPFN | 1.060 | 1.346 | 0.105 | 0.949 | - | ✓ |
| Chronos-KernelSynth | 0.688 | 0.987 | 0.113 | 1.003 | - | ✓ |
| Chronos-ForecastPFN | 1.283 | 1.575 | 0.118 | 1.001 | - | ✓ |
| NBEATS-SarSim0 | 0.602 | 0.849 | 0.096 | 0.869 | 46 | ✓ |
| PatchTST-SarSim0 | **0.573** | **0.837** | 0.097 | 0.877 | 47 | ✓ |
| Chronos-SarSim0 | 0.608 | 0.878 | 0.100 | 0.896 | 52 | ✓ |
| MLP-SarSim0 | 0.643 | 0.907 | 0.109 | 0.982 | 45 | ✓ |

Table 1: **Performance on benchmarks, weighted aggregation over datasets.** Lower values are better. In the second block, AutoARIMA is fitted on each test series, while NBEATS and PatchTST are trained from scratch on the training split of each dataset (inference time with * includes training time). A green checkmark denotes models evaluated under a strict zero-shot protocol, a yellow checkmark denotes pretrain-test overlap (details in Appendix C), and a red cross denotes non-zero-shot baselines. Key take-aways include: (1) SarSim0 enables strong zero-shot forecasting from synthetic data alone, even outperforming real data pre-training; (2) SarSim0 outperforms alternative more expensive synthetic data pipelines, such as KernelSynth; (3) small models such as NBEATS with SarSim0 close the gap and even slightly exceed large models like Chronos; (4) on GiftEval, architectures pre-trained on SarSim0 outperform AutoARIMA derived from SARIMA generating process.

To ensure the comparability of our work with the existing literature, we evaluate the accuracy of the forecasts using well-established metrics: *scaled Continuous Ranked Probability Score* (sCRPS, (Gneiting & Raftery, 2007)), and *Mean Absolute Scaled Error* (MASE, (Hyndman & Koehler, 2006)), defined in Appendix D. M-Series results report average sCRPS and MASE across datasets weighted by dataset size. GiftEval results report geometric mean of sCRPS and MASE of each method divided by sCRPS and MASE of seasonal naive on each dataset (Aksu et al., 2024). Importantly, in both benchmarks, we use the datasets solely for evaluation purposes—excluding them from model optimization—to assess the zero-shot forecasting capabilities of our models.

## 5.2 BASELINES AND TRAINING METHODOLOGY

We consider a collection of well-established baselines that includes AutoARIMA from StatsForecast (Garza et al., 2022). Neural forecasting baselines trained on the train splits of target datasets are represented by NBEATS (Oreshkin et al., 2020) and PatchTST (Nie et al., 2023). In terms of foundation models, we use publicly available pretrained checkpoints of Chronos (Ansari et al., 2024), TimesFM (Das et al., 2024), MOIRAI (Woo et al., 2024), and TTM (Ekambaram et al., 2024). Our proposed simulator is trained on NBEATS, MLP (a single block of NBEATS), PatchTST and Chronos-Small T5. These models represent very different types of inductive biases (fully-connected design vs. attention-based and patching designs) and all are generally accepted as strong baselines in the literature. Models are implemented in PyTorch (Paszke et al., 2019) and trained for 250k (NBEATS) and 500k (PatchTST, Chronos) steps. Models predict 512 horizons ahead in one shot. Further details and hyperparameter settings along with SarSim0 simulator settings appear in Appendix E. In all cases, *trained models are applied at inference time with no tuning of parameters or hyperparameters on test datasets.* In addition, NBEATS, PatchTST and Chronos are trained on two baseline synthetic data generators, ForecastPFN (Dooley et al., 2023) and KernelSynth (Ansari et al., 2024) to assess SarSim0's effectiveness w.r.t. existing methods. ForecastPFN training uses the exact same settings as SarSim0, whereas for KernelSynth

| | GiftEval | | M-SERIES | |
|---|---|---|---|---|
| | sCRPS | MASE | sCRPS | MASE |
| PatchTST-SarSim0-500K | **0.573** | **0.837** | 0.097 | 0.877 |
| PatchTST-SarSim0-250K | 0.576 | 0.838 | 0.102 | 0.922 |
| No SARIMA-2-250K | 0.647 | 0.926 | 0.103 | 0.929 |
| No Noisers-250K | 0.594 | 0.859 | **0.096** | **0.861** |
| NBEATS-SarSim0 | **0.602** | **0.849** | **0.096** | 0.869 |
| No SARIMA-2 | 0.655 | 0.913 | 0.104 | 0.941 |
| No Noisers | 0.609 | 0.856 | **0.096** | **0.860** |

Table 2: `SarSim0` **ablation**, where lower values are better. Key take-aways include: (1) Removing SARIMA-2 causes the largest accuracy drop across backbones and benchmarks, positioning it as important generalization driver. (2) Removing Noisers has dataset- and model-dependent effects. For PatchTST and limited train budget of 250K iterations, this hurts on GiftEval, but it helps on the more regular M-Series. For extended training budget of 500K iterations, PatchTST with Noisers recovers performance on the M-Series, while retaining stronger results on GiftEval.

we were only able to generate and save to disk up to 10M time series (which is 10 times the scale of synthetic data reported in (Ansari et al., 2024)) of length 1024 due to the very slow generation process (see Figure 4 in Appendix F for a speed comparison of synthetic pipelines).

## 5.3 RESULTS

We begin by exploring the **fidelity** of the proposed SARIMA-based simulator in relation to real time series data. As an illustration, Figure 3 shows a joint UMAP of z-scored 48k windows (60 steps) from M4-Monthly, containing monthly time series rich with (a) trends and (b) monthly seasonalities with period 12 from demographic, finance, industry domains (Makridakis et al., 2020), and 384k equal-length SARIMA samples. Two features are particularly apparent. (i) *Seasonal strata:* Uniformly sampling the seasonal period $s \in \{0, \ldots, 24\}$ yields blue clusters that align with $s$; some align with the red M4 modes, indicating that SARIMA's seasonal knob reproduces and *extends* the dominant structure of the real data. (ii) *Local co-location:* Within each stratum, real and synthetic windows are intermingled; because UMAP preserves local neighborhoods, intermingling suggests a match in short-lag autocovariances and narrowband peaks, not just marginals. Together with the side-panel look-alikes (instance-level fidelity), these observations make SARIMA a plausible data engine for pretraining forecasters capturing relevant seasonal modes, and reproducing local dynamics.

Our key results on **generalization** and **architecture robustness** appear in Table 1, showing aggregate metrics across two benchmarks. Models pretrained solely on our synthetic data achieve strong zero-shot generalization across heterogeneous benchmarks. In particular, `NBEATS-SarSim0`, `PatchTST-SarSim0` and `Chronos-SarSim0` consistently outperform their counterparts trained on prior synthetic approaches (`KernelSynth`, `ForecastPFN`); and they even more than close the gap to large real-data-pretrained models (*i.e.*, `Chronos`, `MOIRAI`, `TimesFM`, `TTM-R2`). The results support the claim that our simulator provides a sufficiently rich training signal to transfer across domains without exposure to target data. Moreover, the relative gains of `NBEATS-SarSim0`, `PatchTST-SarSim0` and `Chronos-SarSim0` highlight *architecture robustness*: architectures with distinct inductive biases reach competitive performance when trained on the same synthetic corpus. Together, these results suggest that diversity and scale of simulated series can partially complement and substitute for architectural complexity. Beyond aggregate scores, we stratify GiftEval results by domain, sampling frequency, term length, and variate type (Appendix H.3). Across all splits, `SarSim0`-pretrained backbones are consistently competitive with, and often stronger than, real-data foundation models. At the same time, `SarSim0` uniformly outperforms prior synthetic generators (KernelSynth, ForecastPFN) and often matches or surpasses strong per-dataset supervised `NBEATS`/`PatchTST` baselines. This indicates that a single, domain-agnostic SARIMA-based curriculum transfers robustly across GiftEval's diverse regimes.

Table 1 also lets us probe a question: can models trained solely on synthetic trajectories derived from SARIMA exceed their statistical "teacher" (`AutoARIMA` derived from SARIMA) on real data? On GiftEval, `NBEATS-SarSim0`, `PatchTST-SarSim0` and `Chronos-SarSim0` achieve markedly *lower* sCRPS and MASE than `AutoARIMA`. On the M-series benchmark, the evidence is mixed. For example, `NBEATS-SarSim0` matches sCRPS, but it is slightly worse in

MASE. Overall, despite being trained on SARIMA-generated data, these neural forecasters exhibit predictive capacity that extends beyond their generative teacher. While Table 1 is based on the full `SarSim0` (including SARIMA-2 and Noisers), which is not exactly expressible as a single `AutoARIMA` specification, Table 22 in Appendix H.4 shows that `NBEATS` trained *only* on pure SARIMA samples leads to the same qualitative conclusion.

Finally, we tackle the **Ablation** axes. Looking at Table 2 we conclude that removing SARIMA-2 (the interaction layer) causes the largest drop in zero-shot accuracy for both `PatchTST` and `NBEATS`, uniformly across benchmarks, suggesting it as a strong driver of generalization. Removing the Noisers has clear model- and benchmark-dependent effects. For PatchTST, for instance, we find that more iterations (500K vs. 250K) are needed to fully benefit from Noisers. This points to a practical trade-off: under tight compute budgets, a simpler simulator configuration and backbone may be preferable, whereas with more generous budgets, the full `SarSim0` pipeline and more expressive backbones yield, in our experiments, stronger overall zero-shot performance. In addition, we performed the `SarSim0` hyperparameter sensitivity study (see Table 13 in Appendix H.2 for details). Across studied configurations, performance on both GiftEval and the M-Series remains very similar to the default configuration, indicating that `SarSim0` is not at a brittle "sweet spot" in design space. The configuration we use (details in Appendix E.3) appears to be benchmark-agnostic and robust to substantial hyperparameter changes.

To probe behavior far outside our target regime of business-style series, we also ran a zero-shot experiment on four canonical nonlinear dynamical systems (Duffing, Lorenz, Lotka-Volterra, Pendulum), comparing several `Chronos` variants, `TimesFM`, and `SarSim0`-pretrained models (see Appendix H.1 for details). `SarSim0`-pretrained models attain error levels comparable to real-data foundation models. This suggests that `SarSim0` 's inductive biases, while geared toward trend/seasonality/intermittency regimes typical of industrial forecasting, yield models whose out-of-domain generalization is similar to that of real-data-pretrained models on these systems.

## 6  DISCUSSION

By introducing `SarSim0`, we have shown that it is possible to achieve competitive zero-shot forecasting through training on simulated time series data alone. By enforcing stability in autoregressive pole sampling, introducing hierarchical seasonality through superposition, and overlaying heavy-tailed noise processes, our `SarSim0` simulator generates trajectories that are fast to generate, privacy- and license-safe, and sufficiently realistic to train models that generalize across real datasets. **Central findings.** First, on GiftEval and the M-Series, simulator-only pretraining achieves competitive zero-shot performance and sometimes exceeds real-data pretraining, suggesting that simulator fidelity, scale, and diversity can matter as much as architectural inductive biases. Second, we also see that our synthetic data pipeline can more than close the gap between small architectures and large pre-trained models such as `Chronos`, again highlighting the power of data generation. Additionally, on GiftEval we observe early, benchmark-specific evidence of a "student-beats-teacher" effect. Global neural models pretrained on `SarSim0`, whose core stochastic assumptions mirror the SARIMA family underlying `AutoARIMA`, can outperform this strong per-series `AutoARIMA` baseline. This suggests an emergent form of generalization beyond the simulator components, motivating further exploration of synthetic-first training for time series foundation models. **Limitations.** There are many exciting directions for future work. For example, Appendix G presents a novelty analysis comparing the SARIMA distributions against M4-monthly using a Local Outlier Factor detector. A consistent and interpretable pattern emerges among the most novel real windows: they typically exhibit (i) abrupt level shifts or structural breaks (*e.g.*, changes in trend slope or regime changes in the underlying process), and (ii) strong, isolated spikes or outages that do not repeat seasonally (especially isolated aperiodic negative spikes). This motivates natural extensions of `SarSim0` based on richer volatility and regime-switching structures (*e.g.*, GARCH-style components or regime-switching SARIMA), which we leave as important future work. While we expect hybrid synthetic-real training to outperform either source alone in many practical settings, we deliberately focus on the clean synthetic-only zero-shot regime, leaving synthetic-real co-training as an important direction for future work. Exogenous covariates are also not considered. Finally, it would be particularly fascinating to understand in detail when and why learners can surpass their generative teachers.

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

## A  GIFTEVAL BENCHMARK

Table 3: Summary of GiftEval forecasting datasets used in our empirical study.

| Dataset | Source | Domain | Frequency | # Series | Avg | Min | Max | # Obs |
|---|---|---|---|---|---|---|---|---|
| Jena Weather | Autoformer (Wu et al. 2021) | Nature | 10T | 1 | 52,704 | 52,704 | 52,704 | 52,704 |
| Jena Weather | Autoformer (Wu et al. 2021) | Nature | H | 1 | 8,784 | 8,784 | 8,784 | 8,784 |
| Jena Weather | Autoformer (Wu et al. 2021) | Nature | D | 1 | 366 | 366 | 366 | 366 |
| BizITObs - Application | AutoMixer (Palaskar et al. 2024) | Web/CloudOps | 10S | 1 | 8,834 | 8,834 | 8,834 | 8,834 |
| BizITObs - Service | AutoMixer (Palaskar et al. 2024) | Web/CloudOps | 10S | 21 | 8,835 | 8,835 | 8,835 | 185,535 |
| BizITObs - L2C | AutoMixer (Palaskar et al. 2024) | Web/CloudOps | 5T | 1 | 31,968 | 31,968 | 31,968 | 31,968 |
| BizITObs - L2C | AutoMixer (Palaskar et al. 2024) | Web/CloudOps | H | 1 | 2,664 | 2,664 | 2,664 | 2,664 |
| Bitbrains - Fast Storage | Grid Workloads Archive (Shen et al. 2015) | Web/CloudOps | 5T | 1,250 | 8,640 | 8,640 | 8,640 | 10,800,000 |
| Bitbrains - Fast Storage | Grid Workloads Archive (Shen et al. 2015) | Web/CloudOps | H | 1,250 | 721 | 721 | 721 | 901,250 |
| Bitbrains - rnd | Grid Workloads Archive (Shen et al. 2015) | Web/CloudOps | 5T | 500 | 8,640 | 8,640 | 8,640 | 4,320,000 |
| Bitbrains - rnd | Grid Workloads Archive (Shen et al. 2015) | Web/CloudOps | H | 500 | 720 | 720 | 720 | 360,000 |
| Restaurant | Recruit Rest. Comp. (Howard et al. 2017) | Sales | D | 807 | 358 | 67 | 478 | 289,303 |
| ETT1 | Informer (Zhou et al. 2020) | Energy | 15T | 1 | 69,680 | 69,680 | 69,680 | 69,680 |
| ETT1 | Informer (Zhou et al. 2020) | Energy | H | 1 | 17,420 | 17,420 | 17,420 | 17,420 |
| ETT1 | Informer (Zhou et al. 2020) | Energy | D | 1 | 725 | 725 | 725 | 725 |
| ETT1 | Informer (Zhou et al. 2020) | Energy | W-THU | 1 | 103 | 103 | 103 | 103 |
| ETT2 | Informer (Zhou et al. 2020) | Energy | 15T | 1 | 69,680 | 69,680 | 69,680 | 69,680 |
| ETT2 | Informer (Zhou et al. 2020) | Energy | H | 1 | 17,420 | 17,420 | 17,420 | 17,420 |
| ETT2 | Informer (Zhou et al. 2020) | Energy | D | 1 | 725 | 725 | 725 | 725 |
| ETT2 | Informer (Zhou et al. 2020) | Energy | W-THU | 1 | 103 | 103 | 103 | 103 |
| Loop Seattle | LibCity (Wang et al. 2023a) | Transport | 5T | 323 | 105,120 | 105,120 | 105,120 | 33,953,760 |
| Loop Seattle | LibCity (Wang et al. 2023a) | Transport | H | 323 | 8,760 | 8,760 | 8,760 | 2,829,480 |
| Loop Seattle | LibCity (Wang et al. 2023a) | Transport | D | 323 | 365 | 365 | 365 | 117,895 |
| SZ-Taxi | LibCity (Wang et al. 2023a) | Transport | 15T | 156 | 2,976 | 2,976 | 2,976 | 464,256 |
| SZ-Taxi | LibCity (Wang et al. 2023a) | Transport | H | 156 | 744 | 744 | 744 | 116,064 |
| M_DENSE | LibCity (Wang et al. 2023a) | Transport | H | 30 | 17,520 | 17,520 | 17,520 | 525,600 |
| M_DENSE | LibCity (Wang et al. 2023a) | Transport | D | 30 | 730 | 730 | 730 | 21,900 |
| Solar | LSTNet (Lai et al. 2017) | Energy | 10T | 137 | 52,560 | 52,560 | 52,560 | 7,200,720 |
| Solar | LSTNet (Lai et al. 2017) | Energy | H | 137 | 8,760 | 8,760 | 8,760 | 1,200,120 |
| Solar | LSTNet (Lai et al. 2017) | Energy | D | 137 | 365 | 365 | 365 | 50,005 |
| Solar | LSTNet (Lai et al. 2017) | Energy | W-FRI | 137 | 52 | 52 | 52 | 7,124 |
| Hierarchical Sales | Mancuso et al. (2020) | Sales | D | 118 | 1,825 | 1,825 | 1,825 | 215,350 |
| Hierarchical Sales | Mancuso et al. (2020) | Sales | W-WED | 118 | 260 | 260 | 260 | 30,680 |
| M4 Yearly | Monash (Godahewa et al. 2021) | Econ/Fin | A-DEC | 22,974 | 37 | 19 | 284 | 845,109 |
| M4 Quarterly | Monash (Godahewa et al. 2021) | Econ/Fin | Q-DEC | 24,000 | 100 | 24 | 874 | 2,406,108 |
| M4 Monthly | Monash (Godahewa et al. 2021) | Econ/Fin | M | 48,000 | 234 | 60 | 2,812 | 11,246,411 |
| M4 Weekly | Monash (Godahewa et al. 2021) | Econ/Fin | W-SUN | 359 | 1,035 | 93 | 2,610 | 371,579 |
| M4 Daily | Monash (Godahewa et al. 2021) | Econ/Fin | D | 4,227 | 2,371 | 107 | 9,933 | 10,023,836 |
| M4 Hourly | Monash (Godahewa et al. 2021) | Econ/Fin | H | 414 | 902 | 748 | 1,008 | 373,372 |
| Hospital | Monash (Godahewa et al. 2021) | Healthcare | M | 767 | 84 | 84 | 84 | 64,428 |
| COVID Deaths | Monash (Godahewa et al. 2021) | Healthcare | D | 266 | 212 | 212 | 212 | 56,392 |
| US Births | Monash (Godahewa et al. 2021) | Healthcare | D | 1 | 7,305 | 7,305 | 7,305 | 7,305 |
| US Births | Monash (Godahewa et al. 2021) | Healthcare | W-TUE | 1 | 1,043 | 1,043 | 1,043 | 1,043 |
| US Births | Monash (Godahewa et al. 2021) | Healthcare | M | 1 | 240 | 240 | 240 | 240 |
| Saugeen | Monash (Godahewa et al. 2021) | Nature | D | 1 | 23,741 | 23,741 | 23,741 | 23,741 |
| Saugeen | Monash (Godahewa et al. 2021) | Nature | W-THU | 1 | 3,391 | 3,391 | 3,391 | 3,391 |
| Saugeen | Monash (Godahewa et al. 2021) | Nature | M | 1 | 780 | 780 | 780 | — |
| Temperature Rain | Monash (Godahewa et al. 2021) | Nature | D | 32,072 | 725 | 725 | 725 | 780 |
| KDD Cup 2018 | Monash (Godahewa et al. 2021) | Nature | H | 270 | 10,898 | 9,504 | 10,920 | 2,942,364 |
| KDD Cup 2018 | Monash (Godahewa et al. 2021) | Nature | D | 270 | 455 | 396 | 455 | 122,791 |
| Car Parts | Monash (Godahewa et al. 2021) | Sales | M | 2,674 | 51 | 51 | 51 | 136,374 |
| Electricity | UCI ML Archive (Trindade 2015) | Energy | 15T | 370 | 140,256 | 140,256 | 140,256 | 51,894,720 |
| Electricity | UCI ML Archive (Trindade 2015) | Energy | H | 370 | 35,064 | 35,064 | 35,064 | 12,973,680 |
| Electricity | UCI ML Archive (Trindade 2015) | Energy | D | 370 | 1,461 | 1,461 | 1,461 | 540,570 |
| Electricity | UCI ML Archive (Trindade 2015) | Energy | W-FRI | 370 | 208 | 208 | 208 | 76,960 |

In this section, we discuss details of the evaluation GiftEval datasets. The benchmark incorporates data from ten sources, beginning with the Jena Weather dataset, from the Autoformer (Wu et al., 2021). Business and IT observability data from BizITObs (processed via AutoMixer's pipeline) includes Application, Service, and L2C metrics that combine business KPIs with IT event data (Palaskar et al., 2024). This is complemented by cloud workload data from Bitbrains via the Grid Workloads Archiv (Howard et al., 2017). For energy-related applications, we incorporate electricity transformer temperature datasets (ETT1 and ETT2) from Informer (Zhou et al., 2020), crucial for long-term power deployment monitoring; and urban transportation time series come from LibCity (Wang et al., 2023). Solar plant energy output predictions are sourced from LSTNet (Lai et al., 2018), and sales forecasting data are from Mancuso et al. (2021). We also include a carefully curated subset of the Monash time series collection, selected to avoid overlap with pretraining data (Godahewa et al., 2021), and electricity consumption patterns of 370 clients from the UCI ML Archive (Trindade, 2015). Table 3 provides detailed statistics for each dataset, including source information, frequency, prediction length, number of variates, series length, and total observations.

# B  M-SERIES BENCHMARK

Table 4: Summary of M-series forecasting datasets used in our empirical study.

|  | Frequency | Seasonality | Horizon | Series | Min Length | Max Length | % Erratic |
|---|---|---|---|---|---|---|---|
| **M1** | Monthly | 12 | 18 | 617 | 48 | 150 | 0 |
|  | Quarterly | 4 | 8 | 203 | 18 | 114 | 0 |
|  | Yearly | 1 | 6 | 181 | 15 | 58 | 0 |
| **M3** | Other | 4 | 8 | 174 | 71 | 104 | 0 |
|  | Monthly | 12 | 18 | 1428 | 66 | 144 | 2 |
|  | Quarterly | 4 | 8 | 756 | 24 | 72 | 1 |
|  | Yearly | 1 | 6 | 645 | 20 | 47 | 10 |
| **M4** | Hourly | 24 | 48 | 414 | 748 | 1008 | 17 |
|  | Daily | 1 | 14 | 4,227 | 107 | 9933 | 2 |
|  | Weekly | 1 | 13 | 359 | 93 | 2610 | 16 |
|  | Monthly | 12 | 18 | 48,000 | 60 | 2812 | 6 |
|  | Quarterly | 4 | 8 | 24,000 | 24 | 874 | 11 |
|  | Yearly | 1 | 6 | 23,000 | 19 | 841 | 18 |
| **Tourism** | Monthly | 12 | 24 | 366 | 91 | 333 | 51 |
|  | Quarterly | 4 | 8 | 427 | 30 | 130 | 39 |
|  | Yearly | 1 | 4 | 518 | 11 | 47 | 23 |

In this section, we discuss details of the M-series data.

## B.1  M1 DATASET DETAILS

The `M1` competition (Makridakis et al., 1982) analyzed 1,001 demographic, industrial, and economic time series of varying frequencies and lengths (9-132 observations). The competition revealed that simple methods like ETS (Holt, 1957) often outperformed complex ones, establishing a legacy emphasizing accuracy, automation, and parsimony in forecasting practice.

## B.2  M3 DATASET DETAILS

The `M3` competition (Makridakis & Hibon, 2000), held 20 years after `M1`, analyzed 3,003 time series from various domains, with 14-126 observations and multiple frequencies. The competition reaffirmed the superiority of simple forecasting methods, with the `Theta` method (Hyndman & Billah, 2003b) emerging as the top performer.

## B.3  M4 DATASET DETAILS

The `M4` competition marked a substantial increase in both size and diversity, comprising 100,000 time series across six frequencies and multiple domains. The competition introduced prediction interval evaluation and featured 18 percent non-smooth series (Syntetos et al., 2005). Notably, a neural network model - ESRNN(Smyl, 2019) - outperformed traditional methods for the first time, helping popularize cross-learning (Semenoglou et al., 2021) in global models.

## B.4  TOURISM DATASET DETAILS

The Tourism dataset (Athanasopoulos et al., 2011) was designed to evaluate forecasting methods applied to tourism demand data across multiple temporal frequencies. It comprises 1,311 time series at monthly, quarterly, and yearly frequencies. This competition introduced the Mean Absolute Scaled Error (MASE) as an alternative metric to evaluate scaled point forecasts, alongside the evaluation of forecast intervals. Notably, 36% of the series were classified as erratic or intermittent. Due to this high proportion of irregular data, the Naïve1 method proved particularly difficult to outperform at the yearly frequency.

## C  DEVIATIONS FROM THE ZERO-SHOT SETTING IN BASELINES

Table 5: Summary of zero-shot deviations from baselines. Checkmark indicates that a given dataset was used to pretrain a given model.

| Dataset | Source | Chronos | TTM-R2 | MOIRAI | TimesFM |
|---|---|---|---|---|---|
| M1 | Monash (Godahewa et al. 2021) | ✗ | ✗ | ✓ | ✗ |
| M3 | Monash (Godahewa et al. 2021) | ✗ | ✗ | ✓ | ✗ |
| M4 | Monash (Godahewa et al. 2021) | ✓ | ✗ | ✓ | ✓ |
| Tourism | Monash (Godahewa et al. 2021) | ✗ | ✗ | ✓ | ✗ |
| Jena Weather | Autoformer (Wu et al. 2021) | ✗ | ✓ | ✓ | ✓ |
| BizITObs | AutoMixer (Palaskar et al. 2024) | ✗ | ✗ | ✗ | ✗ |
| Bitbrains | Grid Workloads Archive (Shen et al. 2015) | ✗ | ✗ | ✗ | ✗ |
| Restaurant | Recruit Rest. Comp. (Howard et al. 2017) | ✗ | ✗ | ✗ | ✗ |
| ETT | Informer (Zhou et al. 2020) | ✗ | ✗ | ✗ | ✗ |
| Loop Seattle | LibCity (Wang et al. 2023a) | ✗ | ✓ | ✓ | ✗ |
| SZ-Taxi | LibCity (Wang et al. 2023a) | ✗ | ✓ | ✓ | ✗ |
| M_Dense | LibCity (Wang et al. 2023a) | ✗ | ✗ | ✗ | ✗ |
| Solar | LSTNet (Lai et al. 2017) | ✓ | ✓ | ✓ | ✗ |
| Hierarchical Sales | Mancuso et al. (2020) | ✗ | ✓ | ✓ | ✗ |
| Hospital | Monash (Godahewa et al. 2021) | ✗ | ✗ | ✓ | ✗ |
| COVID Deaths | Monash (Godahewa et al. 2021) | ✗ | ✓ | ✓ | ✗ |
| US Births | Monash (Godahewa et al. 2021) | ✗ | ✓ | ✓ | ✗ |
| Saugeen | Monash (Godahewa et al. 2021) | ✗ | ✓ | ✓ | ✗ |
| Temperature Rain | Monash (Godahewa et al. 2021) | ✗ | ✓ | ✓ | ✗ |
| KDD Cup 2018 | Monash (Godahewa et al. 2021) | ✓ | ✓ | ✓ | ✗ |
| Car Parts | Monash (Godahewa et al. 2021) | ✗ | ✗ | ✓ | ✗ |
| Electricity | UCI ML Archive (Trindade 2015) | ✓ | ✗ | ✗ | ✓ |

In this section, we examine in detail how the baselines reported in Table 1 deviate from a strict zero-shot forecasting regime, as defined in Section 4. While several recent models have been described as "foundation forecasting models" many rely directly on the evaluation datasets as training data. As a result, their performance can not be interpreted as that of a foundation forecasting model.

- **Chronos** (Ansari et al., 2024) paper explicitly distinguishes between pre-train only, in-domain, and zero-shot evaluation settings. Unfortunatedly, several of the datasets they include in their in-domain evaluation regime (Table 3 of their paper) overlap with the GiftEval and the M-series benchmark collections. The datasets include M4, Solar, KDD cup and Electricity.

- **TTM-R2** (Ekambaram et al., 2024) model was introduced in the context of long-horizon forecasting, and not framed as a zero-shot model. More recently, TTM-R2 and its fine-tuned variants have been explicitly adapted to GiftEval through task-specific fine-tuning, which further departs from the zero-shot foundation model definition.[1] The deviations from the zero-shot regime include the long-horizon Weather dataset, Loop Seattle, Taxi, Solar, Hierarchical Sales, and a big subset of the Monash repository in Hospital, COVID deaths, US births, Saugeen, temperature rain and the KDD cup 2018.

- **MOIRAI** (Woo et al., 2024) was introduced alongside the LOTSA dataset collection, which aggregates the entire Monash repository. This includes the training portions of M1, M3, M4, and Tourism datasets. While the authors report zero-shot results on long-horizon collections such as ETT, Electricity, and Weather, these results are confounded by the fact that Traffic (also part of their long-horizon evaluation) is included in LOTSA pre-training. The MOIRAI method is the most affected by deviations from zero-shot, as they pretrain using 76% of the evaluation datasets.

- **TimesFM** (Das et al., 2024) pre-trains on the full M4 dataset, as well as Electricity, Traffic, and Weather from the long-horizon benchmark suite. Consequently, its reported results on these datasets should not be interpreted as zero-shot. At the time of this publication, the TimesFM model is the least affected by deviations from the zero-shot regime, using only M4 and Weather.

---

[1]While not yet published, the TTM-R2 pre-train datasets are mentioned here.

# D  METRIC DEFINITIONS

In this section, we describe several metrics used in our evaluation.

To ensure the comparability of our work with the existing literature, we evaluate the accuracy of the forecasts using *scaled Continuous Ranked Probability Score* (sCRPS, (Gneiting & Raftery, 2007)), defined as:

$$\text{sCRPS}\left(\mathbf{y},\ \hat{\mathbf{Y}}\right) = \frac{\sum_{i,t,h} \text{CRPS}(y_{i,t,h}, \hat{Y}_{i,t,h})}{\sum_{i,t,h} |y_{i,t,h}|}. \tag{10}$$

That is, the sCRPS is a scaled version of the CRPS in which we divide by the sum of the absolute values of the series. We use a Riemann integral approximation technique that uniformly averages the quantile loss over a discrete set of quantiles:

$$\text{CRPS}(y, \hat{Y}) = 2 \int \rho_\tau (y - F_{\hat{Y}}^{-1}(\tau)) d\tau. \tag{11}$$

We complement the evaluation of the probabilistic forecasts through the *Mean Absolute Scaled Error* (MASE, (Hyndman & Koehler, 2006)), that considers the ratio between mean absolute error of forecasts over mean absolute error of the `SeasonalNaive` forecast $\tilde{y}_{i,t,h}$ (*i.e.*, a point forecast using the last observation on the previous season), defined as:

$$\text{MASE}\left(\mathbf{y},\ \hat{\mathbf{y}}\right) = \frac{\sum_{i,t,h} |y_{i,t,h} - \hat{y}_{i,t,h}|}{\sum_{i,t,h} |y_{i,t,h} - \tilde{y}_{i,t,h}|}. \tag{12}$$

# E    TRAINING METHODOLOGY AND HYPERPARAMETER SETTINGS

In this section, we describe in more detail our training methodology and hyperparameter settings.

## E.1    TRAINING METHODOLOGY

We train the following zero-shot models on our proposed simulator. We consider `NBEATS`, MLP (basically a single block of `NBEATS`), and `PatchTST`. These models represent very different classes of models based on distinct inductive biases (fully-connected design vs. attention-based patching design). Moreover, these are time-tested and generally accepted as strong baselines in forecasting literature. Models are implemented in PyTorch Paszke et al. (2019) using pytorch lightning framework Falcon & The PyTorch Lightning team (2019) trained for 250K (`NBEATS`) and 500K (PatchTST) steps using the Adam optimizer Kingma & Ba (2015) with default settings and learning rate 0.0001. Batch size is fixed at 4096 and the model takes the history of 4096 points (the maximum history length reported in the literature) and predicts 512 horizons ahead in one shot. `NBEATS` is trained on 8xV100 machine, and training takes about 16 hours. `PatchTST` is trained on 8xA100 machine, and training takes about 48 hours.

*Trained models are applied at inference time with no tuning of parameters or hyperparameters on test datasets.* Hyperparameter settings of the neural networks are provided in Tables 6 and 7. Hyperparameter settings of the proposed `SarSim0` simulator are provided in Table 9.

To train the networks, we take the simulator generated sequence of length 6000 samples and randomly choose start and end points of the training sequence of total length 4096 (history) plus 512 (forecast length). We then supply history sequence to the neural network and compute the multi-horizon multi-quantile loss (1) against the forecast ground truth, learning with backpropagation. In order to train the networks to deal with variable sequence length, the 4096-length window is zeroed out on the left with padding mask to simulated the effects of padding (the length of zeroed segment is sampled uniformly from Uniform$[0, 4088]$). At inference time, short sequences are left padded to length 4096. The output of the network is cut to match the required prediction length (specified by the target dataset). For datasets in GiftEval requiring output lengths longer than 512, we downsample the input by a factor of 2 and upsampled the forecast by the same factor (resulting in output length 1024, which is then trimmed to required forecast length). Downsampling and upsampling are implemented using Kaiser window with filter length 3, rolloff 0.9 from torchaudio Resample function (Yang et al., 2021).

## E.2    HYPERPARAMETER SETTINGS

Hyperparameter settings for various models are given in the following tables.

Table 6: `NBEATS` Hyperparameter Settings

| Hyperparameter | Value |
| --- | --- |
| Batch Size | 4096 |
| Learning rate | 0.0001 |
| Training steps | 250,000 |
| Input length | 4096 |
| Ouput length | 512 |
| Main Activation Function | ReLU |
| Number of Blocks | 10 |
| Layers within block | 3 |
| Block hidden size | 1024 |

Table 7: `PatchTST` Hyperparameter Settings

| Hyperparameter | Value |
|---|---|
| Batch Size | 4096 |
| Learning rate | 0.0001 |
| Training steps | 500,000 |
| Input Length | 4096 |
| Output Length | 512 |
| Main Activation Function | ReLU |
| Patch Length | 64 |
| Patch Stride | 32 |
| Number of Layers | 12 |
| Number of Attention Heads | 8 |
| d_model | 512 |
| FFN hidden_size | 2048 |
| Apply Revin | True |
| Residualized Attention | True |

Table 8: `Chronos`-Small Hyperparameter Settings

| Hyperparameter | Value |
|---|---|
| Batch Size | 1024 |
| Learning rate | 0.0001 |
| Training steps | 500,000 |
| Input Length | 512 |
| Output Length | 512 |
| Main Activation Function | ReLU |
| Patch Length | 1 |
| Patch Stride | 1 |
| Number of Encoder Layers | 6 |
| Number of Decoder Layers | 6 |
| Number of Attention Heads | 8 |
| use_reg_token | True |
| d_model | 512 |
| d_ff | 2048 |
| d_kv | 64 |
| Dropout | 0 |

Table 9: `SarSim0` Hyperparameter Settings

| Hyperparameter | Value |
|---|---|
| History Sequence Length | 4096 |
| Padding Length | Uniform$[0, 4088]$ |
| Forecasting Horizon | 512 |
| SARIMA Sampling Configuration | |
| Base Sequence Length | 6000 |
| Vectorization Batch Size | 256 |
| $p$, (AR order) | Uniform$[0, 10]$ |
| $q$, (MA order) | Uniform$[0, 3]$ |
| $P$, (Seasonal AR order) | Uniform$[0, 2]$ |
| $Q$, (Seasonal MA order) | Uniform$[0, 2]$ |
| $s$, (Seasonality period) | Uniform$[0, 52]$ |
| $r_{\max}$ (AR max pole radius) | 0.9 |
| AR pole angle | Uniform$[0, 2\pi]$ |
| $R_{\max}$ (Seasonal AR max pole radius) | 0.1 |
| Seasonal AR pole angle | Uniform$[0, 2\pi]$ |
| $d$ (Non-seasonal integration order) | Uniform$[0, 1]$ |
| $D$ (Seasonal integration order) | 1 |
| SARIMA-2 Sampling Configuration | |
| Seasonality pairs | Uniform$[[24, 7], [7, 52], [0, 7], [0, 3], [0, 24], [0, 52]]$ |
| $\varrho$ (Modulation depth) | Uniform$[0, 1]$ |
| Additive vs. Multiplicative mixing selection probability | 0.5 |
| Noisers Sampling Configuration | |
| Input selection probability (SARIMA vs. SARIMA-2) | 0.5 |
| Noiser selection | Uniform$[$Poisson, Gamma, Lognormal, Passthrough$]$ |
| $\lambda_0^{\mathrm{P}}$ (Base rate for Poisson) | LogUniform$[0.1, 100]$ |
| $\lambda_0^{\mathrm{G}}$ (Base rate for Generalized Gamma) | LogUniform$[0.1, 100]$ |
| $\lambda_0^{\mathrm{L}}$ (Base rate for Lognormal) | LogUniform$[0.1, 5]$ |
| $\kappa^{\mathrm{P}}$ (Shape for Poisson) | N/A |
| $\kappa^{\mathrm{G}}$ (Shape for Generalized Gamma) | LogUniform$[1, 50.0]$ |
| $\kappa^{\mathrm{L}}$ (Shape for Lognormal) | LogUniform$[1, 3.0]$ |
| $\zeta$ (Power for Generalized Gamma) | Uniform$[0.5, 1.5]$ |

### E.3 SARSIM0 HYPERPARAMTER VALUES AND SELECTION METHODOLOGY

The settings of `SarSim0` simulator parameters appear in Table 9, and the detailed explanation for each setting is provided below.

Vectorization Batch Size is set to 256 for on-the-fly generation efficiency.

Model orders p, q, P, Q are set to values often encountered in practice, *e.g.*, 0, 1, 2, 3, with the exception of p, maximum AR order, set to 10, mostly to demonstrate the efficiency of simulator even with higher-order models that are typically more compute hungry. Smaller settings of this parameter yield similar results.

The seasonality period parameter, $s$, was chosen to cover the typical seasonalities encountered in the data: yearly data typically have seasonality 0 (no seasonality), monthly data often exhibit seasonality 12, quarterly - 4, hourly - 24, weekly - 52, daily - 7. Overall, in order to avoid any benchmark specific biases, we decided to sample seasonalities in SARIMA uniformly in the range covering most common seasonalities: [0, 52].

Similarly, SARIMA-2 was added using the prior knowledge that many practical time series have double seasonalities, with specific motivating examples from the literature laid out at the beginning of Section 4.2. Hourly data with double seasonality often have 24 x 7 pattern (within day and within

week daily patterns); daily data often have 7 x 52 pattern (within week and within year weekly patterns). Additionally, mixing the seasonality 0 process with the typical natural frequencies listed above, in SARIMA-2, enriches the data with the traces that imitate stochastic-volatility patterns through non-linear cross-modulation. These motivate the choice of SARIMA-2 seasonality pairs, uniformly sampled from this set: [24, 7], [7, 52], [0, 7], [0, 4], [0, 24], [0, 52].

Next, AR max pole radius is chosen to be 0.9 to avoid any numerical issues due to near-unit-root instability that may arise in AR process due to floating quantization error accumulation effects. Similarly, Seasonal AR max pole radius is set to 0.1 for the same reason. Recall that with seasonality $s = 24$ the seasonality pole will be $0.1^{1/24} = 0.89$, for example. AR pole angles are sampled uniformly in $[0, 2\pi]$ - *non-informative prior*.

Non-seasonal integration order is sampled uniformly in $[0, 1]$ - *non-informative prior*.

Seasonal integration order is set to 1. We have tried higher integration orders for both, but they didn't work well because of numerical stability issues in long sequences, so we limit the maximum integration order to 1.

SARIMA-2 Modulation depth is sampled uniformly in $[0, 1]$ - *non-informative prior*.

Additive vs. Multiplicative mixing selection probability is set to 0.5 - *non-informative prior*.

Input selection probability (SARIMA vs. SARIMA-2) is set to 0.5 - *non-informative prior*.

Noiser selection is motivated by the review of relevant literature at the beginning of Section 4.3, documenting the heteroscedasticity, intermittency, and heavy-tailed disturbance effects in real data from multiple unrelated application domains. Poisson, LogNormal and Generalized Gamma processes seem to be the dominating and heavy-tail distribution families that cover a variety of real data. This is why those were chosen, along with a pass-through (no noise) noiser instance. This list is, of course, non-exaustive, and we believe multiple other noise models are viable and should be explored in future work. The sampling of noisers follows a *non-informative uniform prior*.

Base rate for Poisson is sampled from LogUniform[0.1, 100] for the following reason. First, for large values of rate $\lambda > 100$ Poisson is largely indistinguishable from Gaussian. Second, for very low rate values the Poisson noise produces mostly zeros; for example, with $\lambda = 0.1$, only roughly 1 in 10 samples is non-zero and we decided not to go beyond that. LogUniform distribution is used for rate sampling to concentrate more significant Poisson Noiser mass in a significantly non-Gaussian region of the rates.

Base rate for Generalized Gamma is sampled from LogUniform[0.1, 100], inheriting the range from Poisson.

Base rate for Lognormal is sampled from LogUniform[0.1, 5], where the upper bound is adjusted to 5 to avoid numerical issues. The mean of Log-Normal process is $\exp(\lambda + \sigma^2/2)$, which explodes quickly if either $\lambda$ or $\sigma$ are larger than 5-10. The shape of Generalized Gamma is sampled from LogUniform[1, 50.0]. The upper bound is set to 50 as the distribution converges to Gaussian for large shape. Alternatively shape below 1, Gamma density blows up near 0, producing a lot of very small positive values, clustered very close to 0, and a long right tail (cf. the Poisson low-rate zero-dominated case). Shape for Lognormal is sampled from LogUniform[1, 3.0], where we adjust the upper bound to 3, so that mean of the process is limited by a reasonable number, $\exp(5 + 3^2/2) \approx 13360$, also effects of rate 5 and variance $3^2/2 = 4.5$ are roughly equal; whereas the lower bound corresponds to the LogNormal with unit variance.

Finally, Power for Generalized Gamma is sampled from Uniform[0.5, 1.5] so that $\zeta = 1$ (standard Gamma) sits in the middle, while the interval explores a moderate range of heavier-tailed ($\zeta < 1$) and lighter-tailed ($\zeta > 1$) shapes. For smaller values of $\zeta$, the density again blows up near zero; whereas for larger values we have observed some numerical issues arising from taking a significant power of a large number.

## F  FORECASTPFN AND KERNELSYNTH SYNTHETIC GENERATORS

In this section, we provide detailed implementation and analysis of two synthetic time series generators used in recent foundation models: ForecastPFN's ETS-based generator (Dooley et al., 2023) and Chronos's KernelSynth GP-based generator (Ansari et al., 2024). Both methods generate diverse training data, but they employ fundamentally different mechanisms—structured multiplicative decomposition versus compositional Gaussian processes. We present their mathematical formulations, our optimized implementations, and empirical comparisons of computational efficiency and scalability.

**ForecastPFN ETS Generator.**  ForecastPFN Dooley et al. (2023) generates synthetic training data using a structured prior based on a multiplicative Error-Trend-Seasonality (ETS) decomposition. Each series is modeled as the product of a trend, a multi-scale seasonal component, and a Weibull-distributed noise factor, with all underlying parameters drawn from prior distributions to create the synthetic training corpus. The formal composition is given by the following: The component periods are set to standard values (*e.g.*, $p_{\text{week}} = 7, p_{\text{month}} = 30.5$). Parameters for trend are drawn from Gaussian priors, while Fourier coefficients for seasonality are scaled as $c_{f,\nu}, d_{f,\nu} \sim \mathcal{N}(0, 1/f)$ and normalized to a unit sum-of-squares. Crucially, the model trains on the noise-free targets $\psi(t)$ rather than the full series $y_t$, accelerating convergence while maintaining generalization to noisy real-world data. Here are the details:

$$y_t = \psi(t)\, z_t \;=\; \text{trend}(t)\, \text{seasonal}(t)\, z_t,$$

$$z_t = 1 + m_{\text{noise}}\,(z - \bar{z}), \quad z \sim \text{Weibull}(1, k), \quad \bar{z} = (\ln 2)^{1/k},$$

$$\text{trend}(t) = \left(1 + m_{\text{lin}}\, t + c_{\text{lin}}\right)\left(m_{\text{exp}}\, c_{\text{exp}}^t\right),$$

$$\text{seasonal}(t) = \prod_{\nu \in \{\text{week,month,year}\}} \text{seasonal}_\nu(t),$$

$$\text{seasonal}_\nu(t) = 1 + m_\nu \sum_{f=1}^{\lfloor p_\nu/2 \rfloor} \left[ c_{f,\nu} \sin\!\left(\tfrac{2\pi f t}{p_\nu}\right) + d_{f,\nu} \cos\!\left(\tfrac{2\pi f t}{p_\nu}\right) \right],$$

$$\text{with } p_{\text{week}} = 7,\ p_{\text{month}} = 30.5,\ p_{\text{year}} = 365.25,\ \sum_f (c_{f,\nu}^2 + d_{f,\nu}^2) = 1.$$

We compared the official ForecastPFN implementation against our optimized vectorized implementation using 50,000 synthetic time series samples to ensure mathematical equivalence. All statistical tests (Kolmogorov-Smirnov, Mann-Whitney U) show no significant differences (p > 0.05), with negligible effect sizes (Cohen's d < 0.01), confirming statistical equivalence between implementations. Our optimized vectorized implementation achieves $14.6\times$ average speedup with $86.8\%$ time reduction while maintaining statistical equivalence to the original implementation.

Table 10: Statistical Comparison: Official vs. Our Vectorized Implementation (n=50,000)

| Metric | Official $\mu \pm \sigma$ | Our Impl. $\mu \pm \sigma$ | KS p-value | MW p-value | Cohen's d |
|---|---|---|---|---|---|
| Range | $20.56 \pm 36.66$ | $20.91 \pm 37.97$ | 0.648 | 0.888 | 0.009 |
| Mean | $0.85 \pm 11.14$ | $0.94 \pm 11.37$ | 0.579 | 0.243 | 0.008 |
| Std Deviation | $5.22 \pm 9.08$ | $5.29 \pm 9.35$ | 0.894 | 0.871 | 0.008 |
| Trends | $0.006 \pm 0.222$ | $0.008 \pm 0.227$ | 0.458 | 0.206 | 0.009 |
| Seasonality | $13.13 \pm 3.55$ | $13.14 \pm 3.55$ | 0.803 | 0.724 | 0.004 |
| Noise Level | $2.33 \pm 4.30$ | $2.36 \pm 4.46$ | 0.336 | 0.799 | 0.006 |

**KernelSynth.**  KernelSynth is a GP-based synthetic time series generator that *inverts* the Automatic Statistician: rather than discovering kernel compositions to explain data, it randomly composes kernels to *produce* diverse temporal patterns (Ansari et al., 2024; Duvenaud et al., 2013). The method maintains a kernel bank $\mathcal{K}$ capturing fundamental patterns—Linear (trends), Periodic (seasonality at daily/weekly/monthly/yearly scales), RBF (smooth local variations), Rational Quadratic (multi-scale structures), Constant (bias), and White Noise (stochastic components). Generation proceeds by sampling $j \sim \mathcal{U}\{1, J\}$ kernels with replacement and hierarchically combining them via random binary

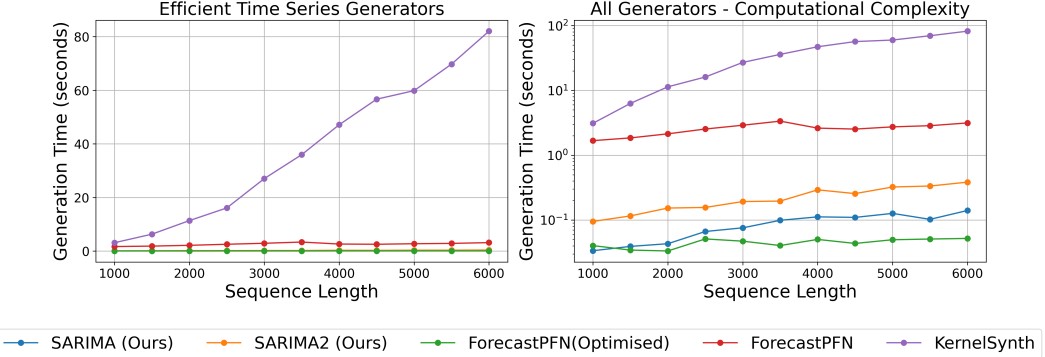

Figure 4: The time per time-series comparison of different simulators.

operations (+ or ×) into a composite covariance $\tilde{\kappa}(t, t')$, where addition superposes independent patterns and multiplication creates localized interactions (*e.g.*, RBF×Periodic yields locally periodic patterns that decay). Finally, a length-$\ell_{\text{syn}}$ time series is drawn from the GP prior $\mathcal{GP}(0, \tilde{\kappa})$. We now detail the mathematical formulation of these three core components. The generation pipeline consists of the following elements:

- **Kernel bank** $\mathcal{K}$**:** Constant $k_{\text{C}} = \sigma_c^2$; White $k_{\text{W}} = \sigma_n^2 \delta_{t,t'}$; Linear $k_{\text{L}} \propto tt'$; RBF $k_{\text{RBF}} \propto \exp(-\Delta t^2/2\ell^2)$; Rational-Quadratic $k_{\text{RQ}} \propto (1 + \Delta t^2/2\alpha\ell^2)^{-\alpha}$; Periodic $k_{\text{Per}} \propto \exp(-2\sin^2(\pi\Delta t/p)/\ell^2)$ with periods $p \in \{4, 6, 12, 24, 26, 30, 48, 52, 60, 96, 365, 730\}/\texttt{LENGTH}$.

- **Compositional grammar:** Sample $j \sim \mathcal{U}\{1, J\}$ kernels $\kappa_1, \ldots, \kappa_j$ from $\mathcal{K}$; fold recursively via random $\star \in \{+, \times\}$ to get $\tilde{\kappa} = (((\kappa_1 \star \kappa_2) \star \kappa_3) \cdots)$. Addition superposes independent effects; multiplication localizes/gates patterns.

- **GP sampling:** Grid $t_{1:n}$ ($n = 1024$), build covariance $K_{ij} = \tilde{\kappa}(t_i, t_j)$, draw $\mathbf{x} \sim \mathcal{N}(\mathbf{0}, K)$.

Table 11: Comparison of synthetic time series generation methods

| Method | Scales with Length | On-the-fly Synthesis* | Generalization |
|---|---|---|---|
| ForecastPFN | ✓ | ✗ | ✗ |
| KernelSynth | ✗ | ✗ | ✗ |
| SarSim0 (ours) | ✓ | ✓ | ✓ |

The time per time-series comparison of different simulators is presented in Figure 4. We note that the KernelSynth generation speed is very slow compared to other alternatives. For ForecastPFN we were able to achieve comparable speed via vectorization of the original code.

# G  UMAP Visualization Analysis

In this section, we provide more details on our visualization analysis.

The UMAP visualization in Figure 3 of the main paper demonstrates a significant and impressive overlap between the synthetic data generated by the SARIMA simulator and the real-world M4-monthly time series. Here, we describe a principled and automated methodology to move beyond visual inspection, systematically identifying and characterizing the real-world patterns that the samples SARIMA simulator do not cover.

## G.1  Methodology

To identify these missing patterns, we employ a two-stage pipeline that first learns a low-dimensional representation of the time series, *e.g.*, through UMAP, and then uses a density-based algorithm to score and rank real series based on their novelty with respect to the synthetic distribution.

**Dimensionality Reduction with UMAP and DensMAP.**  To maintain consistency with the main paper, our initial analysis uses Uniform Manifold Approximation and Projection (UMAP). As a complementary approach for a more robust analysis, we also use its density-preserving variant, DensMAP (Narayan et al., 2020). While UMAP is powerful for visualizing cluster structures, its assumption of uniform point distribution on the manifold can distort local density information. DensMAP refines this by adding a density-preservation term to its objective function, producing an embedding that more faithfully represents the local densities of the original data space, which is critical for the subsequent novelty detection step. Our DensMAP results are not presented here to manage the document size, but results were qualitatively similar.

**Novelty Detection with Local Outlier Factor (LOF).**  We use the LOF algorithm (Breunig et al., 2000) for density-based novelty detection. For each embedding method (UMAP and DensMAP), an LOF model is trained exclusively on the embeddings of the synthetic series. This model learns the density patterns of the "normal" data as defined by the SARIMA sampler. Each real series is then scored by this model, where a more negative score indicates that the series lies in a low-density region of the synthetic data manifold and is thus considered novel. The high-level approach of combining dimensionality reduction with outlier detection is a well-established technique for identifying anomalous time series. While our method learns embeddings directly from raw time series, Hyndman et al. (2015) have successfully used feature-based approaches. Furthermore, Vaidya & Vaidya (2022) provides empirical evidence that dimensionality reduction can preserve or even enhance the performance of outlier detection algorithms, supporting the validity of our pipeline.

We use default values for the hyperparameters of the LOF model. Series with lowest 5% novelty score are considered as "novel."

## G.2  Analysis of M4-monthly Series

**Novelty Detection with LOF.**  We apply our novelty detection pipeline to the M4-monthly dataset across four experimental settings, varying both the dimensionality reduction technique (UMAP vs. DensMAP) and the synthetic data source (basic SARIMA vs. the full `SarSim0` simulator). We supplement the analysis on the samples of the full `SarSim0` because they are the actual samples feed into the model during training.

In total there are four sets of experiments, each of which are demonstrated by two figures:

1. Identifying Novel M4-monthly Series with UMAP/DensMAP and LOF. The left panel shows the 2D embedding space, visualizing the coverage of the M4-monthly dataset by synthetic samples. Real-world series are categorized as covered by the synthetic distribution (green), novel (orange), or among the top-200 most novel (red stars). The right panel displays the same M4-monthly embeddings, colored by their LOF novelty score, highlighting that the most novel series (red) form distinct clusters at the edges of the synthetic data distribution.

2. Visualization of the Top 100 Most Novel M4-monthly Series. This figure visualizes the time series corresponding to the 100 lowest LOF novelty scores from the UMAP/DensMAP analysis.

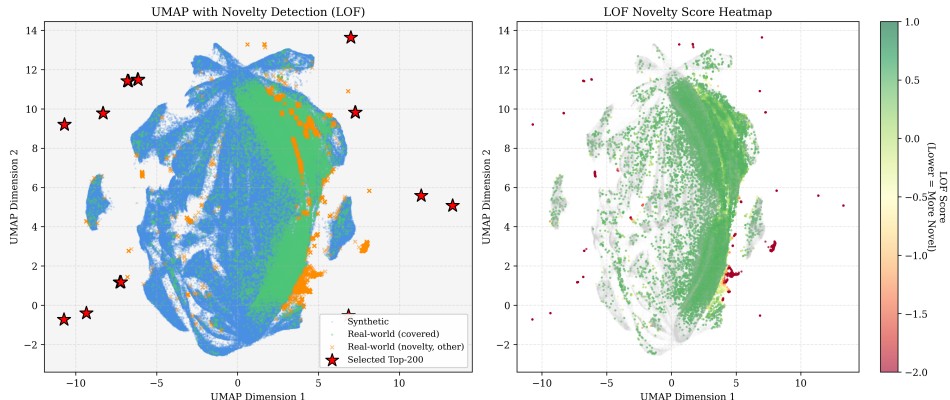

Figure 5: **Baseline SARIMA + M4-monthly + UMAP.** LOF novelty detection results.

**SARIMA + UMAP.**   See Figure 5 and Figure 6.

**SarSim0 + UMAP.**   See Figure 7 and Figure 8.

Our findings can be summarized in two key points:

First, the combination of dimensionality reduction and the LOF algorithm is an effective pipeline for systematically identifying simulator coverage gaps. Across both UMAP and DensMAP embeddings, the LOF novelty scores align with visual inspection: the series that form distinct, uncovered clusters in the embedding space are precisely those that receive the highest novelty scores. This confirms that the "holes" are not mere visualization artifacts, but instead represent statistically significant deviations from the synthetic data distribution.

Second, a consistent and interpretable pattern emerged from the top-ranked novel series across all four experimental settings. These typically correspond to time series windows with: (i) abrupt level shifts or structural breaks (*i.e.*, trend slope change or more generally generating process change); (ii) strong, isolated spikes or outages that do not repeat seasonally (especially isolated aperiodic negative spikes); or (iii) patterns with pronounced quasi-periodic calendar effects that are not captured by purely seasonal-period–based SARIMA specification.

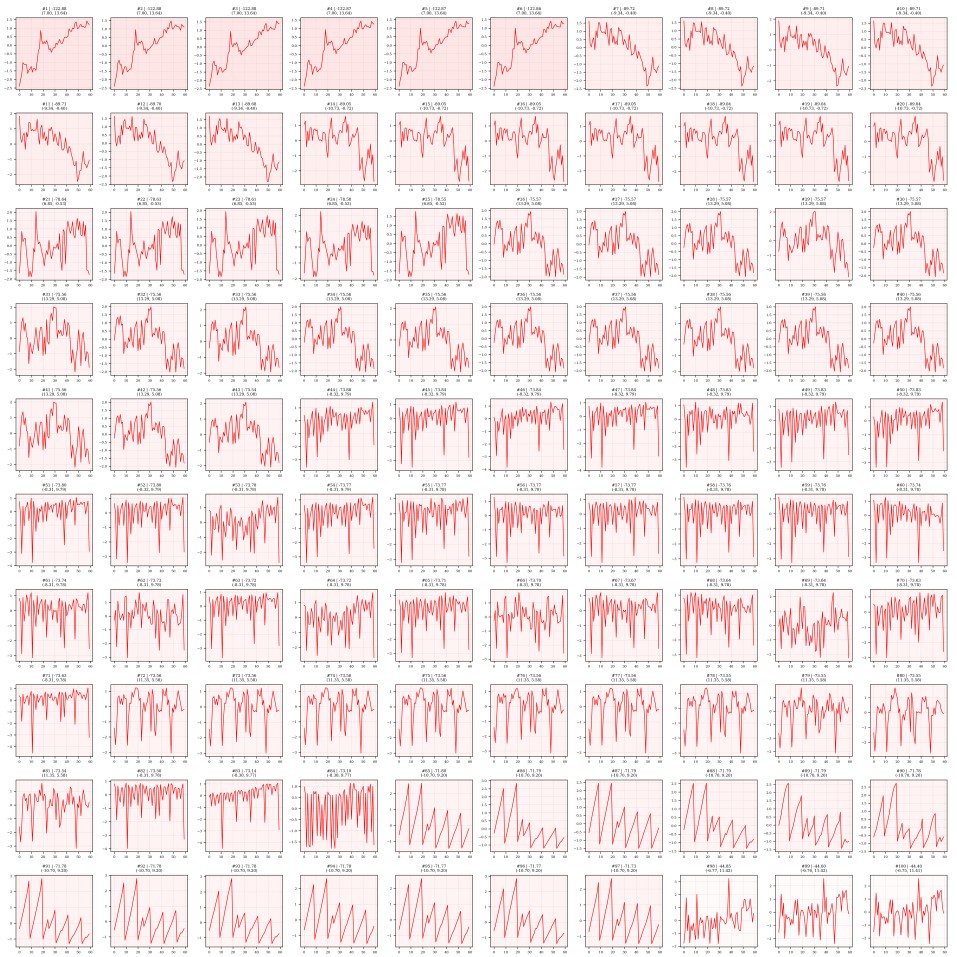

Figure 6: **Baseline SARIMA + M4-monthly + UMAP.** Top 100 novel M4-monthly series visualization

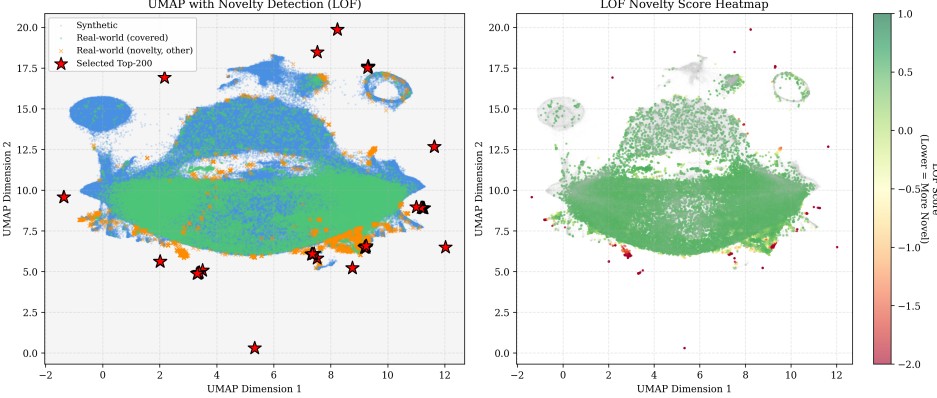

Figure 7: `SarSim0` **+ M4-monthly + UMAP.** LOF novelty detection results.

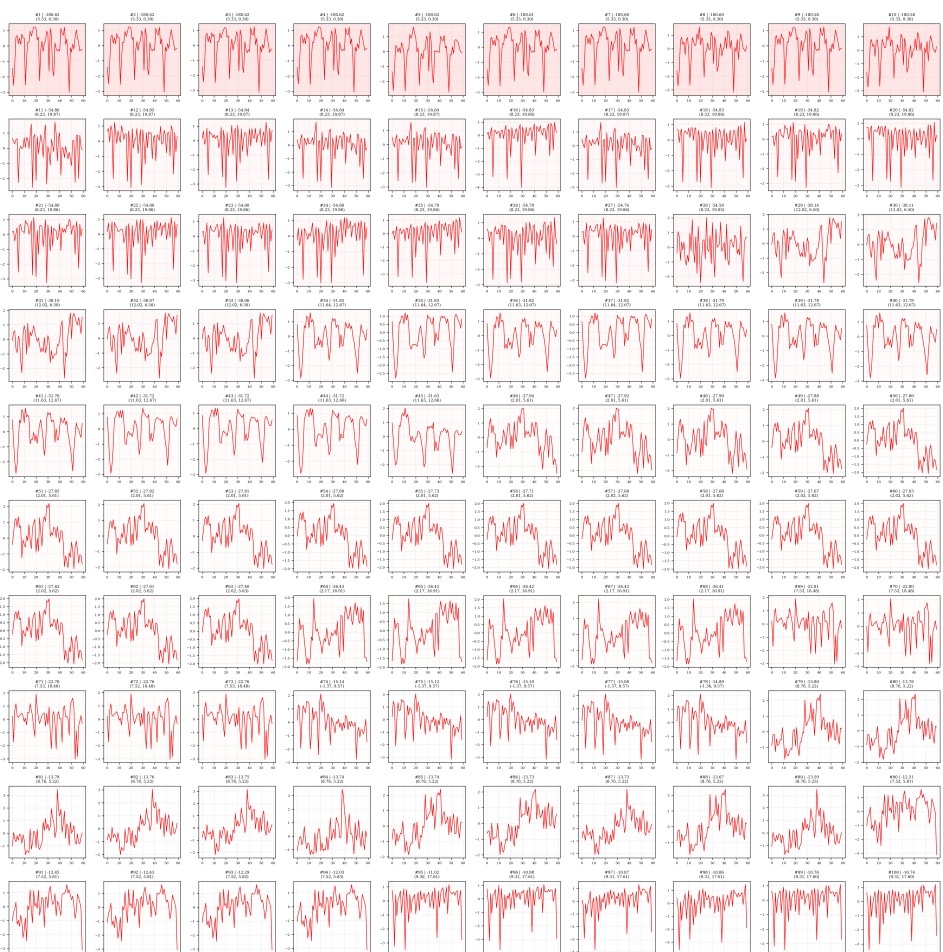

Figure 8: **SarSim0 + M4-monthly + UMAP.** Top 100 novel M4-monthly series visualization

| | Duffing | | Lorenz | | Lotka-Volterra | | Pendulum | |
| --- | --- | --- | --- | --- | --- | --- | --- | --- |
| | MSE | MAE | MSE | MAE | MSE | MAE | MSE | MAE |
| Chronos-Large | 0.709 | 0.524 | 59.2 | 5.3 | 2e-4 | 0.011 | 0.079 | 0.174 |
| Chronos-Base | 0.671 | 0.487 | 57.1 | 5.1 | 3e-4 | 0.011 | 0.020 | 0.070 |
| Chronos-Small | 0.516 | 0.432 | 44.3 | 4.6 | 5e-4 | 0.014 | 0.150 | 0.250 |
| Chronos-Mini | 0.640 | 0.548 | 66.7 | 6.1 | 6e-4 | 0.016 | 0.849 | 0.638 |
| Chronos-Tiny | 0.798 | 0.581 | 57.7 | 5.5 | 2e-3 | 0.026 | 2.700 | 1.100 |
| TimesFM | 0.718 | 0.564 | 77.2 | 6.1 | 9e-5 | 0.006 | 0.807 | 0.533 |
| NBEATS-SarSim0 | 0.609 | 0.550 | 56.0 | 5.7 | 4e-4 | 0.014 | 0.375 | 0.445 |
| Chronos-SarSim0 | 0.650 | 0.510 | 63.2 | 5.8 | 4e-5 | 0.003 | 1.220 | 0.672 |
| PatchTST-SarSim0 | 0.750 | 0.550 | 55.1 | 5.6 | 4e-5 | 0.004 | 0.294 | 0.366 |

Table 12: Zero-shot forecasting performance on four nonlinear dynamical systems (Duffing, Lorenz, Lotka-Volterra, Pendulum), reported as MSE and MAE for each system.

## H  ADDITIONAL RESULTS

In this section, we describe additional empirical results based on non-linear dynamical systems.

### H.1  NON-LINEAR SYSTEMS RESULTS

To probe the behavior well outside the target regime of traditional statistics based time series, we ran an additional zero-shot experiment on four canonical nonlinear dynamical systems (Duffing, Lorenz, Lotka-Volterra, Pendulum), comparing several `Chronos` variants, `TimesFM`, and `NBEATS`/`PatchTST`/`Chronos` models trained on `SarSim0`. We generated datasets for the four non-linear dynamical systems. For each dynamical system, we simulated a single long trajectory of 20,000 time steps. The initial 16,000 steps are treated as burn-in, and the final 4,000 steps are used for forecasting evaluation. Evaluation results appear in Table 12, while the data-generation procedure for each system is summarized below.

Across the four canonical nonlinear systems we explored, the error levels of `SarSim0`-pretrained models are generally in the same range as those of the real-data foundation models. We see this as evidence that, on nonlinear dynamical systems far from our intended application domain and including aperiodic/chaotic behavior, pretraining on a SARIMA-based synthetic family still yields models that behave reasonably and on a similar scale to models trained on real data. This suggests that the inductive biases used to construct the simulator are not overly restrictive: while they clearly favour series dominated by trend, seasonality, and intermittency, they remain compatible with the level of out-of-domain generalization observed in real-data-pretrained models.

**Pendulum.** We consider a nonlinear simple pendulum described by its angular displacement and angular velocity. Each trajectory is initialized with a random starting angle and angular velocity, and then evolved forward in time using the standard equations of motion under gravity. The physical system is a point mass $m$ attached to a massless rigid rod of length $l$, moving in a vertical plane under constant gravitational acceleration $g$. The dynamics of the pendulum are governed by the following second-order differential equation:

$$\theta''(t) + \frac{g}{l}\sin(\theta(t)) = 0, \tag{13}$$

where:

- $\theta(t)$ is the angular displacement of the pendulum at time $t$,
- $\theta''(t)$ is the angular acceleration,
- $g$ is the acceleration due to gravity ($9.81\,\mathrm{m/s}^2$),
- $l$ is the length of the pendulum (set to $1.0\,\mathrm{m}$).

The system is further characterized by its initial conditions:

- the initial angle $\theta_0$, which is randomly chosen from a uniform distribution between $-\pi$ and $\pi$, and

- the initial angular velocity $\omega_0$, which is randomly chosen from a uniform distribution between $-1\,\mathrm{rad/s}$ and $1\,\mathrm{rad/s}$.

The motion of the pendulum is modeled using numerical methods, specifically the Euler method, which approximates the solution of the system of equations over discrete time steps.

**Duffing Oscillator.** We consider a nonlinear oscillator with damping and a cubic restoring term. For each run, we sample random initial displacement and velocity, and we then evolve the system under a periodic external forcing. The underlying dynamics follow the Duffing oscillator, a canonical nonlinear second-order system used to model oscillators with nonlinear stiffness and damping. Its evolution is governed by the differential equation:

$$\ddot{x} + \delta\dot{x} + \alpha x + \beta x^3 = \gamma\cos(\omega t),$$

where $x(t)$ denotes the oscillator's displacement, $y(t) = \dot{x}(t)$ its velocity, and $t$ the time variable. The system parameters are fixed to $\alpha = 1.0$ (linear stiffness), $\beta = 5.0$ (nonlinear stiffness), $\delta = 0.3$ (damping coefficient), $\gamma = 8.0$ (driving force amplitude), and $\omega = 0.5$ (angular frequency of the driving force). The oscillator is subject to a periodic external drive, and its evolution reflects the combined effects of the nonlinear restoring force and damping. We simulate the dynamics by numerically integrating the equations of motion with a simple time-stepping scheme, where dt is the time step and the initial conditions for $x$ and $y$ are drawn at random from a small interval. For this choice of parameters, the resulting motion exhibits chaotic behavior.

**Lotka-Volterra.** We also include a predator-prey population system, in which prey and predator counts evolve through their mutual interaction. Each trajectory starts from randomly chosen initial population levels, and the dynamics are then propagated forward according to the classical Lotka-Volterra equations. The governing equations for the Lotka-Volterra predator-prey model are given by:

$$\frac{dN_{\mathrm{prey}}}{dt} = \alpha N_{\mathrm{prey}} - \beta N_{\mathrm{prey}} N_{\mathrm{predator}}$$

$$\frac{dN_{\mathrm{predator}}}{dt} = \delta N_{\mathrm{prey}} N_{\mathrm{predator}} - \gamma N_{\mathrm{predator}},$$

where:

- $N_{\mathrm{prey}}$ is the population of the prey species,
- $N_{\mathrm{predator}}$ is the population of the predator species,
- $\alpha$ is the natural growth rate of the prey,
- $\beta$ is the predation rate (rate at which predators kill prey),
- $\delta$ is the rate at which predators increase due to consuming prey,
- $\gamma$ is the natural death rate of the predator.

In this formulation, the prey population grows exponentially when predators are absent, while the predator population decays exponentially if no prey are available. Their interaction leads to characteristic cyclical oscillations in both population sizes.

We simulate the system by numerically integrating the differential equations with a simple Euler scheme. At the start of each run, prey and predator populations are initialized at random within a prescribed interval. The parameters $\alpha = 1.1$, $\beta = 0.4$, $\delta = 0.1$, and $\gamma = 0.4$ are then used to update the populations at every time step.

**Lorenz '63.** We also consider a chaotic dynamical system with three state variables $x$, $y$, and $z$. Each trajectory is initialized from random starting conditions and then evolved forward in time using the Lorenz equations under standard parameter settings. The equations governing the Lorenz system are given by:

$$\frac{dx}{dt} = \sigma(y - x)$$

| | GiftEval | | M-SERIES | |
|---|---|---|---|---|
| | sCRPS | MASE | sCRPS | MASE |
| NBEATS-SarSim0-default | 0.602 | 0.849 | 0.096 | 0.869 |
| NBEATS-SarSim0-p=3 | 0.603 | 0.851 | 0.097 | 0.876 |
| NBEATS-SarSim0-p=7 | 0.600 | 0.846 | 0.096 | 0.867 |
| NBEATS-SarSim0-p=15 | 0.599 | 0.850 | 0.097 | 0.869 |
| NBEATS-SarSim0-p=20 | 0.602 | 0.852 | 0.097 | 0.872 |
| NBEATS-SarSim0-$r_{max} = 0.8$ | 0.601 | 0.848 | 0.096 | 0.870 |
| NBEATS-SarSim0-$r_{max} = 0.95$ | 0.600 | 0.847 | 0.097 | 0.873 |
| NBEATS-SarSim0-$R_{max} = 0.2$ | 0.601 | 0.844 | 0.096 | 0.869 |
| NBEATS-SarSim0-$R_{max} = 0.5$ | 0.598 | 0.846 | 0.096 | 0.866 |
| NBEATS-SarSim0-q=6 | 0.604 | 0.848 | 0.097 | 0.875 |
| NBEATS-SarSim0-Q=4 | 0.602 | 0.851 | 0.097 | 0.871 |

Table 13: Study of the sensitivity of NBEATS-SarSim0 to simulator hyperparameters.

$$\frac{dy}{dt} = x(\rho - z) - y$$

$$\frac{dz}{dt} = xy - \beta z,$$

where:

- $x$, $y$, and $z$ represent the state variables of the system, typically interpreted as the variables describing the convection rolls in the atmosphere,

- $\sigma$ is the Prandtl number, a measure of the fluid's viscosity, set to $10.0$,

- $\rho$ is the Rayleigh number, representing the temperature difference between the top and bottom of the fluid, set to $28.0$,

- $\beta$ is a geometric factor, set to $\frac{8}{3}$.

For this choice of parameters, the Lorenz system displays chaotic dynamics: even tiny perturbations in the initial state can evolve into drastically different trajectories over time. In our experiments, we integrate the system of differential equations using the Euler method over a finite sequence of time steps.

## H.2 SARSIM0 HYPERPARAMETER SENSITIVITY ANALYSIS

In our evaluation, we varied several simulator hyperparameters: the maximum AR order $p$, the AR pole radius bounds $r_{max}$ and $R_{max}$, seasonal and non-seasonal MA orders $(q, Q)$ around the default configuration, retraining NBEATS-SarSim0 under the same protocol. Results are summarized in Table 13. Across all these variants, performance on both GiftEval and the M-Series remains very similar to the default configuration. This indicates that SarSim0 is not a brittle "sweet spot": the configuration we use appears relatively benchmark-agnostic and robust to substantial hyperparameter changes, rather than being reliant on finely tuned settings.

## H.3 DETAILED GIFTEVAL RESULTS

### H.3.1 GIFTEVAL DOMAIN SPLIT

We report GiftEval results stratified by domain in Tables 14-15 (see Table 1 and its discussion for model and baseline details). An important contextual point is that the Econ/Fin subset of GiftEval is essentially the M4 competition data, which we show in Appendix C (Table 5) to be included in the pretraining corpora of Chronos, MOIRAI, and TimesFM. Consequently, on Econ/Fin these foundation models are not in a strict zero-shot regime, whereas SarSim0-pretrained backbones never see these series during training and therefore operate under strict zero-shot conditions. Even in this favorable setting for the real-data models, their accuracy advantage on Econ/Fin is relatively

|  | Econ/Fin | Energy | Healthcare | Nature | Sales | Transport | Web/CloudOps |
|---|---|---|---|---|---|---|---|
| Chronos-Base | 0.798 | 0.733 | 0.513 | 0.451 | 0.444 | 0.594 | 0.824 |
| Chronos-Small | 0.810 | 0.753 | 0.525 | 0.472 | 0.449 | 0.614 | 0.797 |
| MOIRAI-Large | 0.778 | 0.732 | 0.565 | 0.382 | 0.445 | 0.451 | 0.748 |
| MOIRAI-Small | 0.834 | 0.763 | 0.744 | 0.413 | 0.442 | 0.551 | 0.777 |
| TTM-R2-Pretrained | 1.316 | 0.944 | 1.375 | 0.506 | 0.765 | 0.750 | 1.055 |
| TimesFM | 0.761 | 0.783 | 0.690 | 0.411 | 0.421 | 0.591 | 0.935 |
| NBEATS | 1.027 | 1.088 | 0.755 | 0.656 | 0.507 | 0.688 | 0.722 |
| PatchTST | 0.854 | 0.712 | 0.609 | 0.428 | 0.426 | 0.535 | 0.553 |
| AutoARIMA | 0.873 | 0.969 | 0.603 | 0.811 | 0.561 | 0.884 | 1.144 |
| NBEATS-KernelSynth | 0.985 | 0.827 | 0.609 | 0.489 | 0.517 | 0.649 | 0.739 |
| NBEATS-ForecastPFN | 3.640 | 1.459 | 2.211 | 0.603 | 0.462 | 0.866 | 1.174 |
| PatchTST-KernelSynth | 0.955 | 0.782 | 0.586 | 0.479 | 0.518 | 0.605 | 0.754 |
| PatchTST-ForecastPFN | 3.904 | 1.751 | 3.029 | 0.656 | 0.515 | 1.141 | 1.333 |
| NBEATS-SarSim0 | 0.845 | 0.718 | 0.516 | 0.413 | 0.420 | 0.556 | 0.641 |
| PatchTST-SarSim0 | 0.821 | 0.687 | 0.511 | 0.380 | 0.417 | 0.546 | 0.596 |
| MLP-SarSim0 | 0.878 | 0.745 | 0.655 | 0.431 | 0.425 | 0.597 | 0.710 |

Table 14: nCRPS on GiftEval, weighted aggregation by domain. Lower values are better.

|  | Econ/Fin | Energy | Healthcare | Nature | Sales | Transport | Web/CloudOps |
|---|---|---|---|---|---|---|---|
| Chronos-Base | 0.783 | 0.924 | 0.644 | 0.823 | 0.726 | 0.712 | 1.140 |
| Chronos-Small | 0.797 | 0.948 | 0.607 | 0.852 | 0.733 | 0.737 | 1.144 |
| MOIRAI-Large | 0.845 | 1.026 | 0.699 | 0.750 | 0.711 | 0.601 | 1.125 |
| MOIRAI-Small | 0.985 | 1.069 | 0.848 | 0.807 | 0.731 | 0.731 | 1.136 |
| TTM-R2-Pretrained | 1.409 | 1.016 | 1.176 | 0.851 | 0.977 | 0.792 | 1.254 |
| TimesFM | 0.824 | 1.017 | 0.698 | 0.880 | 0.701 | 0.741 | 2.394 |
| NBEATS | 0.861 | 1.185 | 0.690 | 0.933 | 0.705 | 0.730 | 0.916 |
| PatchTST | 0.907 | 0.983 | 0.685 | 0.916 | 0.691 | 0.709 | 0.780 |
| AutoARIMA | 0.866 | 1.011 | 0.783 | 1.018 | 0.813 | 0.973 | 1.615 |
| NBEATS-KernelSynth | 1.013 | 1.131 | 0.705 | 0.975 | 0.803 | 0.846 | 1.033 |
| NBEATS-ForecastPFN | 6.009 | 2.066 | 2.229 | 1.134 | 0.769 | 1.165 | 1.688 |
| PatchTST-KernelSynth | 1.020 | 1.072 | 0.699 | 0.936 | 0.805 | 0.793 | 1.012 |
| PatchTST-ForecastPFN | 6.266 | 1.993 | 2.744 | 1.110 | 0.821 | 1.338 | 1.919 |
| NBEATS-SarSim0 | 0.830 | 0.965 | 0.620 | 0.817 | 0.688 | 0.718 | 0.918 |
| PatchTST-SarSim0 | 0.844 | 0.945 | 0.665 | 0.797 | 0.690 | 0.708 | 0.891 |
| MLP-SarSim0 | 0.985 | 1.004 | 0.777 | 0.858 | 0.695 | 0.767 | 0.976 |

Table 15: nMASE on GiftEval, weighted aggregation by domain. Lower values are better.

modest, with SarSim0 models trailing only slightly behind. On the remaining six domains (*i.e.*, excluding Econ/Fin), SarSim0-trained models provide either very comparable or clearly better accuracy. In terms of sCRPS, a SarSim0 backbone achieves the best score in 4 out of 6 domains (Energy, Healthcare, Nature, Sales), and is close to the best method on Transport and Web/CloudOps. For nMASE, SarSim0 is best in 2 out of 6 domains (Sales, Web/CloudOps), and it typically remains within a small margin of the strongest model in the others. Overall, this pattern suggests that the inductive biases of SarSim0 are not tuned to any single domain, but generalize robustly across diverse application areas.

A second observation from Tables 14-15 is the uniform dominance of SarSim0 over other synthetic generators. Fixing the backbone (N-BEATS or PatchTST) and varying only the pretraining corpus, SarSim0 consistently outperforms both KernelSynth and ForecastPFN across all GiftEval domains and for both nCRPS and nMASE. For example, NBEATS-SarSim0 and PatchTST-SarSim0 are

| frequency | 10S | 10T | 15T | 5T | A | D | H | M | Q | W |
|---|---|---|---|---|---|---|---|---|---|---|
| Chronos-Base | 1.442 | 0.546 | 0.833 | 0.712 | 0.984 | 0.476 | 0.562 | 0.854 | 0.850 | 0.621 |
| Chronos-Small | 1.331 | 0.629 | 0.859 | 0.711 | 1.014 | 0.500 | 0.569 | 0.823 | 0.856 | 0.614 |
| MOIRAI-Large | 1.464 | 0.566 | 0.809 | 0.480 | 0.758 | 0.473 | 0.497 | 0.786 | 0.749 | 0.726 |
| MOIRAI-Small | 1.638 | 0.590 | 0.809 | 0.551 | 0.766 | 0.483 | 0.552 | 0.985 | 0.803 | 0.788 |
| TTM-R2-Pretrained | 2.255 | 0.511 | 0.951 | 0.755 | 1.436 | 0.789 | 0.688 | 1.495 | 1.416 | 1.354 |
| TimesFM | 2.177 | 0.528 | 0.853 | 0.701 | 0.853 | 0.520 | 0.570 | 0.737 | 0.863 | 0.689 |
| NBEATS | 1.003 | 0.791 | 1.070 | 0.728 | 0.977 | 0.660 | 0.730 | 0.968 | 0.983 | 1.112 |
| PatchTST | 0.899 | 0.498 | 0.729 | 0.544 | 0.853 | 0.494 | 0.495 | 0.837 | 0.844 | 0.763 |
| AutoARIMA | 1.678 | 1.149 | 1.059 | 1.042 | 0.948 | 0.590 | 0.903 | 0.764 | 0.832 | 0.837 |
| NBEATS-KernelSynth | 1.149 | 0.545 | 0.823 | 0.674 | 1.259 | 0.556 | 0.658 | 0.829 | 0.950 | 0.761 |
| NBEATS-ForecastPFN | 2.142 | 0.823 | 1.425 | 1.001 | 2.940 | 0.909 | 1.013 | 1.518 | 3.061 | 1.500 |
| PatchTST-KernelSynth | 1.151 | 0.504 | 0.797 | 0.688 | 1.272 | 0.554 | 0.604 | 0.825 | 0.957 | 0.784 |
| PatchTST-ForecastPFN | 3.349 | 1.017 | 1.537 | 1.112 | 3.171 | 0.968 | 1.154 | 2.191 | 2.716 | 2.276 |
| NBEATS-SarSim0 | 1.153 | 0.501 | 0.718 | 0.576 | 0.963 | 0.470 | 0.546 | 0.713 | 0.827 | 0.649 |
| PatchTST-SarSim0 | 1.114 | 0.440 | 0.702 | 0.530 | 0.998 | 0.455 | 0.507 | 0.755 | 0.821 | 0.656 |
| MLP-SarSim0 | 1.408 | 0.534 | 0.743 | 0.616 | 1.211 | 0.482 | 0.570 | 0.840 | 0.912 | 0.716 |

Table 16: nCRPS on GiftEval, weighted aggregation by frequency. Lower values are better.

strictly better than their KernelSynth- and ForecastPFN-pretrained counterparts in Econ/Fin, Energy, Healthcare, Nature, Sales, Transport, and Web/CloudOps.

A third comparison is with the non-zero-shot NBEATS and PatchTST baselines that are fully supervised on each GiftEval dataset. For NBEATS, SarSim0-pretrained NBEATS strictly improves over dataset-specific NBEATS on all seven domains in nCRPS and on 6 of 7 domains in nMASE (the remaining Web/CloudOps gap is negligible), despite never seeing real data during pretraining. For PatchTST, PatchTST-SarSim0 matches or improves upon the supervised PatchTST baseline on the majority of domains (5/7 in nCRPS and 6/7 in nMASE), with only Web/CloudOps (and slightly Transport in nCRPS) still favouring in-domain supervised training.

Overall, this stratified view shows that much of the benefit usually attributed to per-dataset supervised training can instead be recovered from a single, domain-agnostic SarSim0 curriculum, with only a narrow slice of domains where additional task-specific fitting still offers an advantage.

### H.3.2 GIFTEVAL FREQUENCY SPLIT

We further stratify GiftEval by sampling frequency in Tables 16-17. As detailed in Appendix C, the pretraining corpora of Chronos, MOIRAI, TimesFM, and TTM include large portions of the M4 competition data, whose dominant frequencies are annual (A), quarterly (Q), and monthly (M). In fact, the Annual and Quarterly splits in GiftEval consist exclusively of M4-yearly and M4-quarterly, respectively. The Monthly split aggregates five datasets (Car Parts, Hospital, M4-monthly, Saugeen, US Births), several of which are also used in pretraining: Hospital, Saugeen, US Births, and Car Parts appear in MOIRAI 's pretraining corpus, and Saugeen and US Births in TTM 's. Thus, at A/Q/M frequencies, foundation models operate in a regime where their pretraining distribution is especially close to the test distribution, whereas SarSim0-pretrained models never see these series and are evaluated in a strict zero-shot setting.

Despite this strong advantage for real-data-pretrained models at A/Q/M, SarSim0 backbones remain highly competitive (in fact, best on Monthly split) and are often best at other frequencies. For the sub-hourly and minute-level frequencies (10T, 15T, 5T), PatchTST-SarSim0 attains the best nCRPS on 10T and 15T and is among the top performers on 5T, outperforming both foundation models and per-dataset supervised NBEATS in several cases. At daily (D) frequency, NBEATS-SarSim0 and PatchTST-SarSim0 slightly improve over the best real-data foundation models in nCRPS, while also yielding strong nMASE, indicating that the synthetic curriculum transfers well to operationally important daily horizons. For the weekly bucket (W), GiftEval aggregates eight datasets: Electricity, ETT1, ETT2, Hierarchical Sales, M4-weekly, Saugeen, Solar, and US

|  | 10S | 10T | 15T | 5T | A | D | H | M | Q | W |
|---|---|---|---|---|---|---|---|---|---|---|
| Chronos-Base | 2.481 | 1.088 | 0.887 | 0.862 | 0.918 | 0.714 | 0.791 | 0.857 | 0.768 | 0.762 |
| Chronos-Small | 2.479 | 1.204 | 0.919 | 0.872 | 0.943 | 0.737 | 0.802 | 0.827 | 0.774 | 0.746 |
| MOIRAI-Large | 2.486 | 1.125 | 0.984 | 0.672 | 0.749 | 0.727 | 0.792 | 0.825 | 0.712 | 0.931 |
| MOIRAI-Small | 2.419 | 1.161 | 0.986 | 0.773 | 0.751 | 0.752 | 0.899 | 1.016 | 0.774 | 0.968 |
| TTM-R2-Pretrained | 2.933 | 0.917 | 0.905 | 0.863 | 1.294 | 0.981 | 0.883 | 1.207 | 1.265 | 1.225 |
| TimesFM | 3.733 | 1.274 | 0.956 | 2.380 | 0.845 | 0.746 | 0.855 | 0.800 | 0.874 | 0.847 |
| NBEATS | 1.284 | 1.213 | 1.016 | 0.884 | 0.794 | 0.775 | 0.905 | 0.851 | 0.755 | 1.083 |
| PatchTST | 1.063 | 1.189 | 0.877 | 0.786 | 0.830 | 0.749 | 0.803 | 0.859 | 0.824 | 0.929 |
| AutoARIMA | 4.744 | 1.001 | 0.978 | 1.000 | 0.935 | 0.882 | 1.060 | 0.759 | 0.799 | 0.947 |
| NBEATS-KernelSynth | 1.784 | 1.146 | 0.958 | 0.874 | 1.268 | 0.888 | 1.003 | 0.846 | 0.923 | 0.924 |
| NBEATS-ForecastPFN | 3.844 | 1.794 | 1.765 | 1.261 | 3.317 | 1.317 | 1.648 | 1.685 | 3.996 | 2.011 |
| PatchTST-KernelSynth | 1.760 | 1.103 | 0.939 | 0.858 | 1.274 | 0.867 | 0.907 | 0.841 | 0.935 | 0.986 |
| PatchTST-ForecastPFN | 5.423 | 1.811 | 1.529 | 1.409 | 3.719 | 1.362 | 1.566 | 2.152 | 3.781 | 2.855 |
| NBEATS-SarSim0 | 1.830 | 0.996 | 0.842 | 0.757 | 0.944 | 0.746 | 0.818 | 0.746 | 0.800 | 0.807 |
| PatchTST-SarSim0 | 1.712 | 0.989 | 0.828 | 0.744 | 0.989 | 0.737 | 0.791 | 0.800 | 0.802 | 0.837 |
| MLP-SarSim0 | 1.961 | 1.110 | 0.864 | 0.800 | 1.312 | 0.763 | 0.868 | 0.897 | 0.947 | 0.891 |

Table 17: nMASE on GiftEval, weighted aggregation by frequency. Lower values are better.

Births. As summarized in Appendix C, this segment is heavily represented in the pretraining corpora of the real-data foundation models: Electricity, M4-weekly, and Solar are used to pretrain Chronos; Electricity and M4-weekly are used by TimesFM; and M4-weekly, Solar, Saugeen, and US Births are used by MOIRAI. In other words, weekly-series evaluation is favorable to these models, whose pretraining distributions are very close to the test distribution, whereas SarSim0-pretrained backbones remain strictly zero-shot on all eight datasets. Even in this regime, SarSim0 models are competitive: in nCRPS, NBEATS-SarSim0 and PatchTST-SarSim0 (0.649 and 0.656) trail only Chronos-Base/Small (0.621/0.614), and outperform TimesFM, MOIRAI. A similar pattern holds for nMASE, where NBEATS-SarSim0 and PatchTST-SarSim0 (0.807 and 0.837) sit just behind Chronos-Base/Small. This supports the view that SarSim0 does not rely on any single "sweet spot" frequency, but instead it transfers robustly even in settings where foundation models benefit from substantial pretraining overlap with the evaluation datasets.

When we move attention to synthetic pretraining only, SarSim0 shows a consistent advantage over KernelSynth and ForecastPFN for every frequency bucket. The best synthetic model in terms of both nCRPS and nMASE is always either NBEATS-SarSim0 or PatchTST-SarSim0. The gap is especially pronounced at coarse resolutions (A, Q, M) and at fine-grained sub-hourly/minute frequencies. This suggests that on forecasting tasks represented in GiftEval, the SARIMA-based simulator provides a more robust and broadly useful synthetic curriculum than either KernelSynth or ForecastPFN.

Finally, we compare SarSim0-pretrained models to the strong per-dataset supervised baselines, NBEATS and PatchTST. Both SarSim0 models achieve lower nCRPS than per-dataset counterparts on 7 out of 10 frequencies (10T, 15T, 5T, H, D, M, W). At a few regimes, supervised models retain a small edge, which we see as natural given their direct access to the target train split. Overall, this comparison reinforces that a single SarSim0-based pretraining run can match or exceed carefully tuned per-dataset training across wide range of frequencies.

### H.3.3 GIFTEVAL TERM SPLIT

In Tables 18-19, we further stratify GiftEval by term length: long (including horizons 720, 900), medium (including horizons 480, 600), and short (including horizons 6, 8, 12, 13, 14, 18, 30, 48, 60). Before discussing the results, it is important to note that, among these three strata, the short-term split is the most affected by pretraining overlap. It contains 28 datasets, of which six are the M4 competition subsets (daily, hourly, monthly, quarterly, weekly, yearly) used to pretrain all real-data foundation models (Chronos, MOIRAI, TimesFM, TTM), as documented in Appendix C. In addition, Chronos sees three more short-term datasets (Solar, KDD Cup 2018, Electricity);

|  | long | medium | short |
|---|---|---|---|
| Chronos-Base | 0.721 | 0.748 | 0.596 |
| Chronos-Small | 0.746 | 0.743 | 0.607 |
| MOIRAI-Large | 0.598 | 0.605 | 0.597 |
| MOIRAI-Small | 0.626 | 0.636 | 0.666 |
| TTM-R2-Pretrained | 0.774 | 0.816 | 0.939 |
| TimesFM | 0.742 | 0.748 | 0.635 |
| NBEATS | 0.809 | 0.805 | 0.823 |
| PatchTST | 0.527 | 0.548 | 0.628 |
| AutoARIMA | 1.152 | 0.990 | 0.808 |
| NBEATS-KernelSynth | 0.667 | 0.710 | 0.710 |
| NBEATS-ForecastPFN | 0.994 | 1.056 | 1.270 |
| PatchTST-KernelSynth | 0.646 | 0.666 | 0.696 |
| PatchTST-ForecastPFN | 1.137 | 1.257 | 1.517 |
| NBEATS-SarSim0 | 0.574 | 0.601 | 0.612 |
| PatchTST-SarSim0 | 0.538 | 0.561 | 0.592 |
| MLP-SarSim0 | 0.601 | 0.641 | 0.660 |

Table 18: nCRPS on GiftEval, weighted aggregation by term length. Lower values are better.

|  | long | medium | short |
|---|---|---|---|
| Chronos-Base | 1.040 | 1.041 | 0.768 |
| Chronos-Small | 1.078 | 1.048 | 0.780 |
| MOIRAI-Large | 0.985 | 0.961 | 0.807 |
| MOIRAI-Small | 1.035 | 1.021 | 0.888 |
| TTM-R2-Pretrained | 1.043 | 1.036 | 1.005 |
| TimesFM | 1.623 | 1.447 | 0.823 |
| NBEATS | 1.056 | 1.036 | 0.863 |
| PatchTST | 0.881 | 0.859 | 0.833 |
| AutoARIMA | 1.615 | 1.026 | 0.935 |
| NBEATS-KernelSynth | 1.069 | 1.101 | 0.925 |
| NBEATS-ForecastPFN | 1.676 | 1.660 | 1.729 |
| PatchTST-KernelSynth | 1.022 | 1.001 | 0.912 |
| PatchTST-ForecastPFN | 1.681 | 1.666 | 1.879 |
| NBEATS-SarSim0 | 0.923 | 0.907 | 0.802 |
| PatchTST-SarSim0 | 0.896 | 0.876 | 0.801 |
| MLP-SarSim0 | 0.959 | 0.955 | 0.871 |

Table 19: nMASE on GiftEval, weighted aggregation by term length. Lower values are better.

MOIRAI sees twelve more (Car Parts, Covid Deaths, Hierarchical Sales, Hospital, Jena Weather, KDD Cup 2018, Loop Seattle, Saugeen, Solar, SZ-Taxi, Temperature Rain, US Births); and TimesFM sees Jena Weather and Electricity. Consequently, a substantial fraction of the short-term evaluation split consists of datasets that foundation models have already seen during pretraining, whereas SarSim0 models see none of them. Despite this favorable overlap for the real-data models, on short-term nCRPS SarSim0-pretrained backbones (PatchTST-SarSim0: 0.592, NBEATS-SarSim0: 0.612) outperform all foundation models, and on short-term nMASE they are only slightly behind Chronos. Across long and medium terms, which are much less affected by leakage, SarSim0-pretrained models are noticeably ahead of the foundation models.

When we move attention to synthetic-only pretraining, SarSim0 clearly dominates the other synthetic baselines across all term lengths. In Tables 18-19, NBEATS- and PatchTST-SarSim0 achieve lower nCRPS and nMASE than their KernelSynth and ForecastPFN counterparts in all three splits (long, medium, and short).

| | multivariate | univariate |
|---|---|---|
| Chronos-Base | 0.685 | 0.627 |
| Chronos-Small | 0.684 | 0.647 |
| MOIRAI-Large | 0.635 | 0.572 |
| MOIRAI-Small | 0.640 | 0.659 |
| TTM-R2-Pretrained | 0.833 | 0.907 |
| TimesFM | 0.717 | 0.652 |
| NBEATS | 0.790 | 0.837 |
| PatchTST | 0.556 | 0.613 |
| AutoARIMA | 1.032 | 0.826 |
| NBEATS-KernelSynth | 0.700 | 0.700 |
| NBEATS-ForecastPFN | 1.113 | 1.194 |
| PatchTST-KernelSynth | 0.683 | 0.674 |
| PatchTST-ForecastPFN | 1.217 | 1.501 |
| NBEATS-SarSim0 | 0.594 | 0.607 |
| PatchTST-SarSim0 | 0.546 | 0.595 |
| MLP-SarSim0 | 0.628 | 0.655 |

Table 20: nCRPS on GiftEval, weighted aggregation by variate type. Lower values are better.

### H.3.4 GIFTEVAL VARIATE TYPE SPLIT

| | multivariate | univariate |
|---|---|---|
| Chronos-Base | 1.013 | 0.780 |
| Chronos-Small | 1.026 | 0.797 |
| MOIRAI-Large | 1.023 | 0.773 |
| MOIRAI-Small | 1.021 | 0.890 |
| TTM-R2-Pretrained | 1.072 | 0.980 |
| TimesFM | 1.497 | 0.829 |
| NBEATS | 0.998 | 0.892 |
| PatchTST | 0.906 | 0.805 |
| AutoARIMA | 1.318 | 0.912 |
| NBEATS-KernelSynth | 1.098 | 0.914 |
| NBEATS-ForecastPFN | 1.802 | 1.627 |
| PatchTST-KernelSynth | 1.049 | 0.885 |
| PatchTST-ForecastPFN | 1.740 | 1.825 |
| NBEATS-SarSim0 | 0.943 | 0.781 |
| PatchTST-SarSim0 | 0.904 | 0.788 |
| MLP-SarSim0 | 0.979 | 0.854 |

Table 21: nMASE on GiftEval, weighted aggregation by variate type. Lower values are better.

We next stratify GiftEval by variate type (Tables 20-21). The univariate split contains 20 datasets (Car Parts, Covid Deaths, Electricity, Hierarchical Sales, Hospital, KDD Cup 2018, Loop Seattle, M-Dense, M4 Daily/Hourly/Monthly/Quarterly/Weekly/Yearly, Restaurant, Saugeen, Solar, SZ-Taxi, Temperature Rain, US Births). As detailed in Appendix C and Section H.3.3, a large fraction of these are used in pretraining the real-data foundation models (all overlapping datasets are univariate). In contrast, the multivariate split provides a much cleaner zero-shot testbed: to the best of our knowledge, there is no pretrain-test leakage in this segment.

Against this backdrop, SarSim0-pretrained backbones are highly competitive on the univariate split: PatchTST-SarSim0 (0.595) and NBEATS-SarSim0 (0.607) lie close to the leading MOIRAI-Large (0.572) in nCRPS. In nMASE, MOIRAI-Large again leads (0.773), with NBEATS-SarSim0

| | GiftEval | | M-SERIES | |
|---|---|---|---|---|
| | sCRPS | MASE | sCRPS | MASE |
| AutoARIMA | 0.912 | 1.074 | 0.096 | 0.843 |
| NBEATS-SarSim0 | **0.602** | **0.849** | **0.096** | 0.869 |
| No SARIMA-2 | 0.655 | 0.913 | 0.104 | 0.941 |
| No Noisers | 0.609 | 0.856 | **0.096** | **0.860** |
| SarimaOnly | 0.672 | 0.940 | 0.101 | 0.891 |

Table 22: Extended ablation on simulator components, including an additional SARIMA-only row (NBEATS-SarimaOnly) trained solely on pure SARIMA samples, without Noisers or SARIMA-2. On GiftEval, this SARIMA-only student already substantially outperforms the practical per-series baseline AutoARIMA in both sCRPS and MASE.

(0.781) and PatchTST-SarSim0 (0.788) essentially matching the best real-data models and improving over the per-dataset PatchTST/NBEATS baselines (0.805/0.892).

The picture is even more interesting on the multivariate split, where pretrain-test overlap is much less pronounced. Here, PatchTST-SarSim0 attains the best nCRPS (0.546), ahead of MOIRAI-Large (0.635), Chronos-Base (0.685), TimesFM (0.717), and the per-dataset PatchTST baseline (0.556). Across both variate types, SarSim0-pretrained models consistently outperform their KernelSynth and ForecastPFN counterparts. Taken together, these results indicate that a univariate SARIMA-based simulator can support strong generalization not only to unseen univariate series, but also to multivariate benchmarks, without explicitly modelling cross-series structure.

## H.4 ADDITIONAL ABLATION COMPARING SARIMA-ONLY CASE AND AutoARIMA

We define a cleaner experiment here to study the relationship between AutoARIMA and the generalization of models trained on SarSim0 and its components. In particular, Table 22 presents an additional row, in which we study the accuracy of NBEATS trained exclusively on SARIMA process (row SarimaOnly) without the use of Noisers or SARIMA-2. The conclusions remain aligned with our previous observations. On GiftEval, even the SARIMA-only student (NBEATS-SarimaOnly) substantially outperforms AutoARIMA (0.672/0.912 vs. 0.940/1.074 in sCRPS/MASE), despite being trained on essentially the same model family. On the M-Series, the picture is more mixed, as before: AutoARIMA remains better in MASE (0.843 vs. 0.869/0.891), while sCRPS is comparable. This is consistent with our discussion that the classical M-Series are particularly favorable to per-series ARIMA/ETS-type models and with our original characterization of this phenomenon as early, domain-dependent evidence rather than a universal effect.

