# ZERO-SHOT FORECASTING BY SIMULATION ALONE

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

 cross-frequency models that generalize across these zero-shot benchmarks under a strict no-leakage guarantee. On these benchmarks, simulated data can be competitive with, or even outperform, real-data pretraining, substantially closing the gap between small efficient architectures and large foundation models. On GiftEval, we see early evidence for an emergent "student beats teacher" generalization behavior. Here, by "emergent" we mean the ability of a globally trained neural model to generalize beyond the forecasting accuracy implied by the key simulator component, i.e., SARIMA, whose native forecasting capability is instantiated in the strong practical per-series baseline, AutoARIMA.

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

 building our simulator on `SARIMA`, we therefore inherit both the statistical grounding of classical models and a direct connection to `AutoARIMA`, providing a natural "teacher", harmonizing with the complementary inductive biases of other "student" neural architectures, which we will see outperform `AutoARIMA` models after training on `SARIMA` simulations. Additionally, as we will see in Figure 3, `SARIMA` reproduces the statistical structure of real time-series data, providing a credible training signal.

### 4.1 SARIMA SIMULATOR

Without loss of generality, we start the exposition with the *Auto-Regressive Moving Average* (`ARMA`) process of order $p', q$ that can be defined as:

$$y_t - \alpha_1 y_{t-1} - \cdots - \alpha_p y_{t-p'} = \varepsilon_t + \vartheta_1 \varepsilon_{t-1} + \cdots + \vartheta_q \varepsilon_{t-q}. \tag{3}$$

Here $\alpha_{1:p'}$ are auto-regressive coefficients, $\vartheta_{1:q}$ are moving average coefficients, and $\varepsilon_{t-q:t}$ are zero-mean i.i.d. innovations. With the lag operator $L$ defined as $Ly_t = y_{t-1}$, this process can be written compactly in polynomial form:

$$\left(1 - \sum_{i=1}^{p} \phi_i L^i\right)(1 - L)^d y_t = \left(1 + \sum_{i=1}^{q} \vartheta_i L^i\right) \varepsilon_t, \tag{4}$$

where $p = p' - d$ and coefficients $\phi$ are derived from coefficients $\alpha$ to factor out $d$ unit roots of the polynomial, corresponding to the finite difference time domain integrator, $y_t \leftarrow y_t + y_{t-1}$. Consequently, `ARMA` with integrator ($d > 0$) is called `ARIMA` (integrated `ARMA`).

In principle, `ARIMA` could be used directly as a time-series simulator, by randomly generating initial state $y_0$, orders $p', q$; coefficients $\alpha, \vartheta$; excitation noise $\varepsilon$ and unrolling equation (3) through time.

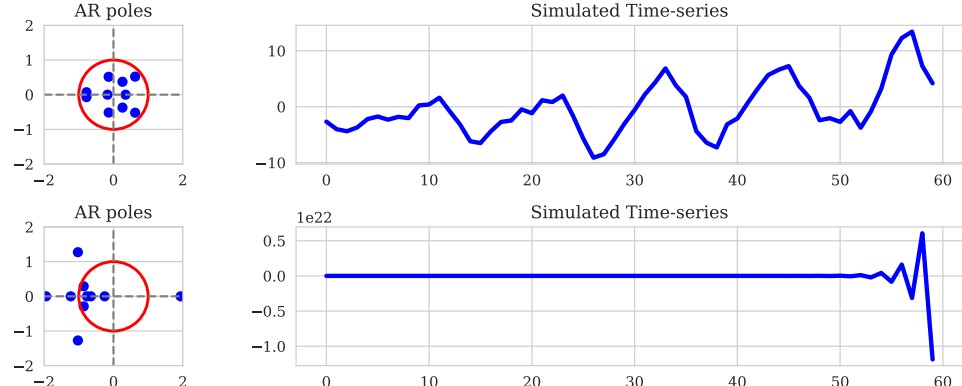

Figure 2: **Sampling of SARIMA poles.** The SARIMA order-10 AR process poles are shown along with the unit circle on the left. The resulting generated processes with these poles are shown on the right. The top pane shows poles sampled according to the proposed procedure, resulting in a realistic looking time-series. The bottom pane shows the result of unconstrained random generation of AR coefficients, resulting in a divergent time-series that is useless from the perspective of training foundation time-series models.

However, as shown in Figure 2 (bottom), setting the coefficients arbitrarily destabilizes the system and results in divergent nonsensical time-series that are not useful for training a forecasting model. To circumvent this stability issue, we propose instead to operate directly in the model's pole space, guaranteeing valid process dynamics by construction. In particular, there is a link between the summation-based polynomial form (4) and the product-based pole representation:

$$\phi(L) = 1 - \sum_{i=1}^{p} \phi_i L^i = \prod_{i=1}^{p} (1 - \varphi_i L). \tag{5}$$

Well-behavedness of the AR part is guaranteed if the poles, $\varphi_i$, lie within unit circle, $|\varphi_i| < 1$, (Hamilton, 1994, Chapter 3). Thus, we propose to sample the poles of the AR and MA transfer functions, $\{\varphi_i\}_{i=1}^{p}$ and $\{\vartheta_j\}_{j=1}^{q}$ respectively, and then derive the system coefficients $\alpha$ and $\vartheta$ from the pole representation using product expansion (*e.g.* numpy's np.poly). A typical realization of the process sampled using this approach is shown in Figure 2 (top). In contrast to the bottom part, all poles of the AR system are located within unit circle, resulting in a well-behaved simulation result.

To complete the polynomial specification of the base SARIMA, we include the seasonal part with parameters $s$ (period), $D$ (integration order), $P$ (seasonal AR order), $Q$ (seasonal MA order):

$$\left(1 - \sum_{i=1}^{p} \phi_i L^i\right) \left(1 - \sum_{j=1}^{P} \Phi_j L^{js}\right) (1-L)^d (1-L^s)^D \, y_t = \left(1 + \sum_{i=1}^{q} \vartheta_i L^i\right) \left(1 + \sum_{j=1}^{Q} \Theta_j L^{js}\right) \varepsilon_t \, .$$

We note that the stability of this overall system is now dictated by the combined polynomial $\phi(L)\Phi(L)$ whose poles emerging from the solution of $\phi(L)\Phi(L^s) = 0$ are the union of the poles $\varphi_i$ of $\phi(L)$ and $\widetilde{\varphi}_i^{1/s}$ of $\Phi(L^s)$. Therefore, to ensure the well-behavedness of the combined system we need to additionally constrain the poles $\widetilde{\varphi}_i^{1/s}$ to also lie inside the unit circle.

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

**M-Series** is composed of four forecasting tasks comprising over 100,000 time-series, curated from established forecasting competitions: M1 (Makridakis et al., 1982), M3 (Makridakis & Hibon, 2000), M4 (Makridakis et al., 2020), and Tourism (Athanasopoulos et al., 2011). These datasets, summarized in Table 4 of Appendix B, represent a broad range of domains and temporal frequencies.

| | GiftEval | | M-SERIES | | Inference | Zero-Shot |
|---|---|---|---|---|---|---|
| | sCRPS | MASE | sCRPS | MASE | Time (m) | |
| Chronos-Base | 0.647 | 0.870 | 0.103 | 0.878 | 2103 | ✓(yellow) |
| Chronos-Small | 0.662 | 0.892 | 0.103 | 0.882 | 702 | ✓(yellow) |
| MOIRAI-Large | 0.599 | 0.874 | 0.128 | 1.027 | 3976 | ✓(yellow) |
| MOIRAI-Small | 0.650 | 0.946 | 0.124 | 1.089 | 994 | ✓(yellow) |
| TTM-R2-Pretrained | 0.873 | 1.020 | 0.155 | 1.118 | 43 | ✓(yellow) |
| TimesFM | 0.680 | 1.077 | 0.098 | 0.930 | 155 | ✓(yellow) |