# OpenReview forum: "Zero-shot Forecasting by Simulation Alone"
_ICLR.cc/2026/Conference — ICLR 2026 Poster_

### Official Review · Reviewer_gKMt · 2025-10-19

**Soundness:** 3
**Presentation:** 2
**Contribution:** 3
**Rating:** 6
**Confidence:** 4

**Summary:**

The authors propose a new pipeline to generate synthetic data for the pre-training of time-series foundation models. They show that models trained purely on their synthetic data can outperform leading time-series foundation models trained on real data.

**Strengths:**

Being able to generate high-quality synthetic data that fully replace real data is a big deal. This has the potential to overcome the scarcity of real-world time-series data and allow scaling time-series foundation models beyond the current billion-parameter regime.

**Weaknesses:**

There are some strong inductive bias built into the SARSIM0 process, such as bi-seasonality, which are well suited for the tasks in GIFT-EVAL and M-Series benchmarks but may fail in other application domains, especially in scientific machine learning. See my questions below.

**Questions:**

* How well do models trained on your synthetic data perform on predicting dynamical processes not described by ARIMA processes? For example, deterministic chaos.
* I am asking this because SARSIM0 has a very strong inductive bias towards time series with seasonality or bi-seasonality, while chaotic time series are aperiodic. Can the authors comment on how models trained with SARSIM0 fair with aperiodic time series in general?
* Have you tried train a model with both SARSIM0 data and real data? Will it outperform models trained on either synthetic data or real data alone?

---

> ### Author Response · Authors · 2025-11-25
> **Author response to Reviewer gKMt, 1/2**
>
> Thank you for the detailed and supportive feedback. We provide our detailed response below. If you have any additional questions based on our response, please don't hesitate to reach out to us.
>
> > How well do models trained on your synthetic data perform on predicting dynamical processes not described by ARIMA processes? For example, deterministic chaos.
>
> Thank you for encouraging us to look into this. To probe behaviour well outside the target regime of  business-related time series, we ran an additional zero-shot experiment on four canonical nonlinear dynamical systems (Duffing, Lorenz, Lotka-Volterra, Pendulum), comparing several Chronos variants, TimesFM, and NBEATS/PatchTST/Chronos models trained on SarSim0.
>
> |         | Duffing-MSE | Duffing-MAE | Lorenz-MSE | Lorenz-MAE | Lotka-Volterra-MSE | Lotka-Volterra-MAE | Pendulum-MSE | Pendulum-MAE |
> | ------- |:-----------:|:-----------:|:----------:|:----------:| ------------------ | ------------------ | ------------ | ------------ |
> | Chronos-Large |    0.709    |    0.524    |    59.2    |    5.3     | 2e-4               | 0.011              | 0.079        | 0.174        |
> | Chronos-Base |    0.671    |    0.487    |    57.1    |    5.1     | 3e-4               | 0.011              | 0.02        | 0.07        |
> | Chronos-Small |    0.516    |    0.432    |    44.3    |    4.6     | 5e-4               | 0.014              | 0.15        | 0.25        |
> | Chronos-Mini |    0.640    |    0.548    |    66.7    |    6.1     | 6e-4               | 0.016              | 0.849        | 0.638        |
> | Chronos-Tiny |    0.798    |    0.581    |    57.7    |    5.5     | 2e-3               | 0.026              | 2.7        | 1.1        |
> | TimesFM |    0.718    |    0.564    |    77.2    |    6.1     | 9e-5               | 0.006              | 0.807        | 0.533        |
> | NBEATS-SarSim0        |       0.609      |     0.550        |       56.0     |       5.7    |        4e-4            |           0.014         |       0.375       |     0.445         |
> | Chronos-SarSim0        |       0.650      |     0.510        |       63.2     |       5.8     |        4e-5            |           0.003         |       1.22       |     0.672         |
> | PatchTST-SarSim0        |       0.750      |     0.550        |       55.1     |       5.6     |        4e-5            |           0.004         |       0.294       |     0.366         |
>
>
> > I am asking this because SARSIM0 has a very strong inductive bias towards time series with seasonality or bi-seasonality, while chaotic time series are aperiodic. Can the authors comment on how models trained with SARSIM0 fair with aperiodic time series in general?
>
> Across the four canonical nonlinear systems we explored, the error levels of SarSim0-pretrained models are generally in the same range as those of the real-data foundation models. We see this as evidence that, on nonlinear dynamical systems far from our intended application domain and including aperiodic/chaotic behaviour, pretraining on a SARIMA-based synthetic family still yields models that behave reasonably and on a similar scale to models trained on real data. This suggests that the inductive biases used to construct the simulator are not overly restrictive: while they clearly favour series dominated by trend, seasonality, and intermittency, they remain compatible with the level of out-of-domain generalization observed in real-data-pretrained models.

---

> ### Author Response · Authors · 2025-11-25
> **Author response to Reviewer gKMt, 2/2**
>
> > Have you tried train a model with both SARSIM0 data and real data? Will it outperform models trained on either synthetic data or real data alone?
>
> In the current work we do not mix SarSim0 data with real data. This is deliberate: our primary goal is to study a strict synthetic-only, leakage-free time series pretraining and zero-shot inference, where the simulator is never exposed to real evaluation series (or closely related data), and to understand how far one can go in that regime.
>
> There is already evidence in the literature that mixing real and synthetic data can be beneficial. For example, Chronos combines GP-based synthetic data (KernelSynth) with large real-data corpora, reporting improved performance compared to using real data alone. Showing that SarSim0 can also be helpful as part of such a mixed synthetic-real curriculum would therefore be a natural but relatively incremental extension of these results, and would not change the core message of our paper.
>
> Instead, we chose to focus on a purely synthetic pretraining and to evaluate it against real-data baselines. In Table 1, we compare to models trained on real data both in the zero-shot regime (Chronos, TimesFM, MOIRAI, TTM, etc., pretrained on large real corpora) and in the per-dataset regime (NBEATS/PatchTST trained directly on each benchmark), and find that SarSim0-only pretraining yields models that are competitive with, and often better than, these real-data alternatives on GIFT-Eval and the M-Series. We view this synthetic-only competitiveness as the main contribution of our work.
>
> To dig a bit deeper in this direction, we additionally took the Chronos-Small T5 backbone, plugged it into our pipeline (using our sequence sampling and loss definition), and trained it with batch size 1024 for 500K steps on SarSim0 using input sequence length 512 for compatibility with the original Chronos configuration. The results are:
>
>
> |                       | GiftEval-sCRPS | GiftEval-MASE | M-Series-sCRPS | M-Series-MASE |
> | --------------------- |:--------------:|:-------------:|:--------------:|:-------------:|
> | Chronos-Small         |     0.662      |     0.892     |     0.103      |     0.882     |
> | Chronos-Small-SarSim0 |     0.634      |     0.921     |     0.098      |     0.886     |
> | PatchTST-SarSim0      |     0.576      |     0.838     |     0.102      |     0.922     |
> | NBEATS-SarSim0        |     0.602      |     0.849     |     0.096      |     0.869     |
>
>
> Chronos-Small-SarSim0 achieves very similar performance to Chronos-Small trained on the original mixed real + synthetic corpus (improving sCRPS on both GIFT-Eval and M-Series, with slightly higher MASE), which supports the view that SarSim0 can serve as a viable standalone pretraining source even for foundation-model style backbones.
>
> We fully agree that hybrid training (e.g., synthetic pretraining followed by real fine-tuning, or joint synthetic-real curricula) is a promising practical direction. We expect that in many real-world settings such mixtures will surpass either pure synthetic or pure real training alone. However, systematically exploring mixing ratios, schedules, and domain-specific trade-offs would require a substantial separate study and would blur the clean zero-shot setting we focus on here --- a focus which we believe provides value. We will clarify our focus and leave a systematic study of synthetic-real co-training as an important direction for future work.

---

> ### Author Response · Authors · 2025-12-02
> **Author 2nd round response to Reviewer gKMt, summary of revisions**
>
> We once again sincerely thank you for your thoughtful comments and your help in improving the paper. We have implemented the following revisions to address concerns you raised.
>
> > How well do models trained on your synthetic data perform on predicting dynamical processes not described by ARIMA processes? For example, deterministic chaos.
>
> > I am asking this because SARSIM0 has a very strong inductive bias towards time series with seasonality or bi-seasonality, while chaotic time series are aperiodic. Can the authors comment on how models trained with SARSIM0 fair with aperiodic time series in general?
>
> To address the concern we have added Appendix H.1 summarizing the non-linear system results as well as the following asociated discussion at the end of Empirical Results section:
>
> *To probe behaviour far outside our target regime of business-style series, we also ran a zero-shot experiment on four canonical nonlinear dynamical systems (Duffing, Lorenz, Lotka-Volterra, pendulum), comparing several Chronos variants, TimesFM, and SarSim0-pretrained models (see Appendix H.1 for details). SarSim0-pretrained models attain error levels comparable to real-data foundation models. This suggests that SarSim0’s inductive biases, while geared toward trend/seasonality/intermittency regimes typical of industrial forecasting, yield models whose out-of-domain generalization is similar to that of real-data-pretrained models on these systems.*
>
> > Have you tried train a model with both SARSIM0 data and real data? Will it outperform models trained on either synthetic data or real data alone?
>
> In the current work we do not mix SarSim0 data with real data. This is deliberate: our primary goal is to study a strict synthetic-only, leakage-free time series pretraining and zero-shot inference, where the simulator is never exposed to real evaluation series (or closely related data), and to understand how far one can go in that regime.
>
> In the revised manuscript, we now include Chronos-Small models pretrained on KernelSynth, ForecastPFN, and SarSim0 in Table 1. Concretely, we take the Chronos-Small T5 backbone, plug it into our training pipeline (our sequence sampling and loss), and train it for 500K steps with batch size 1024 on SarSim0. Chronos-Small-SarSim0 achieves similar or better performance to Chronos-Small trained on the original mixed real + synthetic corpus (improving sCRPS on both GIFT-Eval and M-Series, with slightly higher MASE), which supports the view that SarSim0 can serve as a viable standalone pretraining source even for foundation-model style backbones like Chronos.
>
> We fully agree that hybrid training (e.g., synthetic pretraining followed by real fine-tuning, or joint synthetic-real curricula) is a promising practical direction. We expect that in many real-world settings such mixtures will surpass either pure synthetic or pure real training alone. However, systematically exploring mixing ratios, schedules, and domain-specific trade-offs would require a substantial separate study and would blur the clean zero-shot setting we focus on here, a focus which we believe provides value. In the revised manuscript we clarify our focus and leave a systematic study of synthetic-real co-training as an important direction for future work (see Discussion section):
>
> *While we expect hybrid synthetic-real training to outperform either source alone in many practical settings, we deliberately focus on the clean synthetic-only zero-shot regime, leaving synthetic-real co-training as an important direction for future work.*

---

### Official Review · Reviewer_qasc · 2025-10-20

**Soundness:** 3
**Presentation:** 2
**Contribution:** 2
**Rating:** 4
**Confidence:** 4

**Summary:**

This paper introduces SarSim0, a fast synthetic time‑series generator. The authors train forecast models entirely on simulated data and then test their zero‑shot performance on real datasets from the GIFT-Eval and M4 datasets. The SarSim0 pipeline first samples stable SARIMA processes by sampling AR/MA poles inside the unit circle, then composes bi‑seasonal structure via an additive/multiplicative SARIMA‑2 modulation, and finally overlays level‑dependent heavy‑tailed noise (Poisson, generalized‑Gamma, lognormal) to capture burstiness and intermittency. Because the authors’ approach generates data on‑the‑fly at scale, compact backbones (N‑BEATS, PatchTST, MLP) trained solely on a large set of SarSim0 time series achieve strong cross‑frequency zero‑shot accuracy matching or surpassing larger pretrained models like Chronos. The authors also show that pretraining with their synthetic data leads to stronger results on GIFT-Eval than pretraining with earlier synthetic data sources (PFN and KernelSynth). When comparing to fully-trained models, the authors find that models pretrained on SarSim0 data outperform direct forecasts using AutoARIMA, a phenomenon the authors refer to as “student beats teacher” generalization.

**Strengths:**

The SARIMA-2 approach is an appropriate generating model that describes many practical time series. As noted by the authors, many demand or utility time series benchmarks consist of a slow process (like inflation in a financial time series) modulating a fast process (like seasonal demand). Within this generating model class, the authors make appropriate choices to enforces stability in the AR dynamics by sampling poles inside the unit circle. This likely helps preserve diversity in the generated time series.

The evaluations and experiments are appropriate in scope. The authors use GIFT-Eval, which is the current leading benchmark for zero-shot evaluations. They also perform ablations of each component of their generator, allowing them to identify that SARIMA-2 (biseasonality) is a key driver of their approach.
I also like the experiment design of comparing fully-trained NBEATS/PatchTST against variants pretrained on prior synthetic data generators (KernelSynth/PFN).

The authors give a nice demonstration that their generated data co-clusters with real datasets from the M4 competition. Because nonlinear embeddings are neighbor based, this helps show that their data convincingly “passes” as real time series data

**Weaknesses:**

**Novelty.** This is not the first paper to train a zero-shot forecast model  purely on simulation data and report strong results. ForecastPFN and TabPFN first used this approach, while Mamba4Cast uses an SSM on PFN synthetic data to achieve strong results on the original GluonTS datasets. While the idea itself is not the first of its kind, the arguments in favor of this paper could be (1) the particular choice of synthetic data generation, and (2) the empirical results. I do not currently feel that either is strong enough to recommend acceptance.

Ad hoc data augmentation. Regarding the data generation procedure, I consider the approach to be sufficiently ad hoc that it’s hard to see why a SARIMA approach would be universally preferable. There are not any theoretical guarantees or general bounds that establish that this approach will always produce more diverse or realistic data than prior methods. Prior approaches fall broadly into PFN and its variants (as in ForecastPFN/TabPFN), augmentations thereof (as in Mamba4cast, which introduces spike injection and damping augmentation, or Gaussian-process based methods like KernelSynth used by Chronos. There’s also structured approaches like CauKer, which combine Gaussian Processes with a structured causal model. I can see the author’s ARIMA-based procedure as helping cover data types not covered by other methods; for example, long history dependence when the ARIMA process has a long effective memory due to slow damping. However, it’s hard to argue that any of these dataset generators are universally preferable.

**Experiments.** The experiments show that models like NBEATS or PatchTST become competitive with models like Chronos when pretrained on the SarSim0 dataset. I don’t consider these results strong enough to believe that they justify selecting this data generator over others, because I think there is a bit of a “no free lunch” here: the authors pick a generator that handles multiseasonality well, and it turns out this is useful for M4 and GIFT-Eval, which has many time series that fall into that class. But had the authors picked a multi-scale or trend-dominated test set, a different heuristic for data generation approach could have been preferable. Since there are infinite choices that could be made during the data generation procedure, the overperformance of this particular set of choices mainly tells us about the properties of GIFT-Eval and M4.

**Emergence.** I do not agree with the “emergent effect” argument.  A neural model pretrained with SARIMAX datasets outperforming AutoARIMA does not imply emergent effects. Firstly, Sarima-2 does not directly fall within the model class directly expressible by AutoARIMA. Secondly, initialization and hyperparameters can prevent a model like AutoARIMA from exactly fitting the true generator.

**Questions:**

1. The unit circle pole requirement acts as a stability condition on the AR process generating the data. Are there any mathematical conditions or theoretical guarantees on the distributional properties of the resulting time series?

2. Can you directly quantify the verisimilitude and diversity of your generated data, particularly compared to prior PFN or structured causal model approaches? For example, since you have a UMAP, you could calculate a silhouette score. A statistical significance test would add depth to this comparison.

3. Are you fine-tuning Chronos, Moirai, etc on the SarSim0 dataset? I could not tell if these experiments are what are shown in the first rows Table 1, or if that shows the base performance of those models pretrained on their original pretraining splits.

4. I am confused by Table 1. Are the three models (lines 387-389) fully-trained on the context? And why are Chronos-Base and others not marked as zero-shot models?

---

> ### Author Response · Authors · 2025-11-25
> **Author response to Reviewer qasc, 1/5**
>
> Thank you for your thoughtful and constructive feedback. We provide our detailed response below. If you have any additional questions based on our response, please don't hesitate to let us know.
>
> > Novelty. This is not the first paper to train a zero-shot forecast model purely on simulation data and report strong results. ForecastPFN and TabPFN first used this approach, while Mamba4Cast uses an SSM on PFN synthetic data to achieve strong results on the original GluonTS datasets. While the idea itself is not the first of its kind, the arguments in favor of this paper could be (1) the particular choice of synthetic data generation, and (2) the empirical results. I do not currently feel that either is strong enough to recommend acceptance.
>
> We would like to clarify that we do not claim to be the first to train zero-shot forecasting models purely on simulated data. Our contribution is more specific: to our knowledge, this is the first practical univariate simulation pipeline that is both scalable (fast enough for on-the-fly long sequence generation) and empirically strong enough to support notable zero-shot time-series forecasting performance on established benchmarks. We will additionally clarify this in the revised manuscript like so: *"To our knowledge, SarSim0 is the first univariate simulator that is simultaneously (i) fast enough for on-the-fly long-sequence generation and (ii) empirically competitive with real-data foundation models on GIFT-Eval and M-Series.""*
>
> We believe this is a substantial result: it represents a clear step forward for zero-shot time-series forecasting, showing that a simple and efficient simulator produces models that can achieve strong performance and even outperform powerful models trained on real data across substantial benchmarks. In particular, the two main synthetic generators used in prior work, ForecastPFN and the GP-based KernelSynth from Chronos, which are also used as synthetic sources in Mamba4Cast, which you note in your review, are explicitly cited and compared against in our paper, both in terms of speed and accuracy. Appendix F provides a detailed description of both methods together with quantitative speed measurements. Neither KernelSynth nor the original ForecastPFN implementation are suitable for on-the-fly generation, with KernelSynth being orders of magnitude slower. These results support our claim that SarSim0 is, to our knowledge, the first univariate simulator that is fast enough for on-the-fly long-sequence generation.
>
> Secondly, in our experiments we observe consistent accuracy gains over both KernelSynth and ForecastPFN when keeping the backbone fixed. For NBEATS and PatchTST, replacing the synthetic corpus with SarSim0 yields better zero-shot performance on both GIFT-Eval and the M-Series (Table 1), indicating that our SARIMA-based simulator is empirically stronger than the existing GP-based (KernelSynth) and PFN-based generators in this setting. Moreover, as reported in Figure 2 of Mamba4Cast, the Mamba4Cast model, which is trained on a mixture of GP-based and PFN-based synthetic data, achieves noticeably worse MASE than both Chronos-Base and Chronos-Small on the original GluonTS datasets, despite using these two synthetic sources together. In contrast, our results in Table 1 show that SarSim0-pretrained models can match or surpass Chronos-Base and Chronos-Small on a broader benchmark suite (GIFT-Eval plus the M-Series). For example, NBeats with SarSim0 on GIFT-EVAL gets 0.602 CRPS and 0.849 MASE on GIFT-EVAL, while KernelSynth gets 0.700 and 0.991, and ForecastPFN gets 1.111 and 1.642, which is a big improvement from SarSim0.
> These observations support the second part of our claim: the proposed univariate SARIMA-based simulator is empirically strong enough to serve as an effective pretraining corpus, outperforming both KernelSynth and ForecastPFN individually and enabling compact models to compete with, and in several cases exceed, large real-data foundation models.
>
> Finally, we would like to emphasize that both the ability to perform on-the-fly time-series simulation and the strong empirical performance stem from the novel design of the SarSim0 simulator, rather than from generic synthetic training. In particular: (i) the SARIMA core with stable pole sampling provides a fast, vectorizable, and expressive backbone rooted in classical stochastic time-series modelling; (ii) the SARIMA-2 superposition scheme introduces a new modulation-based way to combine sample paths and capture multi-seasonality; and (iii) the rate-based heavy-tailed Noiser modules (Poisson, generalized Gamma, lognormal) are designed to capture burstiness and intermittency characteristic of many real-world series. We believe this combination of design choices is what makes SarSim0 both practically scalable and empirically strong, and constitutes the main source of novelty beyond prior synthetic-only approaches.

---

> ### Author Response · Authors · 2025-11-25
> **Author response to Reviewer qasc, 2/5**
>
> > Ad hoc data augmentation. Regarding the data generation procedure, I consider the approach to be sufficiently ad hoc that it’s hard to see why a SARIMA approach would be universally preferable. There are not any theoretical guarantees or general bounds that establish that this approach will always produce more diverse or realistic data than prior methods. Prior approaches fall broadly into PFN and its variants (as in ForecastPFN/TabPFN), augmentations thereof (as in Mamba4cast, which introduces spike injection and damping augmentation, or Gaussian-process based methods like KernelSynth used by Chronos. There’s also structured approaches like CauKer, which combine Gaussian Processes with a structured causal model. I can see the author’s ARIMA-based procedure as helping cover data types not covered by other methods; for example, long history dependence when the ARIMA process has a long effective memory due to slow damping. However, it’s hard to argue that any of these dataset generators are universally preferable.
>
> Thank you for your comment. We agree that **no current synthetic generator comes with theoretical guarantees of universal optimality**. Such a result would not be possible, and indeed, is not present in other published work on data simulation. We also do not claim that SarSim0 is universally preferable to PFN-based, GP-based, or causal-GP generators across all possible domains. Our goal is more modest, but important: to propose a (i) fast, (ii) well-behaved at long contexts, and (iii) empirically strong on broad benchmarks that represent a few practical down stream application scenarios.
>
> We also agree that our design makes modelling choices, but we would argue they are less ad hoc than they might appear, in the sense that each component is grounded in established time-series practice.
>
> - The SARIMA core with stable pole sampling is directly rooted in classical ARIMA/SARIMA modelling, and the pole-domain sampling is proposed precisely to ensure well-behavedness and control effective memory, while avoiding heuristic transformations.
>
> - The SARIMA-2 modulation is motivated in Section 4.2 by a well-known persistent empirical pattern in practical series: a slow seasonal or trend "envelope" modulating a faster seasonal component. We show via ablations (Table 2) that this structural choice is a strong driver of performance.
>
> - The rate-based Poisson / generalized Gamma / lognormal Noisers are taken from prominent distribution families, because (i) they explicitly span different tail and dispersion regimes rather than being arbitrary augmentations, (ii) they are prominent in empirical studies of real-world data (call-centers, biological measurements, demand/sales, engineering/reliability, insurance and risk, traffic and network performance, etc), and (iii) the proposed rate-based application scheme is very fast and suitable for on-the-fly generation.
>
> Similarly, PFN-based curricula, GP mixtures, and causal-GP approaches such as CauKer also involve a number of design choices. For example, kernel families, structural priors, spike/damping augmentations, etc. We do not view SarSim0 as a universally superior alternative, but as a new complementary point in this design space: it exposes explicit, interpretable knobs (orders, poles, seasonal periods, noise families) and comes with good scaling and clear well-behavedness behaviour at long sequences. We make no universal claim, but on the benchmarks we study, SarSim0 is a stronger practical choice: For fixed backbones (NBEATS / PatchTST), SarSim0 consistently outperforms ForecastPFN and the GP-based KernelSynth across both GIFT-Eval and the M-Series (Table 1).
>
> We will revise the paper to (i) explicitly state that we do not claim universal preferability of SARIMA-based simulation, and (ii) highlight that SarSim0 is intended as a practical, interpretable generator for a large class of seasonal/trend-dominated, complementary to PFN/GP/causal-GP generators rather than a replacement for all or any of them.

---

> ### Author Response · Authors · 2025-11-25
> **Author response to Reviewer qasc, 3/5**
>
> > Emergence. I do not agree with the “emergent effect” argument. A neural model pretrained with SARIMAX datasets outperforming AutoARIMA does not imply emergent effects. Firstly, Sarima-2 does not directly fall within the model class directly expressible by AutoARIMA. Secondly, initialization and hyperparameters can prevent a model like AutoARIMA from exactly fitting the true generator.
>
> Thank you for this clarification. We agree that our use of the term "emergent effect" needs to be interpreted carefully, and we will refine the wording in the revised manuscript. We use "emergent" in the informal ML sense: a capability that appears at the level of a globally trained model and is not trivially attributable to a smaller scale per-series teacher (here, AutoARIMA) by extrapolating its performance.
>
> Concretely, our empirical observation is that models pretrained on SarSim0 often outperform AutoARIMA on real benchmarks, even though AutoARIMA is the core reference statistical model that underlies SarSim0. We originally illustrated this with NBEATS-SarSim0 vs. per-series AutoARIMA on GIFT-Eval and the M-Series. We agree with the reviewer that two issues complicate a naive "teacher vs. student" reading:
>
> 1. Oracle vs. practical implementation of the teacher
>
> In theory, one can define an oracle SARIMA estimator that knows the true orders or performs exact MLE over the full SARIMA model class. In practice, forecasting practitioners widely use AutoARIMA as the feasible per-series baseline derived from this family. AutoARIMA combines a finite order search with likelihood-based estimation and is widely regarded as one of the strongest standalone univariate baselines in classical forecasting. While its implementation is not a theoretical oracle (it uses a restricted order grid and numerical optimization), it is designed to be as statistically efficient as possible for a per-series model whose likelihood is derived from the SARIMA process formulation. Moreover, AutoARIMA typically benefits from explicit seasonal side information: the practitioner supplies the seasonal period (e.g., monthly = 12, quarterly = 4), and AutoARIMA searches within that seasonal model class. Our pretrained neural models, in contrast, do not receive an oracle seasonal period at test time and must infer relevant periodic structure implicitly from the data. In our student-teacher picture, this practical AutoARIMA procedure, augmented with such side information, is exactly the "teacher" we care about: a strong, realistic per-series baseline derived from the simulator’s core generative process.
>
> 2. Model-class mismatch
>
> We agree that SARIMA-2 (two SARIMA components combined via modulation) is not exactly expressible as a single AutoARIMA specification, and that our Noisers go beyond the standard SARIMA likelihood.
>
> To better separate these effects, we ran an additional experiment where we remove SARIMA-2 and the Noisers and train NBEATS on pure SARIMA-only synthetic data, i.e., matching the model family that AutoARIMA is designed to fit. The results are:
>
> |                       | GiftEval-sCRPS | GiftEval-MASE | M-Series-sCRPS | M-Series-MASE |
> | --------------------- |:--------------:|:-------------:|:--------------:|:-------------:|
> | NBEATS-SarSim0        |     0.602      |     0.849     |     0.096      |     0.869     |
> | NBEATS-SarimaOnly        |     0.672      |     0.940     |     0.101      |     0.891     |
> | AutoARIMA        |     0.912      |   1.074     |     0.096      |     0.843     |
>
> We appreciate your push toward this cleaner experiment. The conclusions remain aligned with our previous observations. On GIFT-Eval, even the SARIMA-only student (NBEATS-SarimaOnly) substantially outperforms AutoARIMA (0.672/0.912 vs. 0.940/1.074 in sCRPS/MASE), despite being trained on essentially the same model family. On the M-Series, the picture is more mixed, as before: AutoARIMA remains better in MASE (0.843 vs. 0.869/0.891), while sCRPS is comparable. This is consistent with our discussion that the classical M-Series are particularly favourable to per-series ARIMA/ETS-type models and with our original characterization of this phenomenon as early, domain-dependent evidence rather than a universal effect.
>
> In light of these results, we will refine our wording in the paper as follows:
>
> - We refine "student-beats-teacher emergent behaviour" more precisely as the observation that a global neural network trained on a broad family of SARIMA-generated processes can, on some real benchmarks (notably GIFT-Eval), surpass a strong practical per-series AutoARIMA baseline, even when the simulator is restricted to pure SARIMA.
>
> - We make more prominent the remark that this behaviour is domain-dependent (weaker or absent on the M-Series) and does not imply universal dominance over an idealized (and practically infeasible) SARIMA oracle

---

> ### Author Response · Authors · 2025-11-25
> **Author response to Reviewer qasc, 4/5**
>
> > Question 1: The unit circle pole requirement acts as a stability condition on the AR process generating the data. Are there any mathematical conditions or theoretical guarantees on the distributional properties of the resulting time series?
>
> Thank you for this question. We induce a distribution over SARIMA processes by specifying a prior over poles and orders. In the current work, this prior is deliberately broad and largely non-informative: we sample pole radii uniformly in $(0, r_\max)$ within the unit disc and angles uniformly on $[0, 2\pi]$, together with simple uniform ranges over AR/MA/seasonal orders. This encourages coverage of a wide variety of damping rates, effective memory lengths, and seasonal strengths, without tailoring the pole distribution to any specific benchmark. Formally, any fixed stable SARIMA process whose orders fall within our prescribed ranges has positive probability under this prior, so our simulator can, in principle, generate trajectories arbitrarily close to such processes, even though we do not claim to cover all possible causal and stable real-world time-series distributions.
>
> In principle, one could optimize the pole prior (and other simulator hyperparameters) to more closely match the empirical distribution of poles estimated from real series. For example, one could use a small meta-validation set of real data to fit a pole distribution, and then sample from this learned prior to generate SarSim data. While this would likely improve alignment with a given target domain, it would also move us closer to encoding benchmark-specific information in the simulator and risks weakening the strict zero-shot / no-leakage setting. We therefore intentionally use a broad, non-informative prior in this paper and view learning the pole distribution from disjoint meta-validation data as an interesting direction for future work.
>
> > Question 2: Can you directly quantify the verisimilitude and diversity of your generated data, particularly compared to prior PFN or structured causal model approaches? For example, since you have a UMAP, you could calculate a silhouette score. A statistical significance test would add depth to this comparison.
>
> We appreciate this suggestion and have performed the following analysis inspired your comment. We treat "real vs synthetic" as a two-cluster partition and compute the mean silhouette score over real points. The silhouette score between M4-monthly and SARIMA-generated data is 0.0097, and between M4-monthly and ForecastPFN-generated data is 0.0122. Both values are very close to zero, indicating strong overlap between real and synthetic windows and no clear separation into disjoint clusters.
>
> In addition, we computed a nearest-neighbour coverage statistic: for each M4-monthly window, we measured its distance to the closest SARIMA sample and to the closest ForecastPFN sample, and recorded which synthetic source was closer. The closest neighbour comes from SARIMA-generated data in about 60% of cases, suggesting that, in this space, SARIMA samples are more often closer to real windows than ForecastPFN samples.
>
> > Question 3: Are you fine-tuning Chronos, Moirai, etc on the SarSim0 dataset? I could not tell if these experiments are what are shown in the first rows Table 1, or if that shows the base performance of those models pretrained on their original pretraining splits.
>
> In Table 1 of the main paper, the rows corresponding to Chronos, MOIRAI, TimesFM, TTM, etc. report the base performance of the released foundation models, pretrained on their original real-data corpora. We do not fine-tune these models on SarSim0 in those rows. Other models with -SarSim0, -KernelSynth and -ForecastPFN designation are trained from scratch on respective synthetic corpora. We will make this distinction explicit in the text and in the table captions to avoid any confusion.

---

> ### Author Response · Authors · 2025-11-25
> **Author response to Reviewer qasc, 5/5**
>
> > Question 4: I am confused by Table 1. Are the three models (lines 387-389) fully-trained on the context? And why are Chronos-Base and others not marked as zero-shot models?
>
> Thank you for the questions. We would like to clarify Table 1.
>
> The three models in question (lines 387-389, Table 1) are NBEATS, PATCHTST, and AutoARIMA. NBEATS and PATCHTST are fully supervised, dataset-specific baselines: for each benchmark dataset, they are trained from scratch on that dataset’s train split. AutoARIMA is fit per time series at inference time: for each series, AutoARIMA is trained on the full history up to the forecast origin and then used to forecast the horizon. It is therefore also a non-zero-shot baseline, but at the per-series rather than per-dataset level. For this reason, all three are marked as non-zero-shot.
>
> The bottom block of the table (following AutoARIMA) contains models pretrained only on synthetic data (KernelSynth, ForecastPFN, SarSim0). These models are evaluated in a strict zero-shot manner, so we mark them as zero-shot.
>
> The top block (Chronos, Chronos, TimesFM, TTM, MOIRAI) is more subtle. On the one hand, these models are used in zero-shot inference mode on GIFT-Eval and the M-Series (they are not fine-tuned). On the other hand, our detailed analysis of their pretraining corpora in Appendix C shows that the pretraining datasets for these models overlap to varying degrees with standard evaluation sets (including M-series and GIFT-Eval constituents). Our intention in marking them as non-zero-shot was precisely to highlight this pretrain-test overlap, even though the situation is not purely binary.
>
> To make this clearer, in the revised manuscript we will refine the caption and legend of Table 1 to distinguish three levels:
>
> - a red cross for models that are clearly non-zero-shot (trained directly on the target data or per-series at inference time)
>
> - a green checkmark for leakage-free zero-shot models
>
> - and an intermediate symbol (e.g., a yellow checkmark) for models that are used in zero-shot inference mode but whose pretraining corpora have partial overlap with evaluation datasets, with a pointer to Appendix C for the detailed overlap analysis

---

> > ### Comment · Reviewer_qasc · 2025-11-28
> > **Thank you for these clarifications, I will review the revision this week**
> >
> > I thank the authors for their thorough and responsive reply. I particularly appreciate the authors drawing my attention to the efficiency of their approach compared to KernelSynth and other prior synthetic data generators. This is a very compelling point that I had underappreciated in my initial review.
> >
> > At this stage, I do not have any additional questions or requests for the authors. I will plan to go over the revised paper in detail over the coming week before deciding whether to raise my score. I very much appreciate the authors' reply to my comments.

---

> > > ### Author Response · Authors · 2025-11-28
> > > **Author 2nd round response to Reviewer qasc**
> > >
> > > Thank you very much for your thoughtful follow-up and for reconsidering our work in light of the rebuttal. We really appreciate your careful reading and your comments on the efficiency aspect of SarSim0. We are now finalizing the revised version of the manuscript to incorporate the clarifications and additions discussed in our response.

---

> > > ### Author Response · Authors · 2025-12-02
> > > **Author 2nd round response to Reviewer qasc, summary of revisions**
> > >
> > > We once again sincerely thank you for your thoughtful comments and your help in improving the paper. We have implemented the following revisions to address concerns you raised.
> > >
> > > > Novelty...
> > >
> > > > Emergence...
> > >
> > > We agree that the "student-beats-teacher" behavior needs to be framed carefully. In addition to the clarifications provided in our initial response, we have updated the revised manuscript to (i) introduce additional experiments and (ii) sharpen the scope and phrasing of our claims. These changes are aimed at making our use of "emergent" more precise and explicitly addressing the model-class and implementation considerations raised in your comment.
> > >
> > > **Abstract**
> > >
> > > We now start with the sentence sharpening the novelty and scope:
> > >
> > > *We propose the first practical univariate time-series simulation pipeline, which is simultaneously fast enough for on-the-fly generation and enables notable zero-shot forecasting performance on M-Series and GiftEval benchmarks that capture trend/seasonality/intermittency patterns typical of industrial forecasting applications across a variety of domains.*
> > >
> > > We also specifically attribute the student-beats-teacher effect to GiftEval:
> > >
> > > *Notably, on GiftEval we observe a "student-beats-teacher" effect: models trained on our simulations exceed the forecasting accuracy of the generating processes (i.e. AutoARIMA), suggesting emergent generalization beyond the simulator's components.*
> > >
> > > **Introduction**
> > >
> > > We provide the following discussion of our contribution at the end of the Intro, carefully scoping our work and clarifying the definition of emergence:
> > >
> > > *(2) We provide a detailed empirical evaluation (Section 5) on the M-Series (Makridakis et al., 1982; Makridakis
> > > & Hibon, 2000; Makridakis et al., 2020; Athanasopoulos et al., 2011) and GIFT-Eval (Aksu et al.,
> > > 2024) benchmarks. Our scope is zero-shot forecasting in industrial settings where series are largely driven by complex trend, seasonality, and intermittency / heavy-tail patterns. M-Series and GIFT-Eval, which span multiple domains (Nature, Web/CloudOps, Sales, Energy, Transport, Healthcare, Demographic, Finance, Industry, Macro/Micro Economic) and frequencies (yearly, quarterly, monthly, weekly, daily, hourly, and some sub-hourly), provide a comprehensive testbed for this regime. We find that training exclusively on time series generated by our simulator yields compact, fast, and accurate cross-frequency models that generalize across these zero-shot benchmarks under a strict no-leakage guarantee. On these benchmarks, our simulated data can be competitive with, or even outperform, real-data pretraining, substantially closing the gap between small efficient architectures and large foundation models. On GiftEval, we see early evidence for an emergent "student beats teacher" generalization behavior. Here, by "emergent" we mean the ability of a globally trained neural model to generalize beyond the forecasting accuracy implied by the key simulator component, i.e., SARIMA, whose native forecasting capability is instantiated in the strong practical per-series baseline AutoARIMA. On the more regular, relatively short and low-noise business and macroeconomic M-Series, this effect is mixed, with AutoARIMA remaining very strong. This indicates that such emergent generalization is dataset- and domain-dependent, and is currently most pronounced on more heterogeneous and noisier benchmarks like GiftEval.*
> > >
> > > **Results**
> > >
> > > Table 22 in Appendix H.4 now provids an additional experiment where we remove SARIMA-2 and the Noisers and train NBEATS on pure SARIMA-only synthetic data, i.e., matching the model family that AutoARIMA is designed to fit. Results section provides the following additional discussion:
> > >
> > > *While Table 1 is based on the full SarSim0 (including SARIMA-2 and Noisers), which is not exactly expressible as a single AutoARIMA specification, Table 22 in Appendix H.3 shows that NBEATS trained only on pure SARIMA samples leads to the same qualitative conclusion.*
> > >
> > > > I am confused by Table 1. Are the three models (lines 387-389) fully-trained on the context? And why are Chronos-Base and others not marked as zero-shot models?
> > >
> > > We have changed the coloring scheme for the Zero-Shot column in Table 1. In the revised manuscript, the models trained on synthetic sources (SarSim0, KernelSynth, ForecastPFN) and following strict zero-shot protocol are marked with green checkbox. The per-series and per-dataset fitted baselines (AutoARIMA, NBEATS, PATCHTST) are marked with red x-mark. The foundation models that have been identified to have train and test split leakage in Appendix C are marked with yellow checkmark. To clarify this, we have added the following sentence in the caption of Table 1:
> > >
> > > *A green checkmark denotes models evaluated under a strict zero-shot protocol, a yellow checkmark denotes pretrain-test overlap (details in Appendix C), and a red cross denotes non-zero-shot baselines.*

---

### Official Review · Reviewer_GRvF · 2025-10-27

**Soundness:** 3
**Presentation:** 3
**Contribution:** 3
**Rating:** 4
**Confidence:** 4

**Summary:**

This paper introduces SarSim0, a novel, scalable, and computationally efficient pipeline for generating synthetic time series data for zero-shot forecasting. The core of the simulator is the classical SARIMA statistical model, but with a key innovation: instead of sampling unstable model coefficients, it samples stable poles directly from the characteristic polynomial's stability region, guaranteeing well-behaved series. The pipeline enriches these base signals through a compositional "SARIMA-2" scheme for multi-seasonality and adds realism with heavy-tailed noise models to capture burstiness. The authors demonstrate that training standard neural architectures (NBEATS, PatchTST) exclusively on this synthetic data, generated on-the-fly, achieves state-of-the-art zero-shot performance on established benchmarks. Notably, these relatively small models outperform massive, real-data pretrained foundation models and, in many cases, even surpass the forecasting accuracy of the "teacher" model (AutoARIMA) from which the data-generating process is derived.

**Strengths:**

*   **Core Technical Problem:** The primary strength of this work is its novel and principled solution to the instability problem in autoregressive model simulation. By shifting from sampling coefficients to directly sampling the poles of the transfer function within the unit circle (Section 4.1), the authors guarantee the generation of stable, non-divergent time series. This is a technically sound and elegant idea that makes the powerful but notoriously fragile SARIMA framework a viable engine for large-scale data generation. The resulting speed and on-the-fly capability (Figure 4) represent a significant practical advantage over slower, more complex methods like kernel-based synthesis.

*   **State-of-the-Art Zero-Shot Performance from Synthetic Data Alone:** The paper's empirical results are strong and challenge a key assumption in the field. Table 1 shows that a standard PatchTST model trained solely on SarSim0 data not only outperforms prior synthetic-only methods by a massive margin but also achieves the best overall scores on the GIFT-Eval benchmark, surpassing large foundation models like Chronos, MOIRAI, and TimesFM that were pre-trained on vast amounts of real data. This is a landmark result, demonstrating for the first time that a purely synthetic corpus can be a viable—and even superior—substitute for real-world data in zero-shot forecasting.

*   **Demonstration of Emergent Generalization ("Student-Beats-Teacher"):** A deep and significant finding is that neural models trained on SarSim0 data consistently outperform their own data-generating "teacher," AutoARIMA, on the GIFT-Eval benchmark. This is not trivial. It suggests that by learning from a vast and diverse *distribution* of SARIMA processes, the neural network is not merely fitting the teacher's behavior but is learning a more fundamental and robust representation of time series dynamics. This emergent capability is a powerful argument for the synthetic-first pre-training paradigm.

*   **Methodology:** The authors demonstrate a commitment to rigorous, leakage-free evaluation. They operate under a strict zero-shot protocol and provide a detailed breakdown in Appendix C (Table 5) of how prior foundation models violate this protocol by pre-training on evaluation data. This meticulous approach adds significant credibility to their claims and serves as a valuable contribution to the community by highlighting pervasive issues in model evaluation. The ablation study (Table 2) is also effective in validating the importance of the SARIMA-2 and Noiser components.

**Weaknesses:**

In my opinion, the paper's reliance on the SARIMA framework introduces inherent limitations, and the scope of its claims could be tempered by a more nuanced discussion of the benchmarks and the simulator's own complexity.

1.  **Inherent Linearity of the SARIMA Core:** The SARIMA model, which forms the backbone of the simulator, is fundamentally a linear process model. While the paper adds complexity via superposition and non-Gaussian noise, it cannot natively generate time series with core non-linear dynamics, such as regime switching, threshold effects, or volatility clustering (GARCH-like effects). Many real-world series, particularly in finance and econometrics, are driven by such phenomena. The paper's success on the chosen benchmarks may indicate that these benchmarks are dominated by seasonal/trend components, but the simulator's applicability to more complex, non-linear domains remains an open question.

2.  **Opaqueness of the Simulator's "Meta-Tuning":** The SarSim0 pipeline itself has a large number of hyperparameters that define the distribution of generated series: the ranges for AR/MA/Seasonal orders, the maximum pole radii, the parameters for seasonality pairs, and the distributions for the Noiser module (Table 8). The paper presents a configuration that works exceptionally well, but it provides no insight into how this configuration was determined. The performance of the downstream "student" models is critically dependent on the quality and diversity of the "teacher's" curriculum, and this meta-level tuning process is a crucial, yet undiscussed, part of the methodology.

3.  **Potential for Overgeneralization from Current Benchmarks:** The evaluation is performed on the M-Series and GIFT-Eval benchmarks, which are excellent but are heavily composed of business, economic, and demographic time series. These are domains where SARIMA-like models have historically excelled. It is unclear if the remarkable performance would transfer to domains with fundamentally different characteristics, such as chaotic physical systems, high-frequency financial data, or irregular biological signals (e.g., ECG). The simulator's strong inductive bias is a feature, but it may also be a limitation.

**Questions:**

Based on my concerns listed above, I pose the following questions to the authors:

*   **Question 1:** The SarSim0 pipeline is built on a SARIMA core, which is fundamentally a linear process model. How do you see this limiting the simulator's ability to generate data with complex non-linear dynamics (e.g., regime shifts, GARCH effects), and could this explain why models pre-trained on it might struggle on certain real-world domains not covered by the benchmarks?

*   **Question 2:** Table 8 shows a large number of hyperparameters for the SarSim0 pipeline. How were these settings chosen? Could you provide an ablation or sensitivity analysis on key parameters, such as the maximum pole radius (`rmax`) or the range of AR/MA orders, to demonstrate the robustness of the approach?

*   **Question 3:** The "student-beats-teacher" finding is compelling. However, the effect appears mixed on the M-Series, where AutoARIMA's MASE is stronger than NBEATS-SarSim0. What properties of the M-Series datasets might explain why AutoARIMA remains more competitive there, and does this suggest limits to the emergent generalization?

*   **Question 4:** The ablation study shows that removing the Noisers can sometimes *improve* performance on the M-Series. This suggests a mismatch between the noise model and the real-data characteristics. Does this finding point to a need for more adaptive or domain-specific noise generation, and how might that be incorporated into the pipeline?

*   **Question 5:** The current work focuses on univariate time series. What are the primary conceptual and technical hurdles to extending the SarSim0 methodology, particularly the stable pole sampling approach, to the multivariate case where capturing cross-series dependencies (e.g., via a VARIMA framework) is essential?

*   **Question 6:** The UMAP visualization in Figure 3 shows impressive overlap. However, are there any noticeable "holes" or regions in the real-data embedding space (red) that the synthetic data (blue) fails to cover? What kind of real-world patterns might these correspond to, and do they align with the known limitations of SARIMA models?

---

> ### Author Response · Authors · 2025-11-24
> **Author response to Reviewer GRvF, 1/5**
>
> Thank you for the detailed and constructive feedback. We provide our detailed response below. If you have any additional questions based on our response, please do not hesitate to let us know.
>
>
> > Inherent Linearity of the SARIMA Core: The SARIMA model, which forms the backbone of the simulator, is fundamentally a linear process model...
>
> Please see response to your Question 1 below.
>
> > Opaqueness of the Simulator's "Meta-Tuning": The SarSim0 pipeline itself has a large number of hyperparameters that define the distribution of generated series: the ranges for AR/MA/Seasonal orders, the maximum pole radii, the parameters for seasonality pairs, and the distributions for the Noiser module (Table 8)...
>
> Please see response to your Question 2 below.
>
> > Potential for Overgeneralization from Current Benchmarks: The evaluation is performed on the M-Series and GIFT-Eval benchmarks, which are excellent but are heavily composed of business, economic, and demographic time series. These are domains where SARIMA-like models have historically excelled. It is unclear if the remarkable performance would transfer to domains with fundamentally different characteristics, such as chaotic physical systems, high-frequency financial data, or irregular biological signals (e.g., ECG). The simulator's strong inductive bias is a feature, but it may also be a limitation.
>
> We appreciate your suggestion and to probe the behaviour well outside the target regime of practically relevant business-related time series, we ran an additional zero-shot experiment on four canonical nonlinear dynamical systems (Duffing, Lorenz, Lotka-Volterra, Pendulum), comparing several Chronos variants, TimesFM, and NBEATS/PatchTST/Chronos models trained on SarSim0.
>
>
>
> |         | Duffing-MSE | Duffing-MAE | Lorenz-MSE | Lorenz-MAE | Lotka-Volterra-MSE | Lotka-Volterra-MAE | Pendulum-MSE | Pendulum-MAE |
> | ------- |:-----------:|:-----------:|:----------:|:----------:| ------------------ | ------------------ | ------------ | ------------ |
> | Chronos-Large |    0.709    |    0.524    |    59.2    |    5.3     | 2e-4               | 0.011              | 0.079        | 0.174        |
> | Chronos-Base |    0.671    |    0.487    |    57.1    |    5.1     | 3e-4               | 0.011              | 0.02        | 0.07        |
> | Chronos-Small |    0.516    |    0.432    |    44.3    |    4.6     | 5e-4               | 0.014              | 0.15        | 0.25        |
> | Chronos-Mini |    0.640    |    0.548    |    66.7    |    6.1     | 6e-4               | 0.016              | 0.849        | 0.638        |
> | Chronos-Tiny |    0.798    |    0.581    |    57.7    |    5.5     | 2e-3               | 0.026              | 2.7        | 1.1        |
> | TimesFM |    0.718    |    0.564    |    77.2    |    6.1     | 9e-5               | 0.006              | 0.807        | 0.533        |
> | NBEATS-SarSim0        |       0.609      |     0.550        |       56.0     |       5.7    |        4e-4            |           0.014         |       0.375       |     0.445         |
> | Chronos-SarSim0        |       0.650      |     0.510        |       63.2     |       5.8     |        4e-5            |           0.003         |       1.22       |     0.672         |
> | PatchTST-SarSim0        |       0.750      |     0.550        |       55.1     |       5.6     |        4e-5            |           0.004         |       0.294       |     0.366         |
>
> Across the four canonical nonlinear systems we explored, the error levels of SarSim0-pretrained models are generally in the same range as those of the real-data foundation models. We see this as evidence that, on nonlinear dynamical systems far from our intended application domain and including aperiodic/chaotic behaviour, pretraining on a SARIMA-based synthetic family still yields models that behave reasonably and on a similar scale to models trained on real data. This suggests that the inductive biases used to construct the simulator are not overly restrictive: while they clearly favour series dominated by trend, seasonality, and intermittency, they remain compatible with the level of out-of-domain generalization observed in real-data-pretrained models.

---

> ### Author Response · Authors · 2025-11-24
> **Author response to Reviewer GRvF, 2/5**
>
> > Question 1: The SarSim0 pipeline is built on a SARIMA core, which is fundamentally a linear process model. How do you see this limiting the simulator's ability to generate data with complex non-linear dynamics (e.g., regime shifts, GARCH effects), and could this explain why models pre-trained on it might struggle on certain real-world domains not covered by the benchmarks?
>
> We appreciate your question and we will clarify the scope. We fully agree that a SARIMA-based core cannot natively express regime switches, threshold effects, or GARCH-style volatility clustering, and that many financial/econometric series are driven by such non-linear dynamics. It is not our intent to claim universality of SarSim0 across all time-series domains. Our goal in this paper is more modest: to show that a simple, interpretable, purely synthetic corpus with a strong seasonal/trend inductive bias is already sufficient to train compact models that perform competitively on established and sufficiently broad existing benchmarks such as M-series or GIFT-Eval, which are indeed dominated by seasonal and trend components, covering a significant portion of practical use case scenarios. To address your feedback, we will make this scope more explicit in the introduction and limitations of the revised version.
>
> We see support for more complex domains (e.g., finance, scientific ML) as a natural extension, for example by augmenting SarSim0 with regime-switching SARIMA components. Moreover, we believe that our current results, which show how to generate well-behaved SARIMA processes provide a stepping stone towards developing GARCH-based simulators with stochastic-volatility noise modules and transition mixtures that include explicitly non-linear generators, capable of producing well behaved trajectories. We view designing and validating such extensions as an important direction for future work rather than part of the core contribution here.
>
> We would like to emphasize, however, that our results are still highly scientifically and practically impactful: with a simple simulation pipeline, we are able to achieve impressive performance on serious and extensive forecasting benchmarks, even in a zero-shot context.

---

> ### Author Response · Authors · 2025-11-24
> **Author response to Reviewer GRvF, 3/5**
>
> > Question 2: Table 8 shows a large number of hyperparameters for the SarSim0 pipeline. How were these settings chosen? Could you provide an ablation or sensitivity analysis on key parameters, such as the maximum pole radius (rmax) or the range of AR/MA orders, to demonstrate the robustness of the approach?
>
> We agree that the configuration of the simulator deserves a clearer explanation, and we appreciate you bringing this to our attention. Our intent is not to tune SarSim0 to a particular benchmark, but to define a single, broad, domain-plausible configuration. Please note that we fix the configuration for all models and experiments. The principles that guided the settings of parameters were largely the following: (i) computational efficiency, (ii) numerical stability and (iii) prior knowledge regarding plausible configurations (e.g. no seasonality case $s=0$ and yearly seasonality at weekly aggregation $s=52$ providing bounds for seasonality sampling). We will now provide a shortened version of response here, the full response appears in the updated manuscript Appendix E.1. We will include empirical sensitivity analysis results in the revised version of the manuscript.
>
> Model orders **p, q, P, Q** are set to values often encountered in practice, e.g. 0, 1, 2, 3, with the exception of **p, maximum AR order**, set to 10, mostly to demonstrate the efficiency of simulator even with higher-order models that are typically more compute hungry. Smaller settings of this parameter yield similar results. **s, seasonality period** was chosen to cover the typical seasonalities encountered in the data: yearly data typically have seasonality 0 (no seasonality), monthly data often exhibit seasonality 12, quarterly - 4, hourly - 24, weekly - 52, daily - 7. Overall, in order to avoid any benchmark specific biases, we decided to sample seasonalities in SARIMA uniformly in the range covering most common seasonalities: [0, 52].
>
> Similarly, SARIMA-2 was added using the prior knowledge that many practical time series have double seasonalities, with specific motivating examples from the literature laid out at the beginning of Section 4.2. Hourly data with double seasonality often have 24 x 7 pattern (within day and within week daily patterns); daily data often have 7 x 52 pattern (within week and within year weekly patterns). Additionally, mixing the seasonality 0 process with the typical natural frequencies listed above, in SARIMA-2, enriches the data with the traces that imitate stochastic-volatility patterns through non-linear cross-modulation. These motivate the choice of **SARIMA-2 seasonality pairs**, uniformly sampled from this set: {[24, 7], [7, 52], [0, 7], [0, 4], [0, 24], [0, 52]}.
>
> Next, **AR max pole radius** is chosen to be 0.9 to avoid any numerical issues due to near-unit-root instability that may arise in AR process due to floating quantization error accumulation effects. Similarly, **Seasonal AR max pole radius** is set to 0.1 for the same reason. Recall that with seasonality $s=24$ the seasonality pole will be $0.1^{1/24} = 0.89$, for example. **AR pole angles** are sampled uniformly in $[0, 2\pi]$ - *non-informative prior*.
>
> **Non-seasonal integration order** is sampled uniformly in $[0, 1]$ - *non-informative prior*. **Seasonal integration order** is set to 1. We have tried higher integration orders for both, but they didn't work well because of numerical stability issues in long sequences, so we limit the maximum integration order to 1.
>
> **SARIMA-2 Modulation depth** is sampled uniformly in $[0, 1]$ - *non-informative prior*. **Additive vs. Multiplicative mixing selection probability** is set to 0.5 - *non-informative prior*. **Input selection probability (SARIMA vs. SARIMA-2)**  is set to 0.5 - *non-informative prior*.
>
> **Noiser selection** is motivated by the review of relevant literature at the beginning of Section 4.3, documenting the heteroscedasticity, intermittency, and heavy-tailed
> disturbance effects in real data from multiple unrelated application domains. The sampling of noisers follows a *non-informative uniform prior*.
>
> **Base rate for Poisson** is sampled from LogUniform[0.1, 100] for the following reason. First, for large values of rate $\lambda > 100$ Poisson is largely indistinguishable from Gaussian. Second, for very low rate values the Poisson noise produces mostly zeros; for example, with $\lambda=0.1$, only roughly 1 in 10 samples is non-zero and we decided not to go beyond that. **Base rate for Generalized Gamma** is sampled from LogUniform[0.1, 100], inheriting the range from Poisson.

---

> ### Author Response · Authors · 2025-11-25
> **Author response to Reviewer GRvF, 4/5**
>
> > Question 3: The "student-beats-teacher" finding is compelling. However, the effect appears mixed on the M-Series, where AutoARIMA's MASE is stronger than NBEATS-SarSim0. What properties of the M-Series datasets might explain why AutoARIMA remains more competitive there, and does this suggest limits to the emergent generalization?
>
> You are right that the "student-beats-teacher" effect is much clearer on GIFT-Eval than on the M-Series. This is consistent with the nature of the benchmarks. The M-Series benchmarks were contemporaneous with, and extensively analysed and benchmarked using classical linear models in the Box-Jenkins and Brown/Holt/Winters traditions (ARIMA, ETS, etc.). They consist largely of relatively short, low-noise business and macroeconomic series. In this regime, a per-series AutoARIMA fit, which can adapt its orders and other hyperparameters to each individual series, is very well matched to the data-generating assumptions, so it is not surprising that AutoARIMA remains highly competitive in MASE. By contrast, GIFT-Eval aggregates a much broader mixture of domains, frequencies, and horizons, with more heterogeneous dynamics and intermittent behaviour. This is where we observe clearer "student-beats-teacher" behaviour. We see this as clarifying the scope of the effect: SarSim0-pretrained students do not uniformly dominate AutoARIMA, but tend to surpass it when (i) the real data distribution is more heterogeneous than a simple per-series ARIMA model can easily capture, and (ii) benefit can be gained from the broader inductive bias encoded by a globally pretrained neural network. We will clarify this in the revised manuscript to further emphasize that the phenomenon is domain-dependent rather than universal.
>
> > Question 4: The ablation study shows that removing the Noisers can sometimes improve performance on the M-Series. This suggests a mismatch between the noise model and the real-data characteristics. Does this finding point to a need for more adaptive or domain-specific noise generation, and how might that be incorporated into the pipeline?
>
> Thank you for this insightful question. In general, we agree that the noise generation procedure can be further improved. We deliberately did not attempt to tune the simulator hyperparameters to any of the evaluation benchmarks, to avoid leakage and to keep a single, benchmark-agnostic SarSim0 configuration. The three noise families we include (Poisson, generalized Gamma, lognormal) were chosen because they (i) are prominent in empirical studies of real-world data, and (ii) are largely orthogonal in terms of tail behaviour and dispersion. A more adaptive calibration of Noiser hyperparameters to a given domain (e.g., matching variance, over-dispersion, and zero fraction) is likely to further improve performance. A principled way to do this would be to define a meta-validation / meta-test benchmark: a small set of real "meta-validation" datasets, disjoint from the final meta-test benchmarks, would be used to adjust noise settings, and a separate meta-test suite would be used for reporting. Designing such a benchmark and running this second-level optimization is outside the scope of the present work, but we fully agree it is a promising direction and will highlight adaptive, domain-aware noise generation as an important avenue for future work in the revised manuscript.
>
> We would also like to point out that the interaction with the Noisers is subtle as it is both dataset- and model-dependent. For example, NBEATS-SarSim0 trained without Noisers (Table 2, row 6) performs very similarly to the full NBEATS-SarSim0 (Table 2, row 4) on the M-Series, so the effect is small for that backbone. For PatchTST, we found that the apparent underperformance of the Noiser variant on the M-Series is largely due to undertraining. The table below summarizes the results:
> |                       | GiftEval-sCRPS | GiftEval-MASE | M-Series-sCRPS | M-Series-MASE |
> | --------------------- |:--------------:|:-------------:|:--------------:|:-------------:|
> | PatchTST-SarSim0-250K, **No Noisers**      |     0.594      |     0.859     |     0.096      |     0.861     |
> | PatchTST-SarSim0-250K      |     0.576      |     0.838     |     0.102      |     0.922     |
> | PatchTST-SarSim0-500K      |     0.573      |     0.837     |     0.097      |     0.877     |
>
> With 250K training iterations, adding Noisers improves GIFT-Eval but hurts the M-Series. However, with 500K training iterations, performance on the M-Series recovers most of the gap while retaining strong results on GIFT-Eval. Our takeaway is twofold: (i) the effect of Noisers depends on the target domain **and** on the backbone/training budget; (ii) adding Noisers increases the complexity of the training distribution, so some models require more steps to fully "digest" and internalize this additional structure. We will add these new results and Noiser-model-domain interaction discussion to the revised manuscript.

---

> ### Author Response · Authors · 2025-11-25
> **Author response to Reviewer GRvF, 5/5**
>
> > Question 5: The current work focuses on univariate time series. What are the primary conceptual and technical hurdles to extending the SarSim0 methodology, particularly the stable pole sampling approach, to the multivariate case where capturing cross-series dependencies (e.g., via a VARIMA framework) is essential?
>
> Thank you for this insightful question. We agree that extending SarSim0 to multivariate series and VARIMA-type dynamics is both natural and important, but it raises several conceptual and technical challenges that we deliberately avoided in this first univariate study.
>
> Conceptually, the main hurdle is that one must now model not only the marginal dynamics of each series, but also structured cross-series dependencies. For example, shared seasonal components and lead-lag effects (e.g. marketing spend today -> sales a week later). There are many plausible ways to encode such structure (dense VAR, sparse VAR, factor models, block-diagonal plus low-rank, etc.), and different choices imply very different inductive biases. In this work we focused on answering a simpler question: whether a univariate SARIMA-based simulator is sufficient to obtain strong zero-shot performance on broad univariate benchmarks, before committing to a particular family of cross-series structures.
>
> We view designing a stable, structured VARIMA simulator as a natural next step: the pole-domain perspective we use here directly generalizes to the cross-series matrix case (e.g. via sampling stable eigenvalues of a block companion matrix), but choosing a realistic and computationally efficient structure for cross-series dependencies is non-trivial and beyond the scope of this initial univariate work. We will add a short paragraph in the discussion explicitly outlining these hurdles and pointing to multivariate extensions as an important direction for future research.
>
> > Question 6: The UMAP visualization in Figure 3 shows impressive overlap. However, are there any noticeable "holes" or regions in the real-data embedding space (red) that the synthetic data (blue) fails to cover? What kind of real-world patterns might these correspond to, and do they align with the known limitations of SARIMA models?
>
> Thanks also for this insightful question. Visually, we do not observe large, well-separated red clusters that are completely devoid of blue points. However, following your suggestion, we conducted a more systematic "novelty" analysis comparing the SarSim0/SARIMA distributions against M4-monthly. Specifically, we embedded windows and used a Local Outlier Factor detector to identify high-novelty real series relative to the synthetic distribution. The results are presented in Appendix G of the revised manuscript.
>
> A consistent and interpretable pattern emerged among the top-ranked novel real series across experimental settings. These windows typically correspond to (i) abrupt level shifts or structural breaks (e.g., changes in trend slope or, more generally, changes in the underlying generating process), (ii) strong, isolated spikes or outages that do not repeat seasonally (especially isolated, aperiodic negative spikes), and (iii) patterns with pronounced quasi-periodic calendar effects that are not captured by a purely seasonal-period-based SARIMA specification. These uncovered regions align well with the known limitations of single-regime SARIMA models and reinforce the reviewer’s suggestion that exploring extensions based on richer volatility and regime-switching structures (e.g., GARCH-style components) is a promising direction for future work.

---

> > ### Comment · Reviewer_GRvF · 2025-11-26
> >
> > Thank you for your detailed responses and additional experiments explanation.
> >
> > **Re: Overgeneralization and Linearity (Question 1)**
> > The new experiment on nonlinear dynamical systems is a necessary, albeit minimal, sanity check. While it demonstrates that the model does not catastrophically fail on chaotic systems, the performance is merely "in the same range" as real-data models, none of which are designed for this domain. This result does not prove the simulator's effectiveness for non-linear dynamics; it only shows that the learned features are not entirely useless. The core limitation remains: the simulator's inductive bias is fundamentally linear and seasonal, and the paper must be more explicit and upfront about this constraint on its applicability. The significance of the work is necessarily limited to domains where SARIMA-like dynamics are dominant.
> >
> > **Re: Simulator "Meta-Tuning" (Question 2)**
> > The provided justification for the hyperparameter settings in Table 8 is a post-hoc rationale, not a systematic analysis of robustness. Stating that choices were based on "prior knowledge" or were "non-informative" is insufficient. The paper lacks a proper sensitivity analysis for critical parameters like the AR pole radius or the bounds of the noiser distributions. Without this, the claim that the simulator configuration is a general, "benchmark-agnostic" setting is unsubstantiated. The current configuration could be a brittle "sweet spot," and the paper provides no evidence to the contrary, which is a methodological weakness.
> >
> > **Re: "Student-Beats-Teacher" Effect (Question 3)**
> > The argument that AutoARIMA's strength on M-Series is due to per-series fitting is a plausible hypothesis, but it also highlights a limitation of the "student" model. It suggests that the global pre-training on SarSim0 data is insufficient to match a specialized statistical model on its home turf. This weakens the "emergent generalization" claim by confining it to more heterogeneous, and perhaps noisier, benchmarks like GIFT-Eval. The phenomenon is therefore not a universal law of your method but a dataset-dependent outcome. This nuance must be stated more clearly to avoid overclaiming.
> >
> > **Re: Noiser Ablation (Question 4)**
> > The new experiment showing that the PatchTST model simply requires more training to benefit from the Noisers is a critical piece of information that was missing. However, this also reveals a negative interaction: adding the Noisers can significantly degrade performance if the training budget is insufficient. This is an important and negative practical result. It implies that using the full SarSim0 pipeline comes with a substantial cost in terms of sample complexity and that practitioners risk worse performance if they cannot afford to double their training budget. This trade-off must be explicitly discussed as a key limitation.
> >
> > **Re: Multivariate and Simulator "Holes" (Questions 5 & 6)**
> > The Local Outlier Factor analysis is a good addition and confirms what is already known: SARIMA models cannot capture structural breaks or aperiodic shocks. While this analysis is a useful confirmation, it does not change the fundamental limitations of the generative process. Acknowledging that the simulator cannot produce these critical real-world patterns underscores the limited scope of the current work.
> >
> > **Conclusion:**
> > While the additional experiments have clarified some points, several core methodological and practical concerns remain. The lack of a proper sensitivity analysis for the simulator's many hyperparameters is a significant weakness. The performance degradation observed with insufficient training for the Noiser module is a major practical caveat that needs to be highlighted. The claims regarding generalization must be more carefully scoped to the specific linear, seasonal domain that the simulator is designed for. The paper is a solid contribution within this narrow domain, but the broader claims about its robustness and general applicability are not fully supported by the evidence provided.

---

> > > ### Author Response · Authors · 2025-11-28
> > > **Author 2nd round response to Reviewer GRvF, 3/4**
> > >
> > > > Re: "Student-Beats-Teacher" Effect (Question 3) The argument that AutoARIMA's strength on M-Series is due to per-series fitting is a plausible hypothesis, but it also highlights a limitation of the "student" model. It suggests that the global pre-training on SarSim0 data is insufficient to match a specialized statistical model on its home turf. This weakens the "emergent generalization" claim by confining it to more heterogeneous, and perhaps noisier, benchmarks like GIFT-Eval. The phenomenon is therefore not a universal law of your method but a dataset-dependent outcome. This nuance must be stated more clearly to avoid overclaiming.
> > >
> > > Thank you for sharpening this point, we agree that the "student-beats-teacher" behaviour needs to be framed carefully. To address the concern we have implemented the following modifications.
> > >
> > > **Abstract**
> > >
> > > We now start with the sentence clearly specifying the scope:
> > >
> > > *We propose the first practical univariate time-series simulation pipeline which is scalable, fast enough for on-the-fly generation, and enables notable zero-shot forecasting performance on M-Series and GiftEval benchmarks that capture trend/seasonality/intermittency patterns typical of industrial forecasting applications across a variety of domains.*
> > >
> > > We also specifically attribute the student-beats-teacher effect to GiftEval:
> > >
> > > *Notably, on GiftEval we observe a "student-beats-teacher" effect: models trained on our simulations exceed the forecasting accuracy of the generating processes i.e. AutoARIMA, suggesting emergent generalization beyond the simulator's components.*
> > >
> > > **Introduction**
> > >
> > > We provide the following discussion of our contribution at the end of the Intro, carefully scoping our work, as you suggested:
> > >
> > > *(2) We provide a detailed empirical evaluation (Section 5) on the M-Series (Makridakis et al., 1982; Makridakis
> > > & Hibon, 2000; Makridakis et al., 2020; Athanasopoulos et al., 2011) and GIFT-Eval (Aksu et al.,
> > > 2024) benchmarks. Our scope is zero-shot forecasting in industrial settings where series are largely driven by complex trend, seasonality, and intermittency / heavy-tail patterns. M-Series and GIFT-Eval, which span multiple domains (Nature, Web/CloudOps, Sales, Energy, Transport, Healthcare, Demographic, Finance, Industry, Macro/Micro Economic) and frequencies (yearly, quarterly, monthly, weekly, daily, hourly, and some sub-hourly), provide a comprehensive testbed for this regime. We find that training exclusively on time series generated by our simulator yields compact, fast, and accurate cross-frequency models that generalize across these zero-shot benchmarks under a strict no-leakage guarantee. On these benchmarks, our simulated data can be competitive with, or even outperform, real-data pretraining, substantially closing the gap between small efficient architectures and large foundation models. On GiftEval we see early evidence for an emergent "student beats teacher" generalization behavior, where models trained on SARIMA source exceed the forecasting performance of AutoARIMA. On the more regular, relatively short and low-noise business and macroeconomic M-Series, this effect is mixed, with AutoARIMA remaining very strong. This indicates that such emergent generalization is dataset- and domain-dependent, and is currently most pronounced on more heterogeneous and noisier benchmarks like GiftEval.*
> > >
> > > **Discussion**
> > >
> > > We additoinally state the observed evidence more carefully to avoid overclaiming, as you suggested:
> > >
> > > *Additionally, on GiftEval we observe early, benchmark-specific evidence of a "student-beats-teacher" effect. Global neural models pretrained on SarSim0, whose core stochastic assumptions mirror the SARIMA family underlying AutoARIMA, can outperform this strong per-series AutoARIMA baseline. This suggests an emergent form of generalization beyond the simulator components, motivating further exploration of synthetic-first training for time-series foundation models.*

---

> > > ### Author Response · Authors · 2025-11-28
> > > **Author 2nd round response to Reviewer GRvF, 4/4**
> > >
> > > > Re: Noiser Ablation (Question 4) The new experiment showing that the PatchTST model simply requires more training to benefit from the Noisers is a critical piece of information that was missing. However, this also reveals a negative interaction: adding the Noisers can significantly degrade performance if the training budget is insufficient. This is an important and negative practical result. It implies that using the full SarSim0 pipeline comes with a substantial cost in terms of sample complexity and that practitioners risk worse performance if they cannot afford to double their training budget. This trade-off must be explicitly discussed as a key limitation.
> > >
> > > Thank you for highlighting this practical aspect.
> > >
> > > Our new PatchTST experiment indeed shows a compute-data trade-off: when we enrich the synthetic distribution with the Noisers, PatchTST requires a larger training budget to fully benefit from this additional variability. We also note that this pattern was already visible in the original results: PatchTST showed better generalization than NBEATS (a significantly lighter baseline) on GIFT-Eval, especially with Noisers in the training mix, whereas NBEATS was much less sensitive to including or excluding Noisers. There is clearly a noticeable model-dependent effect here. This aligns with broader experience in ML, where different architectures interact differently with a given training distribution. In this light, we do not view this as a "negative result", but as a familiar phenomenon: moving from a simpler to a richer training distribution (e.g., larger or more diverse corpora) often requires more compute to convert that additional variability into downstream gains. We agree that, from a practitioner’s point of view, this trade-off needs to be documented clearly.
> > >
> > > To address this, we have (i) added an additional row in ablation Table 2 and (ii) updated the caption as follows:
> > >
> > > *SarSim0 ablation, lower values are better. Takeaways: ... (2) Removing Noisers has dataset- and model-dependent effects. For PatchTST and limited train budget of 250K iterations, this hurts on GIFT-Eval, but helps on the more regular M-Series. For extended training budget of 500K iterations, PatchTST with Noisers recovers performance on the M-Series, while retaining stronger results on GIFT-Eval.*
> > >
> > > We have also revised the discussion of Table 2 as follows:
> > >
> > > *Removing the Noisers has clear model- and benchmark-dependent effects. For PatchTST, for instance, we find that more iterations (500K vs. 250K) are needed to fully benefit from Noisers. This points to a practical trade-off: under tight compute budgets, a simpler simulator configuration and backbone may be preferable, whereas with more generous budgets, the full SarSim0 pipeline and more expressive backbones yield, in our experiments, stronger overall zero-shot performance.*
> > >
> > > > Re: Multivariate and Simulator "Holes" (Questions 5 & 6) The Local Outlier Factor analysis is a good addition and confirms what is already known: SARIMA models cannot capture structural breaks or aperiodic shocks. While this analysis is a useful confirmation, it does not change the fundamental limitations of the generative process. Acknowledging that the simulator cannot produce these critical real-world patterns underscores the limited scope of the current work.
> > >
> > > Thank you for summarizing this concern. To address it, we have extended the discussion of the limitations of the current work by explicitly linking it to the results in Appendix G and by outlining promising future directions as follows:
> > >
> > > *There are many exciting directions for future work. For example, Appendix G presents a "novelty" analysis comparing the SarSim0/SARIMA distributions against M4-monthly using a Local Outlier Factor detector. A consistent and interpretable pattern emerges among the most novel real windows: they typically exhibit (i) abrupt level shifts or structural breaks (e.g., changes in trend slope or regime changes in the underlying process), (ii) strong, isolated spikes or outages that do not repeat seasonally (especially isolated aperiodic negative spikes), and (iii) pronounced quasi-periodic calendar effects that are not well captured by a purely seasonal-period-based SARIMA specification. This motivates natural extensions of SarSim0 based on richer volatility and regime-switching structures (e.g., GARCH-style components or regime-switching SARIMA), which we leave as important future work.*

---

> ### Author Response · Authors · 2025-11-28
> **Author 2nd round response to Reviewer GRvF, 1/4**
>
> > Conclusion: While the additional experiments have clarified some points, several core methodological and practical concerns remain. The lack of a proper sensitivity analysis for the simulator's many hyperparameters is a significant weakness. The performance degradation observed with insufficient training for the Noiser module is a major practical caveat that needs to be highlighted. The claims regarding generalization must be more carefully scoped to the specific linear, seasonal domain that the simulator is designed for. The paper is a solid contribution within this narrow domain, but the broader claims about its robustness and general applicability are not fully supported by the evidence provided.
>
> Thank you for summarizing your remaining concerns and for characterizing the paper as a solid contribution within its domain. We have revised the manuscript to directly address the points you highlight:
>
> - we added a sensitivity analysis for key simulator hyperparameters (see response to *Simulator "Meta-Tuning" (Question 2)*)
> - we now explicitly describe the compute-data trade-off introduced by the Noisers, update the ablation table and caption (see response to *Noiser Ablation (Question 4)*)
> - we have carefully clarified the scope of our claims as you suggested and revised the abstract, introduction and discussion accordingly (see responses to *Overgeneralization and Linearity (Question 1)*, *Multivariate and Simulator "Holes" (Questions 5 & 6)*, *"Student-Beats-Teacher" Effect (Question 3)*)
>
> With these changes, we believe the paper now directly addresses your remaining concerns and incorporates your feedback into both the methodology and the presentation. We sincerely thank you for the detailed and insightful comments that led to these improvements, and we would be very grateful if you could consider raising your score in light of the revised manuscript.
>
>
> > Re: Overgeneralization and Linearity (Question 1) The new experiment on nonlinear dynamical systems is a necessary, albeit minimal, sanity check. While it demonstrates that the model does not catastrophically fail on chaotic systems, the performance is merely "in the same range" as real-data models, none of which are designed for this domain. This result does not prove the simulator's effectiveness for non-linear dynamics; it only shows that the learned features are not entirely useless. The core limitation remains: the simulator's inductive bias is fundamentally linear and seasonal, and the paper must be more explicit and upfront about this constraint on its applicability. The significance of the work is necessarily limited to domains where SARIMA-like dynamics are dominant.
>
> To address the concern we have added Appendix H.1 summarizing the non-linear system results as well as the following asociated discussion at the end of Empirical Results section:
>
> *To probe behaviour far outside our target regime of business-style series, we also ran a zero-shot experiment on four canonical nonlinear dynamical systems (Duffing, Lorenz, Lotka–Volterra, pendulum), comparing several Chronos variants, TimesFM, and SarSim0-pretrained models (see Appendix H.1 for details). SarSim0-pretrained models attain error levels comparable to real-data foundation models. This suggests that on these systems, SarSim0’s inductive biases, while geared toward trend/seasonality/intermittency regimes typical of industrial forecasting, yield models whose out-of-domain generalization is similar to that of real-data-pretrained models.*
>
> We have further clarified the scope and target application domain of our work in abstract, intro and discussion as described in responses to *Overgeneralization and Linearity (Question 1)*, *Multivariate and Simulator "Holes" (Questions 5 & 6)*, *"Student-Beats-Teacher" Effect (Question 3)*.

---

> ### Author Response · Authors · 2025-11-28
> **Author 2nd round response to Reviewer GRvF, 2/4**
>
> > Re: Simulator "Meta-Tuning" (Question 2) The provided justification for the hyperparameter settings in Table 8 is a post-hoc rationale, not a systematic analysis of robustness. Stating that choices were based on "prior knowledge" or were "non-informative" is insufficient. The paper lacks a proper sensitivity analysis for critical parameters like the AR pole radius or the bounds of the noiser distributions. Without this, the claim that the simulator configuration is a general, "benchmark-agnostic" setting is unsubstantiated. The current configuration could be a brittle "sweet spot," and the paper provides no evidence to the contrary, which is a methodological weakness.
>
> Thank you for summarizing the concern. To address it, we executed the sensitivyty study by training NBEATS-SarSim0 on different hyperparameter configurations of SarSim0. We summarized our results in Table 12 of Appendix H.2 and also provide this table below for convenience. We have also added a short discussion of the simulator sensitivity study in the Ablation section of the main body:
>
> *In addition, we performed the SarSim0 hyperparameter sensitivity study (see Table 12 in Appendix H.2 for details). Across studied configurations, performance on both GIFT-Eval and the M-Series remains very similar to the default configuration, indicating that SarSim0 is not a brittle "sweet spot". The configuration we use appears to be benchmark-agnostic and robust to substantial hyperparameter changes.*
>
> |                | GiftEval-sCRPS | GiftEval-MASE | M-Series-sCRPS | M-Series-MASE |
> | -------------- |:--------------:|:-------------:|:--------------:|:-------------:|
> | NBEATS-SarSim0-default |     0.602      |     0.849     |     0.096      |     0.869     |
> | NBEATS-SarSim0-p=3 |     0.603      |     0.851     |     0.097      |     0.876     |
> | NBEATS-SarSim0-p=7 |     0.600      |     0.846     |     0.096      |     0.867     |
> | NBEATS-SarSim0-p=15 |     0.599     |     0.850     |     0.097      |     0.869     |
> | NBEATS-SarSim0-p=20 |     0.602     |     0.852     |     0.097      |     0.872     |
> | NBEATS-SarSim0-rmax=0.8 |     0.601      |     0.848     |     0.096      |     0.870     |
> | NBEATS-SarSim0-rmax=0.95 |     0.600      |     0.847     |     0.097      |      0.873    |
> | NBEATS-SarSim0-Rmax=0.2 |      0.601     |     0.844    |   0.096        |    0.869      |
> | NBEATS-SarSim0-Rmax=0.5 |    0.598       |   0.846     |     0.096      |      0.866    |
> | NBEATS-SarSim0-q=6 |    0.604       |   0.848     |     0.097      |      0.875    |
> | NBEATS-SarSim0-Q=4 |    0.602       |   0.851     |     0.097      |      0.871    |

---

### Official Review · Reviewer_jA3h · 2025-10-31

**Soundness:** 2
**Presentation:** 3
**Contribution:** 3
**Rating:** 6
**Confidence:** 3

**Summary:**

The authors propose a scalable synthetic time-series generation pipeline based on a seasonal autoregressive integrated moving average (SARIMA) model. Their proposed simulator follows a three-step procedure that samples traces from the “polynomial stability region”, then combines multiple paths into rich multi-seasonality traces, then adds noise to capture burstiness and intermittency. They show that their simulator allows for strong zero-shot generalization.

**Strengths:**

- The work is well-motivated. The authors point out how training with real series can be limited by licensing barriers, data scale, domain etc. And that training with synthetic data offers unique levers that can be controlled.

**Weaknesses:**

- There is very limited analysis conducted with the experiments. First, GIFT-Eval allows for easy analysis of performance stratified by domain/frequency/term length/ variate type etc. These are more important to understand the limitations and strengths of the model, more than the aggregate score on GIFT-Eval.

- To further improve the evaluation, I would suggest adding one of the foundation model baselines (e.g. Chronos) trained from scratch on KernelSynth/ForecastPFN/SarSim0.
Then it would allow for apples-to-apples comparison with the pretrained Chronos.

- There is no analysis on how this dataset can be used to complement other datasets, which is important to understand. I suggest the authors combine GIFT-Eval train (or a different training set) with this training dataset.


**Final verdict**: I give a score of 6 as I see this as a paper that can be accepted. Conditional on addressing the weaknesses and answering questions, l might consider upgrading my score to 8 (if I am satisfied with the answers).

**Questions:**

1. Regd the second set of rows in Table 1: What is NBEATS, PATCHTST and AutoARIMA trained on here? It should be made clear in the table.
2. Suggestion: can the authors bold the best numbers in each column, say in Table 1, 2 etc.? It is difficult to spot which models are the best

---

> ### Author Response · Authors · 2025-11-24
> **Author response to Reviewer jA3h, 1/2**
>
> We sincerely appreciate your thoughtful and supportive feedback. We provide our detailed response below. If you have any additional questions based on our response, please do not hesitate to let us know.
>
> > There is very limited analysis conducted with the experiments. First, GIFT-Eval allows for easy analysis of performance stratified by domain/frequency/term length/ variate type etc. These are more important to understand the limitations and strengths of the model, more than the aggregate score on GIFT-Eval.
>
> We agree that GIFT-Eval is particularly valuable because it supports stratified analysis, and we appreciate the suggestion to better characterize where SarSim0 helps and where it may be less beneficial. In the revised version, we will therefore add a stratified analysis of GIFT-Eval along the axes suggested by the reviewer: domain/frequency/term length/variate type.
>
> > To further improve the evaluation, I would suggest adding one of the foundation model baselines (e.g. Chronos) trained from scratch on KernelSynth/ForecastPFN/SarSim0. Then it would allow for apples-to-apples comparison with the pretrained Chronos.
>
> Thank you for this suggestion. We agree that training one of the foundation model baselines, such as Chronos, on the proposed data generator provides additional valuable support for our procedure. We will therefore evaluate the Chronos model trained from scratch on KernelSynth/ForecastPFN/SarSim0 under the same zero-shot protocol on GIFT-Eval and the M-Series, and report the results in new rows in Table 1. Inspired by your comments, we currently have the following preliminary result available. We took the Chronos-Small T5 backbone and dropped it into our pipeline (using our sequence sampling and loss definition) and trained with batch size 1024 for 500K steps on SarSim0 using input sequence length 512 for compatibility. Our results appear below and confirm that Chronos-Small can be trained on the proposed simulator to achieve accuracy comparable to the original approach trained on a mix of real and synthetic data. The additional results of training Chronos-Small on KernelSynth/ForecastPFN will be posted once they are ready.
>
> |                       | GiftEval-sCRPS | GiftEval-MASE | M-Series-sCRPS | M-Series-MASE |
> | --------------------- |:--------------:|:-------------:|:--------------:|:-------------:|
> | Chronos-Small         |     0.662      |     0.892     |     0.103      |     0.882     |
> | Chronos-Small-SarSim0 |     0.634      |     0.921     |     0.098      |     0.886     |
> | PatchTST-SarSim0      |     0.576      |     0.838     |     0.102      |     0.922     |
> | NBEATS-SarSim0        |     0.602      |     0.849     |     0.096      |     0.869     |

---

> ### Author Response · Authors · 2025-11-24
> **Author response to Reviewer jA3h, 2/2**
>
> > There is no analysis on how this dataset can be used to complement other datasets, which is important to understand. I suggest the authors combine GIFT-Eval train (or a different training set) with this training dataset.
>
> Thank you for this comment, we now clarify our position.
>
> In the current work we do not mix SarSim0 data with real data. This is deliberate: our primary goal is to study a strict synthetic-only, leakage-free time series pretraining and zero-shot inference, where the simulator is never exposed to real evaluation series (or closely related data), and to understand how far one can go in that regime.
>
> There is already evidence in the literature that mixing real and synthetic data can be beneficial. For example, Chronos combines GP-based synthetic data (KernelSynth) with large real-data corpora, reporting improved performance compared to using real data alone. Showing that SarSim0 can also be helpful as part of such a mixed synthetic-real curriculum would therefore be a natural but relatively incremental extension of previous results, and would not change the core message of our paper.
>
> Instead, we chose to focus on a purely synthetic pretraining and to evaluate it against real-data baselines. In Table 1, we compare to models trained on real data both in the zero-shot regime (Chronos, TimesFM, MOIRAI, TTM, etc., pretrained on large real corpora) and in the per-dataset regime (NBEATS/PatchTST trained directly on each benchmark), and find that SarSim0-only pretraining yields models that are competitive with, and often better than, these real-data alternatives on GIFT-Eval and the M-Series. We view this synthetic-only competitiveness as the main contribution of our work.
>
> We fully agree that hybrid training (e.g., synthetic pretraining followed by real fine-tuning, or joint synthetic-real curricula) is a promising practical direction. We expect that in many real-world settings such mixtures will surpass either pure synthetic or pure real training alone. However, systematically exploring mixing ratios, schedules, and domain-specific trade-offs would require a substantial separate study and would blur the clean zero-shot setting we focus on here. We will clarify this in the discussion/limitations section and leave a systematic study of synthetic-real co-training as an important direction for future work.
>
> > Regd the second set of rows in Table 1: What is NBEATS, PATCHTST and AutoARIMA trained on here? It should be made clear in the table.
>
> We thank the reviewer for catching this ambiguity. In the second block of rows in Table 1, AutoARIMA is fitted separately for each test series, using only the historical observations preceding the forecasting horizon. It is therefore a per-time-series baseline. NBEATS and PatchTST are trained from scratch on the training split of each benchmark dataset, and then evaluated on the respective test splits. They are therefore per-dataset baselines, not zero-shot models. We will clarify this explicitly in the caption of Table 1 and in Section 5.2 to avoid confusion.
>
> > Suggestion: can the authors bold the best numbers in each column, say in Table 1, 2 etc.? It is difficult to spot which models are the best
>
> Thank you for this suggestion. In the revised manuscript, we will bold the best value in each column of Tables 1, 2.

---

> ### Author Response · Authors · 2025-12-02
> **Author 2nd round response to Reviewer jA3h, summary of revisions**
>
> > There is very limited analysis conducted with the experiments. First, GIFT-Eval allows for easy analysis of performance stratified by domain/frequency/term length/ variate type etc. These are more important to understand the limitations and strengths of the model, more than the aggregate score on GIFT-Eval.
>
> In line with this suggestion, the revised manuscript now includes Appendix H.3, which reports accuracy on GiftEval stratified by domain, sampling frequency, term length, and variate type. We also explicitly reference this appendix in the main text and summarize the key trends as follows:
>
> *Beyond aggregate scores, we stratify GiftEval results by domain, sampling frequency, term length, and variate type (Appendix H.3). Across all splits, SarSim0-pretrained backbones are consistently competitive with, and often stronger than, real-data foundation models. At the same time, SarSim0 uniformly outperforms prior synthetic generators (KernelSynth, ForecastPFN) and often matches or surpasses strong per-dataset supervised NBEATS/PATCHTST baselines, indicating that a single, domain-agnostic SARIMA-based curriculum transfers robustly across GiftEval's diverse splits.*
>
> > To further improve the evaluation, I would suggest adding one of the foundation model baselines (e.g. Chronos) trained from scratch on KernelSynth/ForecastPFN/SarSim0. Then it would allow for apples-to-apples comparison with the pretrained Chronos.
>
> Thank you for this constructive suggestion. In the revised manuscript, we now include Chronos-Small models pretrained on KernelSynth, ForecastPFN, and SarSim0 in Table 1. Concretely, we take the Chronos-Small T5 backbone, plug it into our training pipeline (our sequence sampling and loss), and train it for 500K steps with batch size 1024 on SarSim0. These additional experiments reinforce our earlier conclusions: (i) on GiftEval and the M-Series benchmarks, Chronos-Small trained on SarSim0 achieves accuracy comparable to or better than Chronos-Small pretrained on real data; and (ii) the same Chronos-Small backbone trained on SarSim0 generalizes better than when trained on KernelSynth or ForecastPFN on both benchmarks.
>
> > Regd the second set of rows in Table 1: What is NBEATS, PATCHTST and AutoARIMA trained on here? It should be made clear in the table.
>
> In the revised manuscript, we have included the following clarification in the caption of Table 1:
>
> *In the second block, AutoARIMA is fitted on each test series, while NBEATS and PATCHTST are trained from scratch on the training split of each dataset (inference time with $^\star$ includes training time)*
>
> > Suggestion: can the authors bold the best numbers in each column, say in Table 1, 2 etc.? It is difficult to spot which models are the best
>
> Thank you for this suggestion. In the revised manuscript, we bold the best value in each column of Tables 1, 2.

---

### Meta-Review · Area_Chair_DBAu · 2026-01-07

**Summary:**

The paper presents an intriguing finding that time series forecasting models can be learned from simulated data generated by a simple and efficient simulator, while achieving competitive zero-shot generalization performance. The initial reviews were mixed. Most reviewers found the core finding significant and acknowledged the strong empirical results, while also raising several concerns. Two major issues shared across reviewers were: (1) the need for more extensive empirical validation (e.g., stratified analyses and additional baselines), and (2) limitations of the simulator, including its inability to model nonlinearity, the degree of tuning required, and the possibility that it may be over-designed for the target datasets. Reviewer qasc also noted that training zero-shot forecasting models purely on simulated data has been explored in prior work and requested a clarification of the core contribution. Many if not all of these concerns have been addressed in the rebuttal. After considering the paper, reviews, and rebuttal, and in light of the overall positive evaluations, the AC recommends acceptance.

**Reviewer Concerns:**

The authors provided an extensive rebuttal and further revised the paper. The rebuttal, including additional clarifications and experimental results, addressed most of the reviewers' concerns. During the early discussion, Reviewer qasc indicated that the rebuttal resolved all outstanding issues and clarified "a very compelling point" regarding the paper's novelty that had been "underappreciated" in the initial review. Reviewer jA3h also expressed a willingness to raise the score to 8, if the rebuttal adequately addressing the weaknesses and questions.

**Reviewer Scores:**

Based on the reviews and rebuttal, the area chair expects that most reviewers will raise their scores.

---

### Decision · Program_Chairs · 2026-01-26

Accept (Poster)